# Estimating the Snow Density using Collocated Parsivel and MRR Measurements: A Preliminary Study from ICE-POP 2017/2018

Wei-Yu Chang[1], Yung-Chuan Yang[1], Chen-Yu Hung[1], Kwonil Kim[2], Gyuwon Lee[3], Ali Tokay[4,5]

[1]Department of Atmospheric Sciences, National Central University, Taoyuan, Taiwan
[2]School of Marine and Atmospheric Sciences, Stony Brook University, NY, USA
[3]Department of Atmospheric Sciences, Center for Atmospheric REmote sensing (CARE), Kyungpook National University, Daegu, Republic of Korea
[4]Goddard Earth Sciences Technology and Research (GESTAR-II), University of Maryland, Baltimore County, Baltimore, Maryland, USA
[5]NASA Goddard Space Flight Center, Greenbelt, Maryland, USA

*Correspondence to*: Wei-Yu Chang (wychang@g.ncu.edu.tw)

**Abstract.** A new method is developed to derive bulk density and bulk water fraction of a population of particles from collocated measurements from Micro-Rain Radar (MRR) and Particle Size and Velocity disdrometer (Parsivel). Rigorous particle scattering simulation, namely the T-matrix method, is applied to Parsivel's particle size distribution data to calculate reflectivity ($Z_{HH}$). The possible combinations of the particle's ice, air, and water are derived to compare them with the MRR-measured $Z_{HH}$. The combination of minimum water fraction and maximum ice fraction subsequently determines the bulk density ($\rho_{bulk}$). The proposed method is applied to the data collected from the International Collaborative Experiments for Pyeongchang 2018 Olympic and Paralympic Winter Games (ICE-POP 2018) Projects and its pre-campaign. The estimated $\rho_{bulk}$ was examined independently by comparison of the liquid-equivalent snowfall rate (SR) of collocated Pluvio. The bias values are adequately low (SR: -0.25~0.06 mm hr$^{-1}$). The retrieved bulk density also shows good consistency with collocated Precipitation Imaging Package (PIP) retrievals. The results indicate the capability of the proposed algorithm to derive reliable $\rho_{bulk}$, leveraging the compact and easily deployable designs of MRR and Parsivel. The derived bulk density of the two warm-low cases (28 February and 07 March 2018) shares a similar transition as the systems were decaying. The higher bulk density and bulk water fraction were found in the coastal sites (BKC and GWU: median value of $\rho_{bulk}$ are 0.05 to 0.12 g cm$^{-3}$), typically accompanied by higher liquid-water constituents (mean values of the top 5% bulk water fraction are 0.07 to 0.45) than the inland sites (YPO and MHS: median value of $\rho_{bulk}$ are 0.06 to 0.10 and mean values of the top 5% bulk water fraction are 0.001 to 0.008) during such synoptic conditions.

## 1 Introduction

The particle size distributions (PSD) and physical properties of hydrometers in winter storms are essential to discriminate among hydrometeor types (e.g., rimed particles) and develop algorithms (e.g., quantitative precipitation estimation of snow) for estimating the liquid/ice water content (LWC/IWC) of hydrometer with remote sensors (e.g., polarimetric radar and

satellite). Radar-based quantitative precipitation estimation (QPE) of liquid equivalent snowfall rate (SR) from equivalent reflectivity (Ze) inheres great uncertainty due to the diversity of snow properties (Huang et al. 2010, 2015, and 2019). Pre-assumed snow density in Ze-SR relation is one of the critical factors. The snow density caused by various degrees of riming,

melting, and aggregation processes is essential to derive the Ze-SR relation (Huang et al. 2014).

These physical properties, including terminal fall speed, shape, composition, and density, are also crucial to verifying and improving microphysical parameterizations in numerical forecast models (Yuter et al. 2006; Kim et al. 2021). Simulating proper riming, freezing, and aggregation processes is challenging in numerical models. The riming processes led to snowfall velocity and density diversity in the same particle diameter size (Zhang et al. 2021). The supercooled liquid water freezes on

snow particles and fills in the holes of snow; therefore, snow's mass and fall velocity increase while size has little changes (Heymsfield 1982; Moisseev et al. 2018). On the other hand, in a warm environment, the melting process induces higher density as well as higher fall velocity by the aerodynamic process. The dispersed fall velocity caused by particles melted from dry snow to rain leads to higher collision efficiency and facilitates the aggregation and accretion processes (Yuter et al. 2006). Higher snow density is associated with steeper fall velocity-diameter (V-D) relations, which result from stronger riming (Lee

et al. 2015) or melting processes (Yuter et al. 2006). To emulate the diverse physical properties of hydrometers, Morrison and Milbrandt (2015) proposed a new bulk method to parameterize ice-phase particles with evolvable density to study the role of density in numerical simulation. A robust density estimation algorithm can evaluate microphysical simulations from numerical models.

As the snow density cannot be measured directly, various techniques with diverse measuring principles have been

developed to investigate snow density using observational data. Brandes et al. (2007) estimated the bulk snow density from a two-dimensional video disdrometer (2DVD) derived precipitation volume and the collocated gauge-measured precipitation mass. A bulk snow density and median volume diameter ($D_0$) relation is obtained. Nevertheless, various factors influence the density of frozen precipitation, which cannot be well illustrated by size-density relation (Roebber et al. 2003); Zhang and Luchs (2011) proposed a terminal velocity–based modification to the density value derived from the equation for terminal fall

velocity (Pruppacher and Klett 1997). Several studies have proven that after differentiating the degree of riming, the density of snow can be derived correctly from the corresponding size (Li et al. 2018; Zhang et al. 2021; Lee et al. 2015; Zhang and Luchs 2011). An alternative approach is to use radar reflectivity to constrain the estimated bulk snow density. Huang et al. (2010) utilized a 2DVD to derive snow's particle size distribution (PSD) and a C-band dual-polarimetric radar to obtain the reflectivity above 2DVD. The measured reflectivity (Ze) was subsequently used to estimate the snow density by minimizing

the difference between the measured Ze from radar and calculated Ze from 2DVD. Wood et al. (2014) utilized the Bayesian optimal estimation retrieval method. Snow microphysical parameters are retrieved from near-Rayleigh radar reflectivity, particle size distribution, snowfall rate, and size-resolved particle fall speeds.

Other sophisticated instrumentations are developed to investigate the microphysical characteristics of snow particles. The precipitation imaging package (PIP), a video disdrometer, provides the PSD, fall speed, density, and snowfall rate of

hydrometers (Newman et al., 2009; Pettersen et al., 2020). The Multi-Angle Snowflake Camera (MASC) captures high-

resolution photographs of hydrometeors from three angles while simultaneously measuring their fall speed (Garrett et al. 2012). Mroz et al. (2021) proposed an algorithm utilizing triple-frequency (X, Ka, W) radar measurements to retrieve the size, ice water content (IWC), and degree of riming of ice clouds. The results indicate that the mass-weighted diameters ($D_m$) and IWC estimates are adequately accurate. Yet, the degree of riming remains challenging.

Even though the 2DVD provides state-of-the-art hydrometer particle observation, the 2DVD is challenging to maintain and not ideal for continuous unattended operation (Tokay et al. 2017). A viable alternative method utilizing collocated Micro Rain Radar (MRR, Löffler-Mang et al. 1999) and Particle Size and Velocity (Parsivel, Loffler-Mang and Joss 2000) disdrometer is proposed in this study to derive the bulk snow density. Parsivel and MRR are reliable, robust, easy to maintain, and relatively affordable. Hence, they are widely used in the research community and typically collocated. The MRR is a

vertically pointing frequency modulated continuous wave (FMCW) radar at 24.23 GHz. The radar transmits radiation vertically upward, and the hydrometeor above scatters a portion of the energy back to the antenna. The magnitude and frequency of the backscatter signal provide the vertical profiles of reflectivity ($Z_{HH}$) and the reflectivity weighted fall velocity ($V_Z$) (Kneifel et al. 2011). Parsivel is a laser-based optical disdrometer for simultaneous particle size and velocity measurements. As the hydrometeor passes through the laser beam generated by the transmitter, particle extinction leads to a decrease in the energy

detected by the receiver. Hence, the particle size is derived from the amplitude of the energy reduction. The particle velocity is subsequently derived from the energy reduction duration and particle size.

      Similar to Huang et al. (2010), the measured $Z_{HH}$ from MRR is applied to constrain the probability of volume ratio of ice and liquid water (vi and vw) on the simulated $Z_{HH}$ from particle size distribution (PSD) of Parsivel in the proposed method. The bulk density of snow ($\rho_{bulk}$) is consequently derived from the vi and vw, corresponding to the most consistently measured

and simulated $Z_{HH}$. Subsequently, the measurement of $Z_{HH}$ weighted fall velocity ($V_Z$) from MRR is compared with the calculated $V_Z$ from the derived bulk density and Parsivel PSD measurement. The inconsistency between the measured and calculated $V_Z$ identifies the possible attenuation effect on MRR reflectivity measurements. Finally, independent measurements of the liquid-equivalent snowfall rate (SR) observed by Pluvio evaluate the bulk snow density from the proposed method.

      The main objective of this study is to evaluate the algorithm as introduced earlier. The data of the microphysical

instruments, namely MRR, Parsivel, and Pluvio, from the International Collaborative Experiments for Pyeongchang 2018 Olympic and Paralympic Winter Games (ICE-POP 2018) Projects and its pre-campaign is applied to estimate the bulk density of snow. The performance and applicability of the proposed algorithm are examined using the ICE-POP data. The instruments and data are introduced in section 2. The methodology is detailed in section 3. The results are summarized in section 4. Finally, the conclusion is shown in section 5.

**2 Instruments and Data Processing**

      The data of MRR, Parsivel, and Pluvio (OTT Pluvio² - Weighing Rain Gauge) were collected during the ICE-POP 2018 (2017/2018 winter) and the pre-ICE-POP campaign (2016/2017 winter). The instruments were located in nineteen sites across

the Gangwon region on the east coast of Korea (see Kim et al. 2021 for detailed information of each site). Five sites with collocated MRR and Parsivel were available for this study. These sites aligned across the Taebaek Mountains from mountain to coast are YPO (YongPyong Observatory, 772 m MSL), MHS (MayHills Supersite, 789 m MSL), CPO (Cloud Physics Observatory, 855 m MSL), BKC (BoKwang 1-ri Community Center, 175 m MSL), and GWU (Gangneung-Wonju National University, 36 m MSL), respectively. The YPO, MHS, and CPO sites are in the mountainous region, while GWU and BKC sites are in the coastal area (Kim et al. 2021). All of the Pluvios were equipped with double windshields. The Pluvios at MHS, BKC, and GWU were equipped with a double windshield with inner Tretyakov and outer Alter shields. The Pluvio at YPO was equipped with a Belfort double Alter windshield. The Pluvio at the MHS was within the DFIR (double fence intercomparison reference) in addition to the double shield. All the sites investigated in this study have no taller trees or buildings near the MRR antenna and Parsivel. Each site's detailed layout and information can be found in Kim et al. (2021). The available data from these five sites during the pre-campaign and ICE-POP campaign are listed in Table 1.

The snow observation from Parsivel suffers from various issues due to the measuring principle (Battaglia et al. 2010; Wood et al. 2013). Friedrich et al. (2016) indicate that Parsivel can suffer from splashing of particles (observed as a small diameter with large fall velocity when particles fall on the head of the sensor) and margin fallers (observed as a faster velocity than true fall velocity when particles fall through the edge of the sampling area). The one-minute Parsivel data was quality-controlled using the fall velocity filtering technique (Lee et al. 2015). The mean fall velocity and standard deviation ($\sigma$) for a given diameter were calculated, and the particles that deviate from the mean fall velocity of more than one standard deviation were filtered. The quality-controlled Parsivel data was subsequently processed to derive the PSD.

The MRRs had the same configuration during ICE-POP; the vertical resolution was 150 m, and there were 31 gates up to 4.65 km. The MRR data was post-processed using the algorithm from Maahn and Kollias (2012). The sensitivity of MRR has been enhanced, and the Doppler velocity has been dealiased. The third gate (450 m above ground) of $Z_{HH}$ data from MRR was selected to retrieve snow density since the first two gates contain some clutter contamination.

The MRR and Parsivel have different measuring principles and designs. The measurement inconsistency between collocated MRR and Parsivel degrades the accuracy of estimating the bulk density. To ensure the observation consistency between MRR and Parsivel data and to minimize the measurement bias, the pure rain precipitation events (13-14 April 2018 and 22-23 April 2018) were selected to calculate the bias in rainfall rate between MRR and Parsivel. The data was quality controlled by examining the rainfall rate of the MRR, Parsivel, and collocated Pluvio. The rainfall rate measurements from Parsivel and Pluvio were consistent with each other. Thus, the Parsivel PSD data was applied to the T-Matrix simulation to obtain $Z_{HH}$. The bias was derived after excluding one standard deviation outlier data. The bias values of each MRR are listed in Table 2. The results indicate that the MRR consistently underestimated the reflectivity from 2.1 to 10.2 dBZ. The standard deviation between MRR reflectivity and Parsivel calculated reflectivity is about 1.1 to 1.3 dB for each site. All of the MRR data have been bias-corrected by applying the bias values listed in Table 2.

## 3 Methodology

Hydrometeor is composed of particles with combinations of solid ice and liquid water with a density of 0.92 ($\rho_{ice}$) and 1.0 ($\rho_{water}$) g cm$^{-3}$, respectively. Therefore, the hydrometeor bulk density ($\rho_{bulk}$) can be determined by its volume ratio of solid ice (vi) and liquid water (vw) as follows,

$$\rho_{bulk} = \text{vi} \times 0.92 + \text{vw}; \text{ g } cm^{-3}. \tag{1}$$

The sum of the values of vi and vw equals one or less than one if it contains air in the particle. Thus, the reflectivity factor ($Z_{HH}$) can be calculated as follows (Bringi and Chandrasekar 2001),

$$Z_{HH} = \left(\frac{\rho_{bulk}}{\rho_{ice}}\right)^2 \frac{|K_{ice}|^2}{|K_w|^2} \frac{\lambda^4}{\pi^5} \int \sigma(D)N(D)dD; \, mm^6 \, m^{-3}. \tag{2}$$

The $K_{ice}$ and $K_w$ are the dielectric factors of solid ice and liquid water, respectively. $\sigma$ is the backscattering cross-section, $D$ is the particle size, and $N(D)$ is the particle size distribution. As shown in (1) and (2), the $Z_{HH}$ is positively correlated to $\rho_{bulk}$. The higher hydrometer bulk density has a higher value of $Z_{HH}$ for a given PSD. Hence, the $Z_{HH}$, the factor relating to the vi and vw of hydrometeor, can estimate the bulk density. Huang et al. (2010) utilized the C-band radar measurements on top of a 2DVD. The $\rho_{bulk}$ was derived from (2) by applying the reflectivity from scanning C-band radar and the PSD from 2DVD.

The bulk density estimation algorithm developed in this study is modified from Huang et al. (2010). Instead of scanning C-band radar and 2DVD, the collocated MRR (Micro Rain Radar, Löffler-Mang et al. 1999) and Parsivel are proposed to minimize the sampling size inconsistency. The estimated density is considered as "bulk" or "equivalent" density since the MRR $Z_{HH}$ measurement is the summation of all hydrometeor within the sampling volume. The procedures of the proposed method are introduced in the following section, and the validation and discussion are described in the next section.

The $Z_{HH}$ values were simulated from each Parsivel PSD measurement. Each $Z_{HH}$ value was calculated using a rigorous T-matrix method with specified vi and vw (Vivekanandan et al. 1991; Bringi and Chandrasekar 2001). The ice and water are assumed to be evenly distributed within the particle. The T-matrix method is a fast numerical solution of Maxwell's equations to compute the scattering properties of particles. The shape of the hydrometeor is regarded as a symmetric sphere since the $Z_{HH}$ measurement of the hydrometer was observed from the bottom of the snow particle by vertical pointing MRR. No canting angle is considered. The sensitivity of the particle shape to bulk density retrieval will be investigated in the discussion section.

An example of simulated $Z_{HH}$ from Parsivel observed snow PSD via T-matrix simulation with different combination of vi/vw and temperature is shown in Fig. 1. The results indicate that the simulated $Z_{HH}$ values remain nearly identical when varying the temperature from -10 to 0 $^0$C. On the other hand, the simulated $Z_{HH}$ varies significantly when altering the composition of vi/vw. The lowest (highest) value of $Z_{HH}$ was from the combination of vi/vw of 1.0/0.0 (0.0/1.0), which was pure ice (rain) with a density of 0.92 (1.0) g cm$^{-3}$. The particle temperature was consequently assumed to have a constant value of 0℃ in the following $Z_{HH}$ T-matrix simulation. On the other hand, all possible combinations of vi and vw ranging from 0.0 to 1.0 were included in the T-matrix simulation of $Z_{HH}$.

A selected example of the simulated $Z_{HH}$ from observed PSD with various combinations of vi/vw is shown in Figure 2. The observed PSD from Parsivel (Fig. 2a) was applied to the T-Matrix backscattering simulation. All possible combinations of

vi/vw were applied to calculate simulated $Z_{HH}$ (Fig. 2b). The corresponding bulk density was derived via (1) and shown as contour dash lines in Fig. 2b. The values of simulated $Z_{HH}$ vary from 0 to 50 dBZ (shaded color in Fig. 2b). The simulated $Z_{HH}$

values increase with increasing of the bulk snow density. In this selected case, the observed $Z_{HH}$ from MRR was 22.23 dBZ (dashed blue line in Fig. 2b). The observed $Z_{HH}$ from MRR was thus applied to constrain the possible combination of vi/vw and bulk density. The possible ranges of vi/vw are 0.0/0.009 to 0.08/0.0, shown as a dashed blue line in Fig. 2b. The corresponding bulk densities are 0.009 and 0.074 (g cm$^{-3}$), respectively. The higher the fraction of ice (e.g., vi), the higher bulk snow density values can be found in Fig. 2b. To determine the bulk snow density from possible combinations of vi/vw, the

maximum bulk density with maximum vi is selected. Choosing the bulk density with maximum vi ensures the minimum value of vw. This assumption is similar to Huang et al. (2010), which assumes that a mixture of snow contains only ice and air. The water fraction is not considered in Huang et al. (2010). Therefore, the contour's maximum density in Fig. 2(b), 0.074 g cm$^{-3}$, is determined by assuming vi and vw are 0.08 and 0.0, respectively. Subsequently, the vw is regarded as "bulk water fraction," which can also be estimated in the proposed method, in addition to the bulk density, and will be analyzed in the following

section. The impact of ice fraction assumption on bulk density retrieval will be investigated in the discussion section.

Since a direct comparison of the bulk water fraction is unavailable, this study will use two approaches to evaluate the bulk density derived from the proposed method. First, the retrieved bulk density was validated by the reflectivity-weighted fall velocity ($V_Z$) from MRR and Parsivel. The fall velocity measurements from MRR ($V_Z^{MRR}$) and the "density-calculated" fall velocity derived from retrieved bulk density ($V_Z^{\rho bulk}$) are compared to ensure the consistency of observed and calculated $V_Z$.

The measurement, $V_Z^{MRR}$, is not used in the bulk density retrieval procedures. This approach utilizes only MRR and Parsivel measurements.

The terminal fall velocity of particle size $D$, $V(D)$, was computed from derived bulk density ($\rho_{bulk}$) as follows (Rogers and Yau 1989),

$$V(D) = \left(\frac{4}{3}\frac{g}{C_d}\frac{\rho_{bulk}}{\rho_{air}}\right)^{0.5} D^{0.5} \quad (3)$$

The $C_d$ is the drag coefficient and equals 0.5 for the sphere hydrometeor assumption. $g$ is the gravity constant (9.81 kg m$^{-2}$). $\rho_{air}$ is the air density which is assumed as constant ($1.29 \times 10^{-3} g\ cm^{-3}$) in the calculation. Subsequently, the corresponding $V_Z^{\rho bulk}$ can be obtained from $V(D)$ as following,

$$V_Z^{\rho bulk} = \frac{\sum \sigma(\rho_{bulk}, D)\, V(D)\, N(D)\, dD}{\sum \sigma(\rho_{bulk}, D)\, N(D)\, dD} \quad (4)$$

The $\sigma(\rho_{bulk}, D)$ indicates the backscattering cross-section of particle size $D$ and retrieved bulk density ($\rho_{bulk}$). $N(D)$

represents the PSD from Parsivel. The comparison of $V_Z^{MRR}$ and $V_Z^{\rho bulk}$ is considered an overall validation of the retrieved bulk density. In addition, the inconsistency between the $V_Z^{\rho bulk}$ and the $V_Z^{MRR}$ can identify inadequate bulk density retrieval. For example, the attenuation effect can lead to underestimating the MRR reflectivity measurement and, thus, underestimating the retrieved bulk density.

The second approach to evaluating the retrieved bulk density is examining the liquid-equivalent snowfall rate (SR, mm hr⁻¹). The collocated Pluvio was utilized to investigate the performance of the derived bulk density. There is no undercatch adjustment is applied in the calculation of SR from Pluvio. The SR was calculated with the derived bulk density $\rho_{bulk}$, fall velocity $V(D)$ and the PSD from Parsivel measurement as shown in (5).

$$SR = 3.6 \sum_{i=1}^{32} \frac{\pi \rho_{bulk}}{6} D^3 \times V(D_i) N(D_i) \, dD \quad (5)$$

The measured and density-calculated SR are integrated into 5 min resolution (Li et al. 2018) to avoid instant fluctuation in the comparison. The SR calculated from the derived density is compared with the Pluvio SR, independent of the Parsivel and MRR observations. Once the PSD is obtained from Parsivel, the volume-weighted diameter ($D_v$) defined in (6) can be derived (Kim et al. 2021),

$$D_v = \frac{\int_{D_{min}}^{D_{max}} D^4 N(D) dD}{\int_{D_{min}}^{D_{max}} D^3 N(D) dD} \quad (6).$$

The $\rho_{bulk} - D_v$ relation is also derived for analysis.

## 4 Results

The bulk density estimation has been applied to all available data during the ICE-POP 2018 and its pre-campaign, as listed in Table 1. There are 17 events: five sites with collocated MRR and Parsivel, and four of them equipped with a Pluvio. The density-calculated fall velocity ($V_Z^{\rho bulk}$) will be examined with MRR measured fall velocity ($V_Z^{MRR}$). Subsequently, the liquid-equivalent snowfall rate (SR) will be validated by collocated Pluvio measurements. The statistical performance of retrieved bulk density will be investigated by comparing the SR. Furthermore, the detailed analysis of the two selected events that produced the most snowfall accumulation during ICE-POP 2018 will be illustrated by examining their environmental condition, precipitation type, and the spatiotemporal evolution of retrieved bulk density and bulk water fraction.

### 4.1 Reflectivity-weighted (Vz)

The normalized number concentration function of measured $V_Z^{MRR}$ from MRR and "density-calculated" $V_Z^{\rho bulk}$ from Parsivel PSD of five sites are shown in Fig. 3. The $V_Z^{MRR}$ and $V_Z^{\rho bulk}$ are in agreement with each other. The majority of the data show reasonably consistent values. The GWU site had the most consistent velocity. On the other hand, YPO, MHS, and CPO sites had second peak values of about 2.0-3.0 m s⁻¹ of $V_Z^{MRR}$ and 1.0-2.0 m s⁻¹ of $V_Z^{\rho bulk}$. In general, the $V_Z^{\rho bulk}$ values of YPO, MHS, and CPO are slightly lower than $V_Z^{MRR}$. The bias values (Table 3) are about -0.81 ms⁻¹ to 0.01 ms⁻¹. The standard deviation values are about 1.02 to 1.88 ms⁻¹, respectively. All sites combined mean bias and standard deviation values are -0.46 ms⁻¹ and 1.35 ms⁻¹, respectively. It is postulated that the lower values of $V_Z^{\rho bulk}$ than of $V_Z^{MRR}$ is caused by the attenuation effect on the MRR reflectivity. The attenuated reflectivity leads to underestimation of retrieved bulk density and "density-

calculated" $V_Z^{\rho bulk}$. An example of attenuated reflectivity will be discussed in the case study of 28 February 2018. The retrieved bulk density influenced by the attenuation effect will be identified and removed by visual examination of Vz comparison.

The deviation of the calculated $V_Z$ is partly attributed to the idealized equation of terminal fall velocity in (3) and (4). In the

225 $V_Z$ calculation, particles are assumed to be spheres, and the drag coefficient is 0.5. The various shapes aerodynamically complicate the falling behaviors of ice-phase and mixed-phase particles (Mitchell and Heymsfield 2005; Heymsfield and Westbrook 2010). Moreover, various measurement issues of MRR and Parsivel also induce some inconsistency. For example, Battaglia et al. (2010) indicated Parsivel's fall velocity measurement error due to the internally assumed relationship between horizontal and vertical snow particle dimensions. The low SNR of MRR reduces Vz measurement quality. In addition, the

230 sampling volume discrepancy increases the Vz inconsistency. Nevertheless, the overall consistency of the $V_Z^{MRR}$ and $V_Z^{\rho bulk}$ suggests that the retrieved bulk density is an adequately reasonable value.

## 4.2 Liquid-equivalent snowfall rate (SR)

The density-derived 5-minute SR is obtained and compared with collocated Pluvio measurements. Fig. 4 demonstrates the normalized number concentration function of the density-derived and measured SR of four sites. The majority of the SR values

are less than 2.0 mm hr[-1]. The mean 5-minute SR from Pluvio is 1.08 mm hr[-1]. Most retrieved and observed SR are around the 1-to-1 line, indicating that the retrieved bulk density SR agrees with the Pluvio SR measurement. Some fractional data scatter away from the 1-to-1. The bias values of each site are about -0.25 to 0.06 mm hr[-1]. The standard deviation values are about 0.88 to 1.35 mm hr[-1], respectively. The overall mean bias and mean standard deviation values of the five sites are -0.12 ms[-1] and 1.07 mm hr[-1], respectively. The results indicate that the density-derived SR is slightly lower than the Pluvio-observed SR.

It is postulated that the remaining attenuation effect caused by the accumulated snow atop the MRR antenna induced the underestimation of the reflectivity and, thus, the retrieved bulk density and SR. Even though the MRRs deployed in ICE-POP were equipped with heating capability, notes taken during the observation indicate a significant amount of accumulated snow on the antenna. Thus, some inconsistency between MRR reflectivity and Parsivel PSD can be noticed. More discussion will be shown in the next section.

In addition to MRR attenuation, part of the inconsistency can be attributed to Pluvio-observed SR bias caused by the wind-induced undercatch issues (Kochendorfer et al. 2017; 2018; 2022; and Colli et al. 2020). Kochendorfer et al. (2017) and Colli et al. (2020) have proposed wind-speed based undercatch correction algorithms for single/no-shield instruments. In this study, instead of applying the undercatch correction to single/no-shield Pluvio measurements, all sites were equipped with double windshields to mitigate wind-induced undercatch issues during ICE-POP 2017/2018 (see section 2). Kochendorfer et al. (2017;

2018;) indicate that the SDFIR (small DFIR) and the Belfort double-Alter windshield have much smaller uncorrected biases and also smaller adjusted RMSE relative to the corresponding reference. Kochendorfer et al. (2018) show that the collection efficiency (CE) for the Belfort double-Alter windshield (YPO site) is about 0.9 at a wind speed of 4 m s[-1]. On the other hand, the CE of the double-Alter windshield (BKC and GWU sites) is dropped to 0.7 at a wind speed of 4 m s[-1]. The MHS site with

DFIR has fewer undercatch issues. It's postulated that wind-induced undercatch issues partially contribute to the discrepancies between density-derived and measured SR. Further investigation of the wind-induced undercatch issues is needed.

Two snow events, 28 February and 7 March 2018, from the ICE-POP 2018, are selected for further investigation. The synoptic pattern of these two events was characterized as warm-low, according to Kim et al. (2021), with the low pressure situated to the south of the polar jet. Most of the precipitation was in the southern and eastern parts of Korea. The warm and moist air was transported from the Yellow Sea and East Sea. This moist air forms supercooled water as it encounters the steep Taebaek mountain, which benefits the growth of ice-phased particles by riming (Kim et al. 2021). The event on 28 February 2018 had the most intense precipitation rate, and the most accumulated snowfall during the ICE-POP (Gehring et al. 2020) is investigated. The data of the MHS site is examined to understand the pros and cons of the proposed bulk density estimation algorithm. The attenuation effect of MRR reflectivity will be discussed. The derived bulk density of the five sites aligned from the Taebaek mountain to the coast (from southwest to the northeast are YPO, MHS, CPO, BKC, and GWU) from the event of 7 March 2018 are investigated to understand the evolution of the derived bulk density in the mountainous and coastal sites.

### 4.3 Case study: 28 February 2018

The time series of observational data from the MHS site on 28 February 2018 is shown in Fig 5. The mid-level precipitation was evaporated and observed by MRR from 00 to 03 UTC (Fig. 5c). Precipitation reached the surface at 03 UTC and continued until 16 UTC (Fig. 5b), the precipitation gradually weakening after 16 UTC (Gehring et al. 2020). Both retrieved bulk density and bulk water fraction decreased gradually from 0.4 to 0.05 g cm$^{-3}$ and 0.5 to 0.0 between 03 and 05 UTC (Fig. 5a), while the temperature slowly dropped from 5°C to 0°C. The derived bulk density between 04 and 16 UTC was low (less than 0.2 g cm$^{-3}$), and the bulk water fraction was nearly zero. The MASC data show aggregate particles during this period (Gehring et al., 2020). The precipitation gradually weakened after 16 UTC, and the bulk density increased again (0.4 to 1 g cm$^{-3}$). The graupel and small particles were identified as the temperature continued to drop till the end of the day, according to MASC data (Gehring et al., 2020). The higher retrieved bulk density during the weakening period is consistent with the hydrometeor classification of MASC observation in Gehring et al. (2020).

The velocities from MRR ($V_Z^{MRR}$) and calculated from retrieved bulk density ($V_Z^{\rho bulk}$) are shown in Fig. 5e. The increasing values of $V_Z^{MRR}$ from 15 UTC to 18 UTC is consistent with the $V_Z^{\rho bulk}$ calculated from bulk density. However, a pronounced discrepancy between the $V_Z^{\rho bulk}$ and the $V_Z^{MRR}$ can be noticed at 06 UTC and between 09 and 14 UTC (Fig. 5e). The fall velocity inconsistency suggests that the retrieved bulk density is not derived adequately. It can be noticed that there was a pronounced $Z_{HH}$ drop in MRR measurement around 06 UTC and 09 to 13 UTC (Fig. 5d), which is inconsistent with PSD measurement (Fig. 5c). This implies that significant snow accumulation on the antenna, particularly associated with large-sized aggregated snow at 06 UTC (Fig. 5c), likely results in strong attenuation. The attenuated reflectivity measurement leads to the degraded performance of the bulk density retrieval algorithm. The unreasonable retrieval data is identified by $V_Z$ criteria introduced in the previous section and is regarded as less credible (a gray area in Fig. 5).

In Fig. 5f, the consistency of the bulk density calculated SR and observed SR from Pluvio can be found from 03 to 08 UTC and 15 UTC to 24 UTC. The underestimation of the bulk density calculated SR from 08 to 15 UTC was caused by the attenuated MRR measurement. Overall, both the $V_z^{\rho_{bulk}}$ and SR show good agreement with the MRR and Pluvio, except the period with inadequate bulk density due to attenuation effect. The result indicates that the algorithm can derive bulk density adequately.

The fall velocity-diameter relation examines the overall microphysical characteristics of the event in Fig. 6. Distinct characteristics of the fall velocity-diameter relation can be noticed before 04 and after 16 UTC. In Figs. 6a and c, most of the hydrometeor size was less than 2 mm (Fig. 5b), and fall velocity was much higher than 1 m s$^{-1}$ (Fig. 5e, close to the fall velocity-diameter relation of rain and graupel) before 04 UTC and after 16 UTC.  Before 04 UTC, the fog near the surface and nimbostratus was observed by W-band radar (Gehring et al. 2020). In addition, the temperature was above 0°C, and the particle

size was less than 5 mm. These features suggest possible wet snow, small raindrops, and drizzle. After 16 UTC, the graupel and small particles with near zero environment temperature were identified by MASC (Gehring et al. 2020). The particles were aggregate-like with a lower derived density from 04 to 16 UTC (Fig. 6 b). The maximum particle size ranged from 8 to 20 mm (Fig. 5b and Fig. 6b). More particles were found between the lines of graupel and dry dendrites fall velocity-diameter relations.

## 4.4 Case study: 7 March 2018

Both the 7 March and 28 February events share similar larger-scale conditions. The difference between these two events is that the precipitation on the 7 March was weaker yet persisted for a longer time than the 28 February event. An east-moving trough from east China became a potential vorticity streamer (Gehring et al. 2020). The low-pressure system developed over the Korean peninsula and produced intensive precipitation. Besides the similarity in larger-scale conditions, the microphysical characteristics of precipitation systems share similar behaviors in these two cases. The intensive precipitation associated with

the low-pressure system started at 10 UTC on 7 March. As the nimbostratus weakened and dissipated into the shallow convection at about 03 UTC on 08 March 2018, the high bulk density can be found in most sites (Fig. 8a-11a). The weakened precipitation with shallow convections can be seen from 08 UTC to 19 UTC on 8 March (reflectivity profiles in Fig. 7c-11c).

The overall bulk density and bulk water fraction in the GWU site (Fig. 11) are the highest in all sites. In contrast, the YPO (Fig. 7) site demonstrates a lower magnitude of bulk density and bulk water fraction, especially after 04 UTC on 8 March. The

contrast may be attributed to their environmental condition, as the YPO site is located in the westernmost mountainous area of the five sites. In contrast, the GWU site is located on the east coast, which faces abundant moisture from the East Sea (Kim et al. 2021). Gradual increase of density, as well as the bulk water fraction, can also be found consistent with an increase in the distribution of fall velocity versus the diameter from MHS, BKC, to the GWU sites (Fig. 12). The GWU site features mostly the particles concentrated around the fall velocity-diameter relation of rain. Meanwhile, in MHS and BKC sites, especially the

MHS site, data are distributed more discretely and scattered between the relation of rain and the relation of dry dendrites. Overall, the retrieved bulk density and bulk water fraction qualitatively reveal distinct fall velocity-diameter relations of each site due to the different mesoclimate environments.

The YPO, MHS, and CPO sites had continual low bulk density and bulk water fraction values (about 0.1 to 0.2 g cm$^{-3}$, Figs. 7a-9a) at the beginning of the precipitation. The MRR reflectivity profiles indicate an intensive precipitation system up to 5 km (Figs. 7c-9c) from 10 UTC on 7 March to 04 UTC on 8 March. The precipitation gradually dissipated at 04 UTC on 8 March. The PSD was featured with large particle size (Figs. 7b-9b) and aggregates-like particles (Gehring et al. 2020). On the other hand, the coastal sites (BKC and GWU) began with nimbostratus cloud and high values of bulk density and bulk water fraction, 10 to 13 UTC for BKC and 10 to 19 UTC for GWU on 7 March (Figs. 10a,b and 11a,b). The high bulk density (about 0.9 g cm$^{-3}$) period corresponds to a higher bulk water fraction from 0.4 to 0.8. The precipitation gradually transitioned to a low bulk density at other sites (YPO, MHS, and CPO) till 04 UTC on 8 March. The fall velocity-diameter relation is consistent with the high bulk water fraction. The Parsivel data reveals the fall velocity-diameter relation of rain from 08 to 19 UTC on 7 March (Fig. 12c), and the graupel relation from 19 UTC to 03 UTC on 8 March in the GWU site (Fig. 12f). The decrease of averaged fall velocity between 08 to 19 UTC on 7 March (Figs. 12a, b) and 19 UTC to 03 UTC on 8 March (Figs. 12d, e) is consistent with the decreasing density in MHS and BKC sites. It is postulated that the consistent changes in bulk density and water fraction with velocity-diameter relation are associated with melting or rimming particles.

After 04 UTC on 8 March, the BKC and GWU sites featured high bulk density and bulk water fraction (Figs. 10a-11a). The PSDs were mainly small particles (Figs. 10b-11b). According to MRR measurements, the precipitation systems were weaker and shallower (Figs. 10c-11c) compared to the period with low bulk density before 04 UTC on 8 March. The Vz of particles also transitioned from consistently low values to slightly higher and more noisy values (Figs. 10e-11e), suggesting high-density particles with high fall velocity. Compared to 19 UTC on 7 March to 03 UTC on 8 March (Figs. 12d, e, and f), the fall velocity-diameter relation from 04 to 12 UTC on 8 March is more consistent with the relation of rain (Figs. 12g, h, and i). After 12 UTC, more particles were distributed between the relation of rain and graupel (Fig. 12j, k, and l).

Moreover, the good agreements between the Pluvio SR and the derived SR calculated from all sites' bulk density can be noticed in Figs. 7f-8f, and 10f-11f. The Pluvio is not available at the CPO site. Two distinct types of precipitation structures (according to MRR) and microphysical characteristics (bulk density, bulk water fraction, PSD) can be noticed in Figs. 7-11. One has deeper and more intensive precipitation structures, higher bulk density, and bulk water fraction, and it contains smaller particles. Overall, the estimation of the bulk density and the bulk water fraction demonstrates the contrast between sites of geographical locations and captures the evolution of the precipitation system. The statistical analysis of retrieved properties of mountain and coastal sites will be discussed in the following section.

**4.5 Statistical analysis of bulk density and bulk water fraction**

The retrieved bulk density and bulk water fraction are investigated statistically to understand the microphysical characteristics of the winter precipitation systems from ICE-POP 2018 and its pre-campaign. Fig. 13 shows the number concentration of retrieved bulk density and observed volume-weighted diameter ($D_v$) from PSD of all sites. The bulk density decreases exponentially as $D_v$ increases. Heymsfield et al. (2004) utilized the aircraft data collected from two field programs, namely the Atmospheric Radiation Measurement (ARM) program, Cirrus Regional Study of Tropical Anvils and Cirrus Layers

(CRYSTAL) Florida Area Cirrus Experiment (FACE) in southern Florida during July 2002. The ARM data is mostly ice clouds formed primarily through large-scale ascent, and the CRYSTAL observations are mainly from convectively generated cirrus anvils. Brandes et al. (2007) utilized the data of 52 storm days from the Front Range in eastern Colorado during October–April 2003 to 2005 of a ground-based 2DVD. The data of Brandes et al. (2007) is dominated by almost spherical aggregates having near-exponential or superexponential size distributions. Early studies, namely Magono and Nakamura (1965), Holroyd (1971), Muramoto et al. (1995), and Fabry and Szyrmer (1999), have documented various density-particle size relationships (Table 4 and Fig. 13). The particle diameter definitions vary in each study (Table 4). Instead of converting various particle diameter definitions, the particle diameter remains as proposed in each study. Despite distinct environmental conditions, instrumentations, and retrieval techniques, most of the particles in this study are consistent with the density-particle size relationship from previous studies. These results indicate that the proposed bulk density estimation algorithm can derive reasonable retrievals with statistically consistent microphysical characteristics of earlier studies.

To further understand the microphysical characteristics of winter precipitation, each site's retrieved bulk density and bulk water fractions are divided into warm-low (nine cases) and cold-low (five cases) events according to the synoptic condition (Gehring et al. 2020; Kim et al. 2021). As shown in Fig. 14a, the median values of bulk density of warm-low events from the mountain site (YPO) to the coastal site (GWU) are about 0.10 to 0.29 g cm$^{-3}$. The GWU site has the highest bulk density. On the other hand, the median values of bulk density of cold-low events from YPO to GWU are about 0.07 to 0.05 g cm$^{-3}$ (Fig. 14b). The overall bulk density values are lower in cold-low events than in warm-low events.

In Fig. 14c and d, more than 90% of bulk water fractions are less than 0.03 for warm- and cold-low events. The YPO site has the lowest bulk water fraction, especially the cold-low events that remain lower than 0.22. The mean value of the top 5% of the bulk water fraction of each site is obtained for further investigation. The values of bulk water fraction gradually increase from the mountain site (YPO) to the coastal site (GWU) for both warm- and cold-low events. The mean values of the top 5% bulk water fraction of YPO, MHS, and CPO sites for warm-low events are 0.0015, and the BKC and GWU are about 0.32 to 0.45. The cold-low events are 0.0013 to 0.19 for each site.

The temperature ($^{0}$C) and water vapor pressure (hPa) measurements from nearby mountain and coastal AWS sites are collected and summarized in Fig. 14e. Warm-low events have warmer and moister conditions than cold-low events. The coastal area's warm- and cold-low events have similar mean temperature values. On the other hand, the water vapor pressure increases significantly from cold-low to warm-low events in the coastal region. The mountain area has similar features, but higher temperature increments and fewer increments of water vapor pressure. These results indicate that the winter precipitation systems of coastal sites with warmer and moister environments have higher bulk density and bulk water fraction than mountain sites.

## 5 Discussion

The proposed retrieval algorithm has shown that it can estimate the bulk density and bulk water fraction with qualitatively reasonable performance. The underestimation of retrievals caused by the attenuation effect of MRR reflectivity has been well

identified by reflectivity-weighted velocity. In addition to the attenuation effect, the uncertainties of the retrieval algorithm can be attributed to observational data quality and the algorithm's basic assumption. To understand the uncertainty in the proposed algorithm's bulk density retrieval, the retrieved bulk density is compared with the retrieval from the collocated precipitation imaging package (PIP), a video disdrometer, at the HMS site during ICE-POP. Furthermore, the credibility of retrieved bulk water fractions will also be investigated. The following discussion will explore the maximum ice fraction assumption and Parsivel's PSD measurement uncertainty. In addition, the impact of bulk density and water fraction retrieval from spherical particle assumption and the uncertainty of MRR reflectivity measurement will be discussed.

## 5.1 The bulk density comparison with collocated PIP

The PIP provides the PSD, fall speed, density, and snowfall rate of hydrometers (Newman et al., 2009; Pettersen et al., 2020) was also deployed at the MHS site during ICE-POP 2018. Tokay et al. (2023) have utilized PIP to investigate the PSD parameters, including mass-weighted diameter and normalized intercept. The bulk density is estimated by Tokay et al. (2023) with various assumptions. The PIP retrieved density was generated from the assumption that $D_{max} = 1.15 \, D_{eq}$, and the mass derivation included was based on Bohm (1989). As shown in Fig. 15, the retrieved bulk density from the proposed algorithm in this study (blue dots) and PIP (gray dots) have high consistency. Both retrieved bulk densities are highly correlated to each other. As shown in Fig. 15a, the retrieved bulk density values from the proposed algorithm and PIP gradually decrease from nearly 1.0 to 0.1 (g cm$^{-3}$) between 03 and 06 UTC. Both algorithms capture the fast transition from the mixed-phase to dry snow. Except for the period of 08 to 15UTC on 28 February (Fig. 15a), due to the attenuation effect of the accumulated snow on the MRR antenna (Fig. 5e), the retrieved bulk density is much lower than PIP. The bulk density retrieval from 06 to 12 UTC on 8 March is not available (Fig. 15b) due to missing MRR data (Fig. 9c). It is postulated

## 5.2 The sensitivity of maximum ice fraction assumption to the bulk density retrieval

The bulk snow density is determined from possible combinations of vi/vw, and the "maximum bulk density" with maximum ice fraction (vi) is selected in the proposed algorithm. This maximum ice fraction assumption has been applied to the entire ICE-POP data. As shown in Fig. 2b, choosing the bulk density with minimum ice fraction (e.g., vi=0) leads to the maximum value of vw and minimum bulk density. However, the particle is unlikely to be composed only of water and air. A sensitivity study of selecting a different water/air/ice combination is conducted. The half-maximum ice fraction is selected to derive the retrieved bulk density. As shown in Fig. 15, the retrieved bulk density from the half-maximum ice fraction (red dots) has systematically lower values than the maximum ice fraction (blue dots). The bulk density retrievals from the half-maximum ice fraction have significant discrepancies compared to PIP retrievals.

On the other hand, the density retrieval from the maximum ice fraction assumption has good agreements with PIP retrievals (gray dots). The consistency of retrieved bulk density from the maximum ice fraction to PIP provides more confidence in the assumption of maximum ice fraction. However, the maximum ice fraction assumption may not be valid in a mixed-phased

condition when the ice particles melt at a nearly freezing temperature environment. Further investigation is needed in the future study.

### 5.3 The measurement uncertainty of Parsivel fall velocity and its impact on the bulk density and water fraction retrieval

As indicated by the study by Battaglia et al. (2010) and Wood et al. (2013), Parsivel and 2DVD have various issues in snowflake particle measurement. This issue is due to the internally assumed relationship between horizontal and vertical snow

particle dimensions. Yuter et al. (2006), Aikins et al. (2016), and Kim et al. (2021) indicate the splashing and border effects of the diameter of < 1 mm in Parsivel fall velocity measurements. The fall velocity issue explains why the consistently high fall velocity of diameter less than 1 mm in the MHS site can be noticed by the fall velocity-diameter relation in Figs. 6 and 12; even the retrieved bulk density and water fraction were low and should be associated with low fall velocity. Yuter et al. (2006) and Aikins et al. (2016) suggest a quality control procedure that discards particles with a diameter of < 1 mm to avoid splashing

and border effects.

As Battaglia et al. (2010) indicated, the Parsivel overestimates the snowfall velocity and underestimates the PSD. A correction factor (CF) derived from comparing the collocated 2DVD in the MHS site is suggested by Dr. Gyuwon Lee (personal communication, summarized in Table 5). The particle size-dependent CF adjusts the fall velocity measurement from Parsivel and thus modifies the PSD. The CF reduces the fall velocity to a factor of two as the particle size is around 10 mm.

The modified PSD has a higher concentration after applying CF adjustment. The bulk density and water fraction retrieval uncertainty due to PSD measurement issues are investigated using the PSD and CF-adjusted PSD. As shown in Fig. 15, the bulk density retrieval (black dots) decreases slightly after applying CF-adjusted PSD in the 28 February and the 7 March 2018 events. The bias of retrieved bulk density is about -0.0153 (g cm$^{-3}$), and the standard deviation is about 0.095 (g cm$^{-3}$). The bias and the standard deviation for bulk water fraction -1.8x10$^{-4}$ and 2.6x10$^{-3}$.

The results indicate that with the PSD measurement uncertainty from Parsivel, the bulk density and water fraction retrieval are fairly low. It is postulated that fall velocity measurement uncertainty slightly impacts the PSD calculation; thus, the bulk density and water fraction retrieval uncertainty is sufficiently low.

### 5.4 The measurement uncertainty of MRR reflectivity and its impact on the bulk density and water fraction retrieval

The simulation of MRR reflectivity can be sensitive to the particle shape assumption. A sensitivity investigation assuming

the particle axis ratio of 0.5 and the mean and standard deviation of the canting angle are 0° and 20°, shows that about 1.5 dBZ variation of MRR reflectivity can be induced. Another possible source of retrieval uncertainty is the measurement of MRR reflectivity. As discussed in the MRR bias calculation from pure rain events, the standard deviation between MRR reflectivity and Parsivel calculated reflectivity is about 1.1 to 1.3 dBZ for each site. A random error of MRR reflectivity with a standard deviation of 1.2 dB is introduced into the retrieval algorithm to imitate the particle assumption and MRR measurement

uncertainty. In Figure 16(a), the algorithm overestimates the bulk density ($\Delta\rho^{bulk} > 0$) as the MRR reflectivity's positive error ($\Delta Z_{HH} > 0$) increases. On the other hand, the negative bias of MRR reflectivity ($\Delta Z_{HH} < 0$) led to an underestimation of the

bulk density retrieval ($\Delta\rho^{bulk} < 0$). The overall standard deviation of bulk density retrieval uncertainty is about 0.025 (g cm$^{-3}$) for a given MRR reflectivity uncertainty of 1.2 dB. The bulk water fraction retrieval has the same feature shown in Fig. 16(b). Given an MRR reflectivity uncertainty of 1.2 dB, the bulk water fraction retrieval uncertainty is about 0.041.

Despite various potential factors that could compromise bulk density retrieval from collocated MRR and Parsivel instruments, the uncertainty study indicates that observational errors have a reasonably low effect, maintaining acceptable performance. The agreement between this study and PIP retrieval further confirms that the proposed algorithm can robustly retrieve the bulk density and water fraction from collocated MRR and Parsivel.

**5.5 The retrieval uncertainty of bulk water fraction**

The performance of retrieved bulk density has been quantitatively validated by comparing collocated Pluvio-derived SR and PIP-derived bulk density. On the other hand, quantitative validation of retrieved bulk water fraction is not available due to the limitation of instrumentation. No instrument is capable of directly measuring the bulk water fraction. This study's retrieved bulk water fraction is considered qualitatively reasonable according to the case studies of the 28 February and 7 March 2018 events and the statistical analysis of warm-/cold-low events over coastal and mountain sites (section 4.3–4.5). The

distinct bulk density and bulk water fraction retrievals of coastal and mountain sites are revealed. The results indicate that the winter precipitation systems of coastal sites with warmer and moister environments have higher bulk density and bulk water fraction than mountain sites.

The composition of water/ice/air fraction determines the bulk density. The retrieved bulk water fraction will differ if a different assumption is made when selecting possible bulk density. Therefore, the performance of the retrieved bulk water

fraction is partially linked with bulk density retrieval. As shown in Fig. 15a, both the proposed algorithm and PIP capture the fast transition from the mixed-phase ($\rho_{bulk} \approx 1.0$ g cm$^{-3}$) to dry snow ($\rho_{bulk} \approx 0.1$ g cm$^{-3}$). Given the absence of direct measurements of bulk water fraction, the consistency between the retrieved bulk density from the two algorithms is indirect evidence of the qualitative reasonableness of the retrieved bulk water fraction. Combining multiple sophisticated instruments (e.g., 2DVD, PIP, SVI, MASC) and developing a more comprehensive technique can improve our understanding of the critical

microphysical characteristics of particles. Further investigation of the particle composition ratio of air/ice/water fraction in different environments is needed.

**6 Conclusion**

The snow density, one of the key characteristics, varies with the microphysical processes under different weather conditions. The variations inherently involve complex behaviors and require more investigation. In the study, the snow density is derived

by compositing collocated MRR and Parsivel data, which can be acquired easily. In the proposed method, the PSD from the Parsivel is applied to T-matrix backscattering simulation and compared with the $Z_{HH}$ from MRR. The bulk density and bulk water fraction are derived from comparing simulated and calculated $Z_{HH}$.

The reflectivity-weighted fall velocity ($V_Z$) of MRR is applied to evaluate the retrieved bulk density and water fraction. Inconsistency of measured Vz from MRR and calculated Vz from retrieved bulk density from 09 to 15 UTC on 28 February 2018 is noticed. It is postulated that the attenuation effect mainly causes the Vz discrepancy due to the accumulated snow on the MRR antenna. The performance of the retrieved bulk density is validated by comparing independent measurements of snowfall rate (SR) from collocated Pluvio and calculated SR from bulk density. General consistency between the measured and the bulk density-calculated SR was found in all available cases (Table 1) of the four sites during the ICE-POP 2018 campaign, as summarized in Table 3.

The bulk density and bulk water fraction of two events with warm-low synoptic patterns (28 February and 7 March 2018) were investigated. Both events show good agreement with the SR calculated from retrieved bulk density and measured from Pulivo. In addition, the retrieved bulk density also shows consistent results with PIP retrieval. Both events can separate lower and higher-density periods with distinct fall velocity-diameter relations. During the transition, particles' bulk density and fall velocity rose with decreasing particle size while the convection precipitation dissipated.

The dissimilar bulk density and bulk water fraction between mountain sites (YPO and MHS, lower bulk density and bulk water fraction) and coastal sites (BKC and GWU, higher bulk density and bulk water fraction) indicates the geographical and mesoclimate environmental effects on distinct microphysical characteristics of winter precipitation systems of each site. Overall, the derivations demonstrated good accordance with the fall speed, the diameter of the particle, and the $V_Z$ and SR in time series, providing an insightful perspective in microphysics analysis.

The SR and Vz validation analysis shows that the algorithm can retrieve the bulk density adequately. The consistency of the retrieved bulk density to collocated PIP at MHS site suggests that the proposed algorithm performs decently in this study. The MHS site had a double windshield with inner Tretyakov and outer Alter shields. The instruments at the MHS were within the DFIR in addition to the double shields to mitigate the wind-induced undercatch issues. The advantage of the proposed algorithm is that it utilizes collocated Parsivel and MRR, which are commercially available, commonly used, and robust instruments. The Parsivel and MRR can operate unattended and need little maintenance. The proposed algorithm provides an alternative choice if a sophisticated instrument (e.g., 2DVD, PIP, SVI, MASC, etc.) is unavailable. Moreover, the proposed algorithm in this study provides a possible approach to estimating the bulk water fraction. Further application of the proposed algorithm helps derive long-term observation data on snow properties.

**Competing interests**

The contact author has declared that none of the authors has any competing interests.

**Acknowledgments**

The authors greatly appreciate the participants in the World Weather Research Programme research development project and forecast demonstration project "International Collaborative Experiments for Pyeongchang 2018 Olympic and Paralympic winter games" (ICE-POP 2018) hosted by the Korea Meteorological Administration (KMA). We would like to thank Sang-Won Joo, Yong-Hee Lee, Kwang-Deuk Ahn, Namwon Kim, and Seung-bo Choi at the KMA for their support for the ICE-POP 2018 field campaign. The authors are grateful to Walter Petersen, Ali Tokay, Patrick Gatlin, and Matthew Wingo at NASA for providing the MRR and PARSIVEL instruments and processing the PARSIVEL data. The PIP data is provided by Dr. Ali Tokay. We would also like to thank Byung-Gon Kim at Gangneung-Wonju National University and Byung-Chul Choi at the High Impact Weather Research Center of the KMA for sharing their instruments. This work was funded by the National Science and Technology Council of Taiwan under Grant 112 WFA0710420 and by the Korea Meteorological Administration Research and Development Program under Grant RS-2023-00237740.

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

# Figures and Tables

Table 1: Data availability of MRR (x), Parsivel (v), and Pluvio (o) for YPO, MHS, CPO, BKC, and GWU sites during pre-campaign and ICE-POP campaign.

| Site \ Dates | YPO | MHS | BKC | GWU | CPO |
|---|---|---|---|---|---|
| 2017/1/4~2017/1/5 | v, x | v, x | | | v, x |
| 2017/1/8 | v, x | v, x | | | v, x |
| 2017/1/29~2017/1/30 | v, x | v, x | | | v, x |
| 2017/2/21~2017/2/22 | v, x | v, x | | | v, x |
| 2017/3/1~2017/3/2 | v, x | v, x | | | v, x |
| 2017/3/14 | v, x | v, x | | | v, x |
| 2017/12/9~2017/12/10 | v, x, o | v, x, o | v, x, o | v, x, o | v, x |
| 2017/12/24 | v, x, o | v, x, o | v, x, o | v, x, o | v, x |
| 2018/1/7~2018/1/8 | v, x, o | v, x, o | v, x, o | v, x, o | v, x |
| 2018/1/16 | v, x, o | v, x, o | v, x, o | v, x, o | v, x |
| 2018/1/22 | v, x, o | v, x, o | v, x, o | v, x, o | v, x |
| 2018/1/30 | v, x, o | x, o | v, x, o | v, x, o | v, x |
| 2018/2/28~2018/3/1 | v, x, o | v, x, o | v, x | v, x, o | v, x |
| 2018/3/4~2018/3/5 | v, x, o | v, x, o | v, x, o | v, x, o | v, x |
| 2018/3/7~2018/3/8 | v, x, o | v, x, o | v, x, o | v, x, o | v, x |
| 2018/3/14~2018/3/15 | v, x, o | v, x, o | v, x, o | v, x, o | v, x |
| 2018/3/20~2018/3/21 | v, x, o | v, x, o | v, x, o | v, x, o | v, x |

Table 2: MRR bias of YPO, MHS, CPO, BKC, and GWU sites derived from rain events.

| site \ MRR bias | YPO | MHS | CPO | BKC | GWU |
|---|---|---|---|---|---|
| Bias (dBZ) | -8.7 | -10.2 | -7.4 | -2.1 | -5.8 |
| Standard deviation (dBZ) | 1.2 | 1.15 | 1.8 | 1.28 | 1.16 |

**Table 3: The $V_Z$ bias is the average of the VZ difference between the MRR measurement and the derived density calculation during all events available (Table 1), while the standard deviation of the bias is the standard deviation of the $V_Z$ difference. Regarding SR (mm hr$^{-1}$), the bias, correlation coefficient, and standard deviation are derived by comparing collocated Pluvio measurements and bulk density-derived SR. The mean values of Pluvio's SR from available comparison data for each site are also shown. There is no available Pluvio on the CPO site.**

|  |  | YPO | MHS | CPO | BKC | GWU | ALL |
|---|---|---|---|---|---|---|---|
| $V_Z$ | Bias (m s$^{-1}$) | -0.81 | -0.69 | -0.46 | -0.39 | 0.01 | -0.46 |
|  | Standard deviation (m s$^{-1}$) | 0.97 | 1.05 | 1.02 | 1.59 | 1.88 | 1.35 |
| SR | Bias (mm hr$^{-1}$) | -0.09 | 0.06 |  | -0.25 | -0.18 | -0.12 |
|  | Standard deviation (mm hr$^{-1}$) | 1.23 | 1.35 | nan | 0.88 | 0.71 | 1.07 |
|  | Mean of Pluvio | 1.03 | 1.30 |  | 1.13 | 0.89 | 1.08 |

**Table 4: Snowflake density–particle size relation comparison.**

| Study | Relation | Particle diameter definition |
|---|---|---|
| Magono and Nakamura (1965) | $\rho_s = 2D^{-2}$ | Geometric mean of the particle major and minor axes |
| Holroyd (1971) | $\rho_s(D) = 0.17D^{-1}$ |  |
| Muramoto et al. (1995) | $\rho_s(D) = 0.048D^{-0.406}$ | Maximum horizontal dimension |
| Fabry and Szyrmer (1999) | $\rho_s(D) = 0.15D^{-1}$ | Equivalent volume diameter |
| Heymsfield et al. (2004) | $\rho_s(D) = 0.104D^{-0.95}$ | Minimum circumscribed circle that encloses the projected area of the particle |
| Brandes et a. (2007) | $\rho_s(D) = 0.178D_0^{-0.922}$ | Median-Volume Diameter ($D_0$) |

**Table 5: The correction factor (CF) for Parsivel fall velocity measurement derived from comparing the collocated 2DVD in the MHS site during ICE-POP. The correction is performed as, $V_{corrected} = V_{observation}$ x CF. Provided by Dr. Gyuwon Lee (personal communication).**

| Particle size (mm) | Correction Factor | Particle size (mm) | Correction Factor |
|---|---|---|---|
| 0.062 | 1.0 | 3.250 | 0.675084 |
| 0.187 | 1.0 | 3.750 | 0.652072 |
| 0.312 | 1.0 | 4.250 | 0.625959 |
| 0.437 | 1.0 | 4.750 | 0.618901 |
| 0.562 | 1.0 | 5.500 | 0.614201 |
| 0.687 | 1.0 | 6.500 | 0.604776 |
| 0.812 | 1.0 | 7.500 | 0.597859 |
| 0.937 | 1.0 | 8.500 | 0.622393 |
| 1.062 | 1.058220 | 9.500 | 0.598366 |
| 1.187 | 0.979561 | 11.000 | 0.563642 |
| 1.375 | 0.934854 | 13.000 | 0.563326 |
| 1.625 | 0.868431 | 15.000 | 0.655855 |
| 1.875 | 0.822200 | 17.000 | 0.556859 |
| 2.125 | 0.780777 | 19.000 | 0.683837 |
| 2.375 | 0.754956 | 21.500 | 0.600309 |
| 2.750 | 0.707859 | 24.500 | 0.679722 |



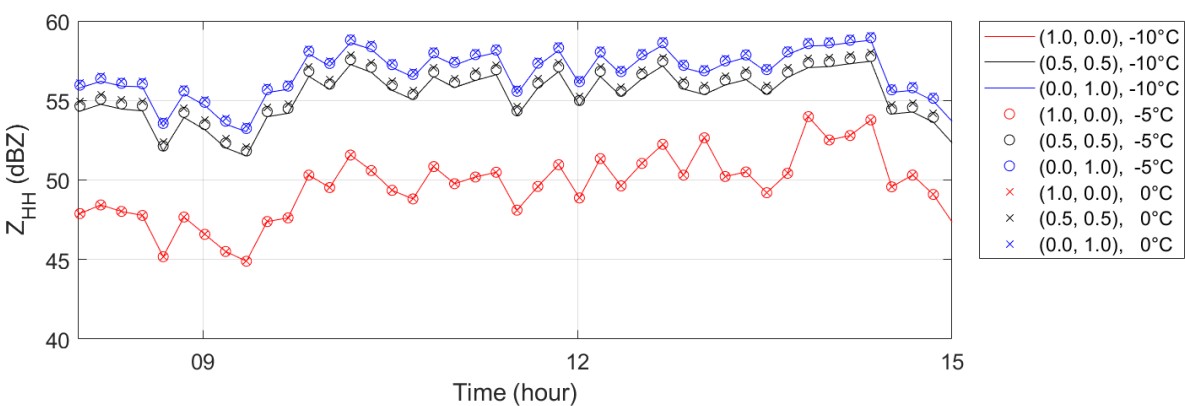


**Figure 1: The simulated $Z_{HH}$ with combinations of the volume ratio of solid ice and liquid water (vi, vw) and temperature is demonstrated by a case from the Parsivel observation of the MHS site on 28 February 2018. The combinations of (vi, vw) were (1.0, 0.0) in red, (0.5, 0.5) in black, and (0.0, 1.0) in blue. The temperature values were -10, -5, and 0°C (thick line, circle, and cross marks).**





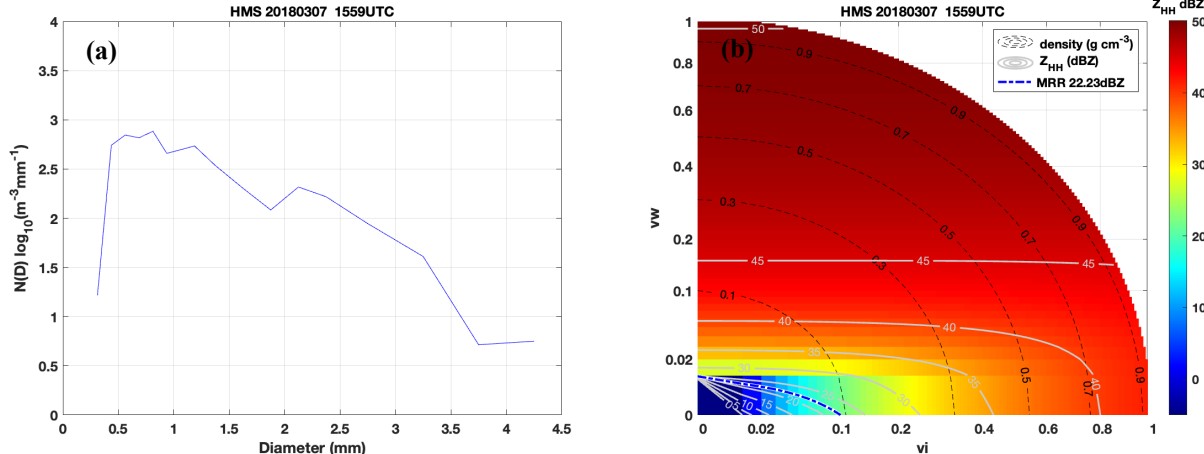

Figure 2: (a) Observed snow size distribution, N(D), from the MHS Parsivel site at 1559 UTC on 7 March 2018. (b) $Z_{HH}$ distribution simulated from the Parsivel PSD. The x- and y-axis are the volume ratio of solid ice (vi) and liquid water (vw), respectively. The shaded color indicates the $Z_{HH}$ magnitude (dBZ), and the black dashed lines are contours of density (g cm$^{-3}$). Grey solid lines are contours of $Z_{HH}$ (dBZ). The blue dashed-dot line is the value of the collocated MRR measurement of $Z_{HH}$ (dBZ).

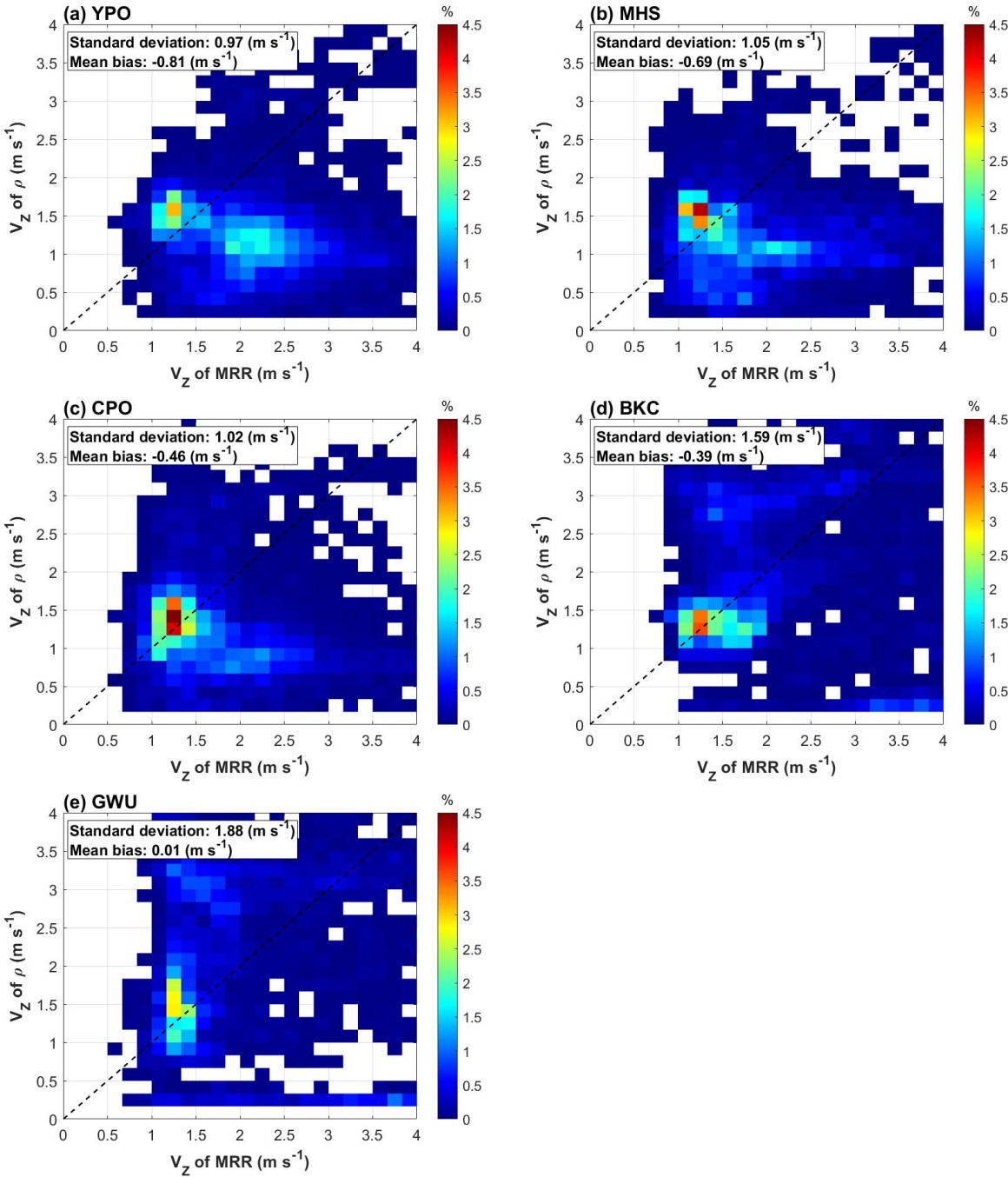

Figure 3: The probability density function of $Z_{HH}$ weighted fall velocity ($V_Z^{\rho bulk}$) calculated from "bulk density" and the MRR measurement ($V_Z^{MRR}$) in (a) YPO, (b) MHS, (c) CPO, (d) BKC, and (e) GWU site.

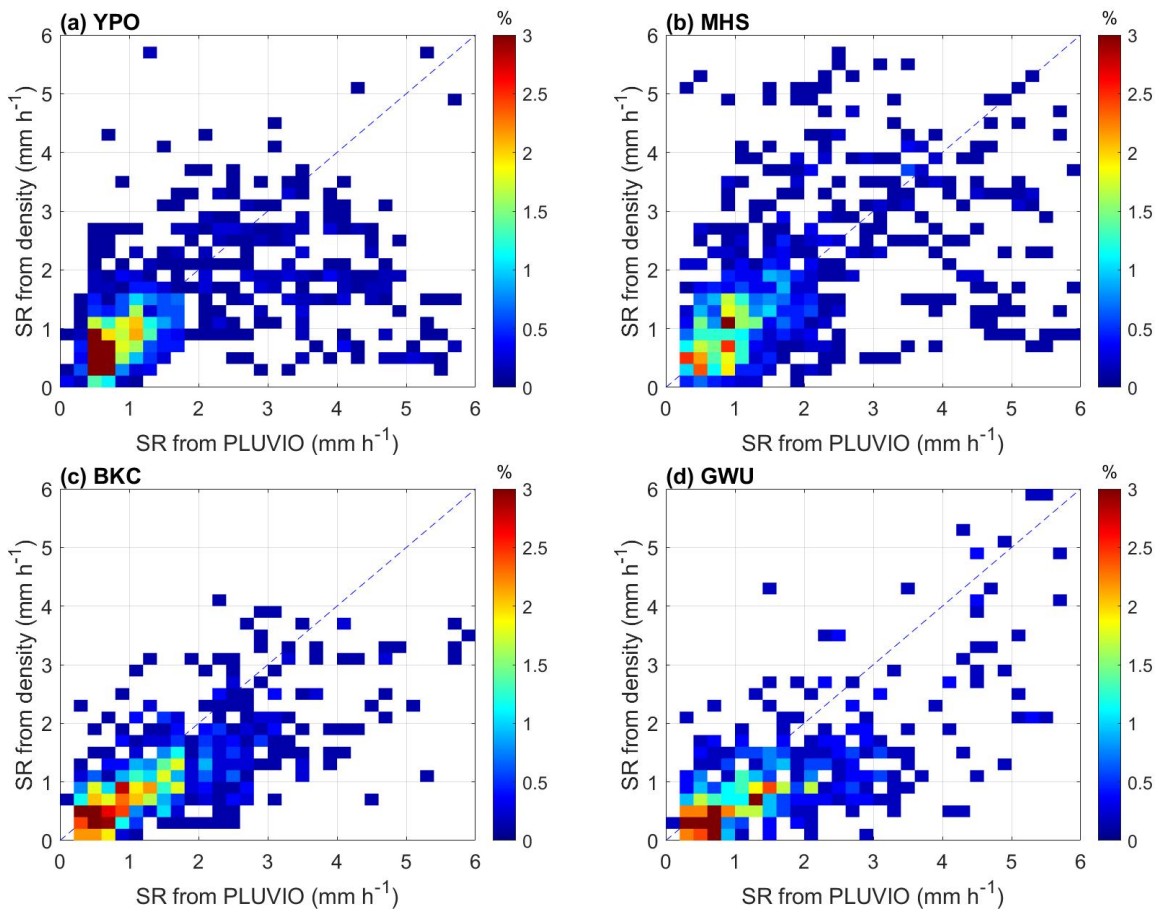

**Figure 4: Normalized number concentration function of SR from density-derived and Pluvio observed SR in (a) YPO, (b) MHS, (c) BKC, and (d) GWU site. The data with attenuated MRR reflectivity has been identified by $V_z$ and removed.**

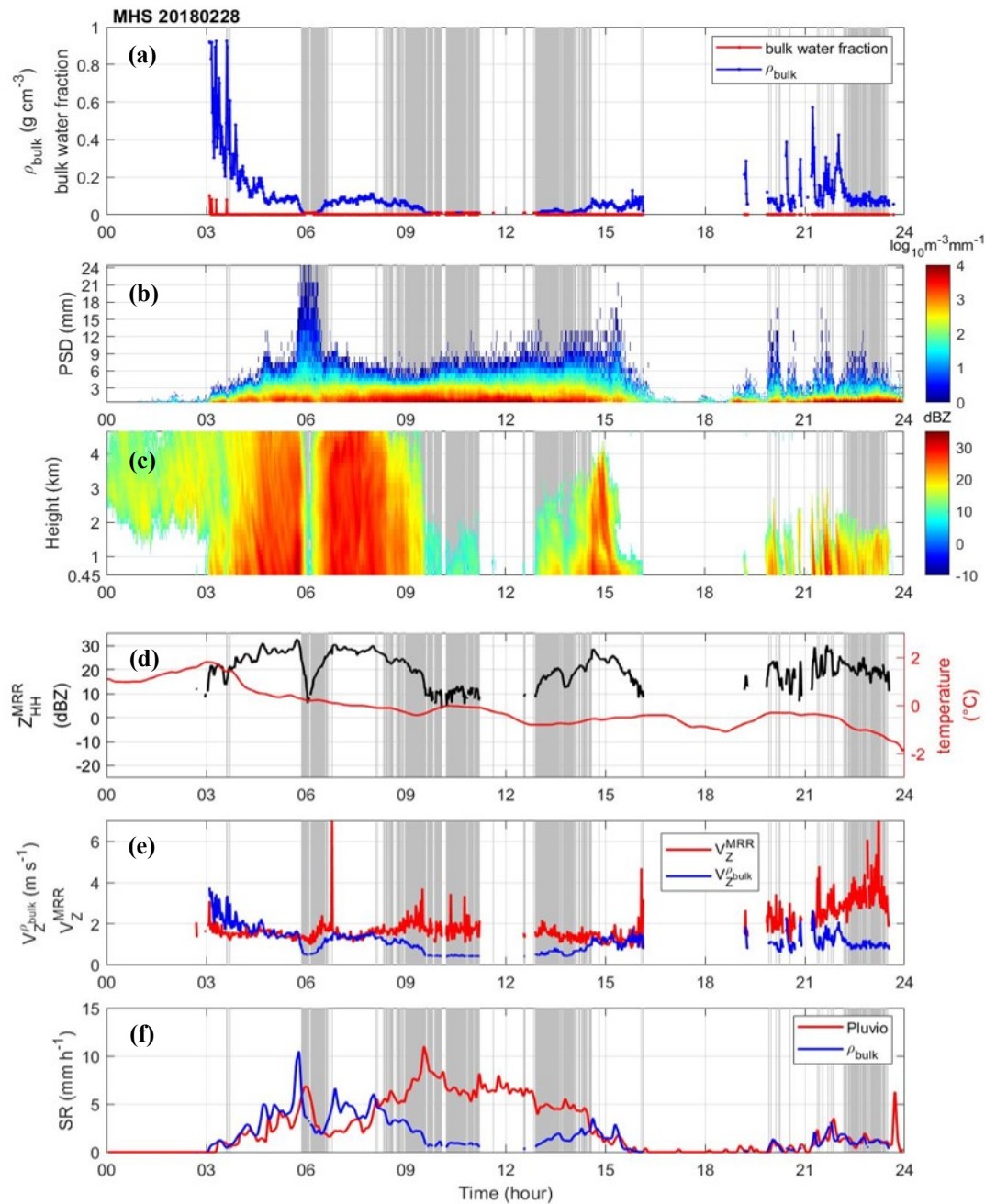

**Figure 5: The data from the MHS site on 28 February 2018. (a) Derived bulk density (g cm⁻³, blue line), and bulk water fraction (red line). (b) Parsivel PSD in logarithm scale. (c) The time series of MRR $Z_{HH}$ vertical profile (dBZ) from the third gate (0.45 km) to the 4.65 km. (d) The temperature (°C, red line) from nearby AWS (Vaisala WXT520) and $Z_{HH}$ from the third layer of MRR. (e) The**
785 **velocity ($V_Z^{MRR}$) of MRR measurement (red line) and the $Z_{HH}$ weighted velocity ($V_Z^{\rho_{bulk}}$) calculated from bulk density (blue line). (f) The liquid-equivalent snow-rate (SR) of Pluvio measurement (red line) and bulk density derived SR (blue line).**

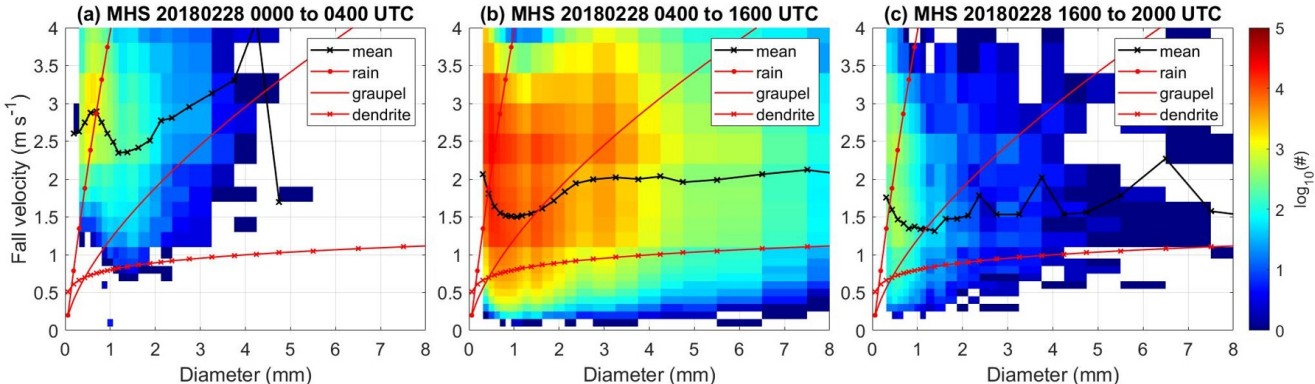

**Figure 6: The number concentration function of fall velocity and diameter from Parsivel (logarithm scale shown in color-shaded) collected from 28 February 2018. The average of fall velocity in each diameter bin (m s⁻¹, black line), relations of fall velocity-diameter of rain, graupel, and dry dendrites (from the upper to lower red line) are shown for (a) 00 to 04 UTC, (b) 04 to 16 UTC, and (c) 16 to 20 UTC in the MHS site. The relation of rain is from Brandes et al. (2002). The relationships of graupel and dry dendrites are from Locatelli and Hobbs (1974).**

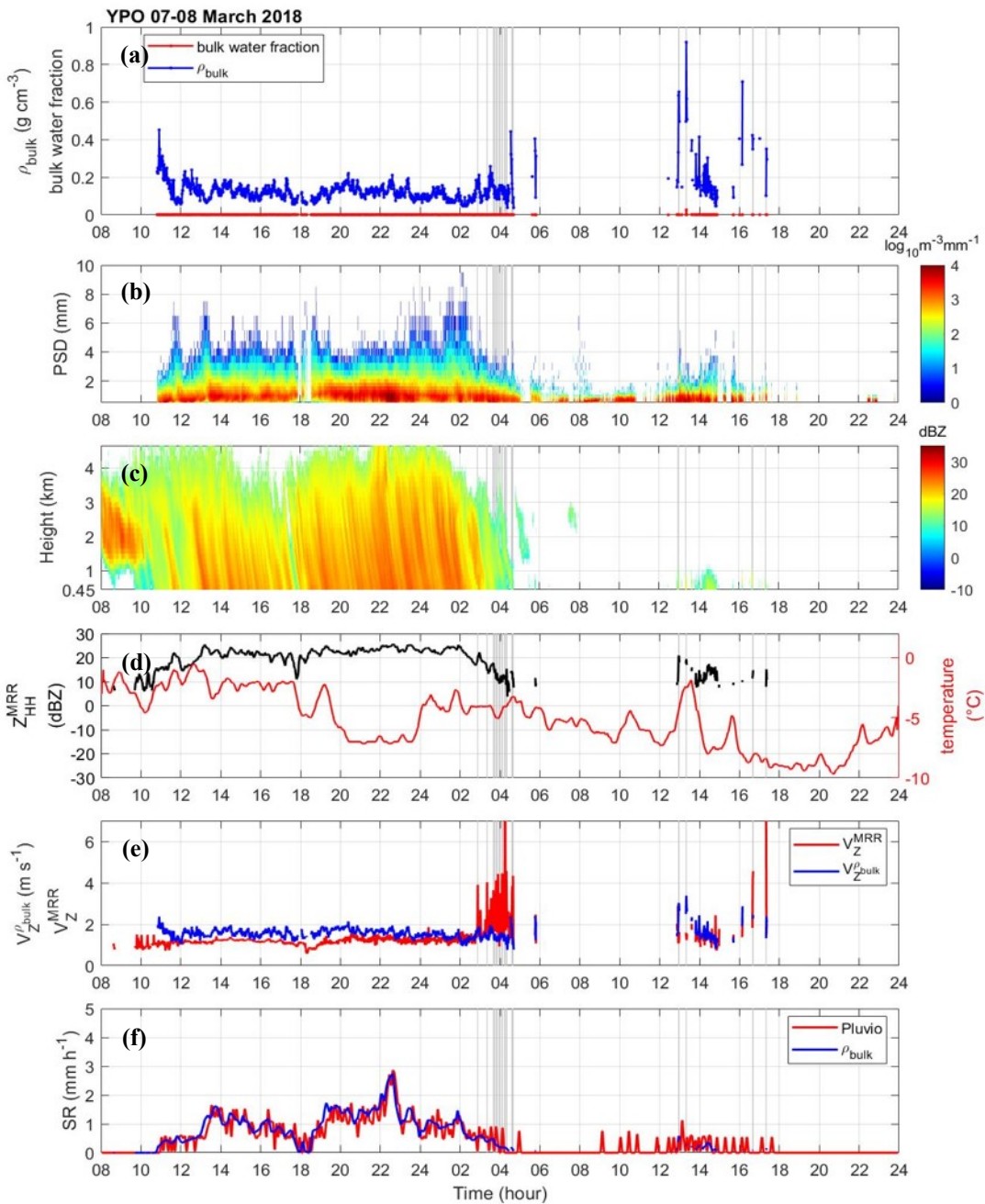

Figure 7: The data from the YPO site on 7 March 2018. (a) Derived bulk density (g cm⁻³, blue line), and bulk water fraction (red line). (b) Parsivel PSD in logarithm scale. (c) The vertical profile of $Z_{HH}$ from the third layer (0.45 km) to the top (dBZ, shaded area) of MRR. (d) The temperature (℃, red line) from nearby AWS (Vaisala WXT520) and $Z_{HH}$ from the third layer of MRR. (e) The velocity ($V_Z^{MRR}$) of MRR measurement (red line) and the $Z_{HH}$ weighted velocity ($V_Z^{\rho bulk}$) calculated from bulk density (blue line). (f) The liquid-equivalent snow-rate (SR) of Pluvio measurement (red line) and bulk density derived SR (blue line).

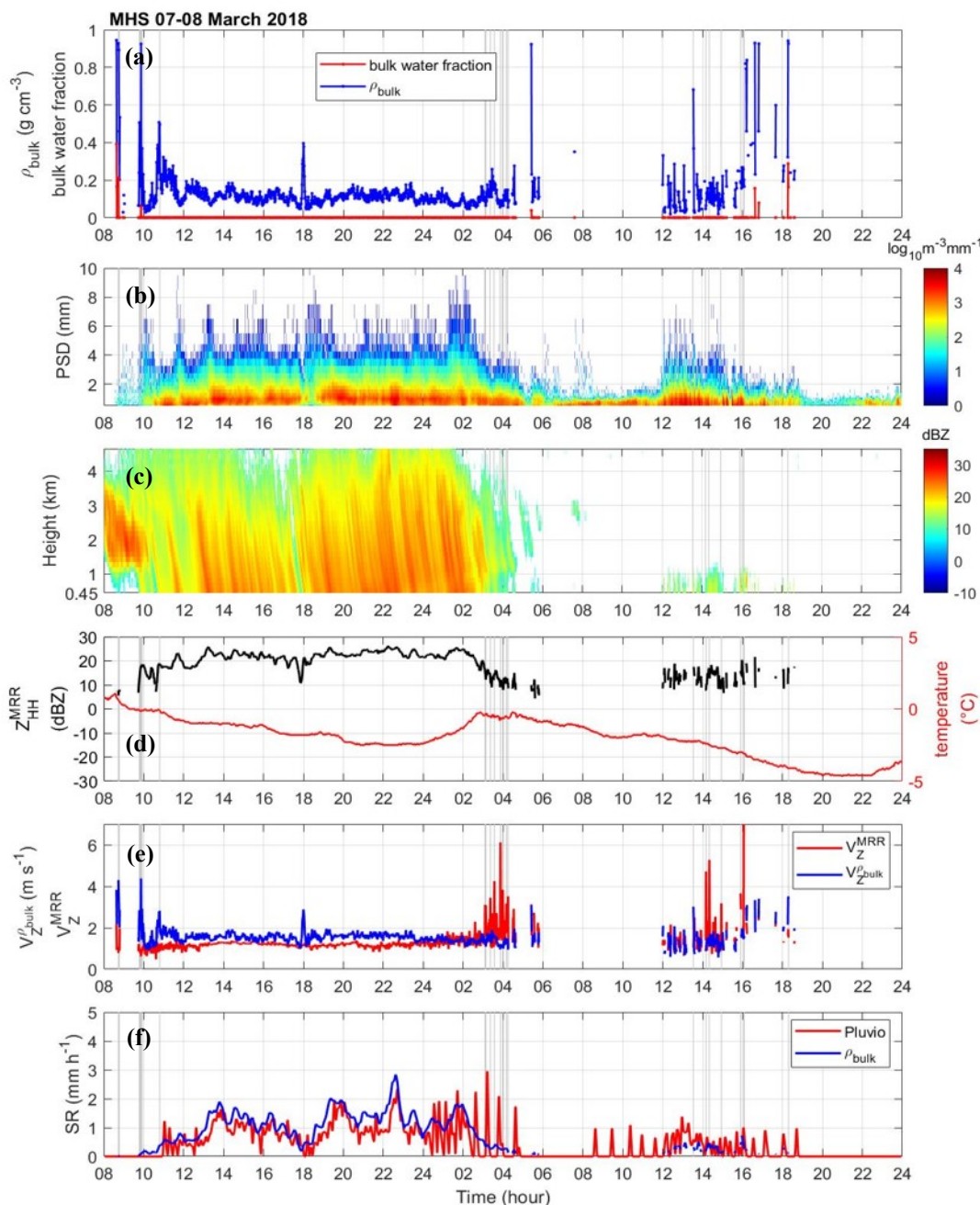

Figure 8: The data from the MHS site on 7 March 2018. (a) Derived bulk density (g cm⁻³, blue line), and bulk water fraction (red line). (b) Parsivel PSD in logarithm scale. (c) The vertical profile of $Z_{HH}$ from the third layer (0.45 km) to the top (dBZ, shaded area) of MRR. (d) The temperature (℃, red line) from nearby AWS (Vaisala WXT520) and $Z_{HH}$ from the third layer of MRR. (e) The velocity ($V_Z^{MRR}$) of MRR measurement (red line) and the $Z_{HH}$ weighted velocity ($V_Z^{\rho bulk}$) calculated from bulk density (blue line). (f) The liquid-equivalent snow rate (SR) of Pluvio measurement (red line) and bulk density derived SR (blue line).

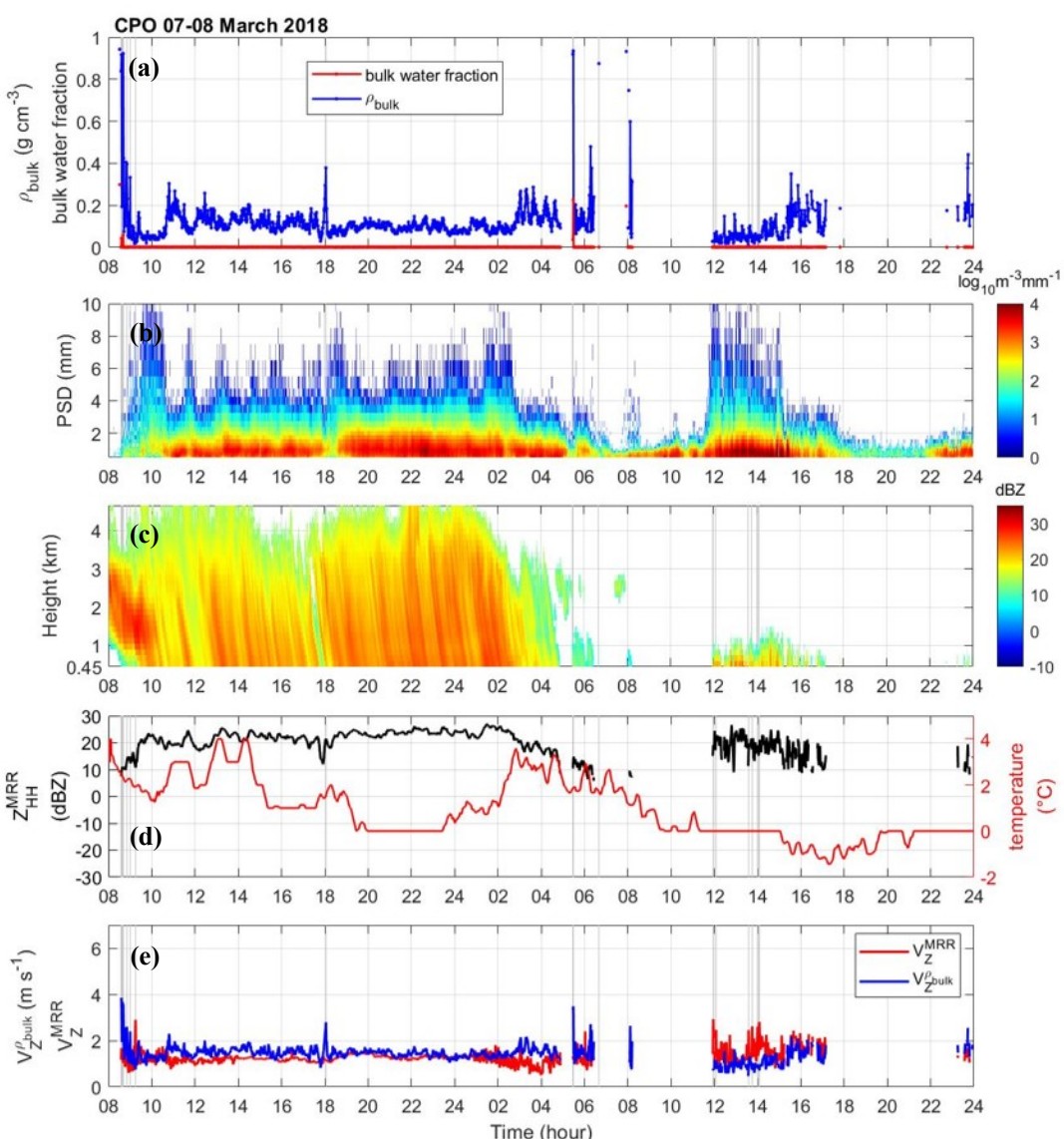

**Figure 9: The data from the CPO site on 7 March 2018. (a) Derived bulk density (g cm$^{-3}$, blue line), and bulk water fraction (red line). (b) Parsivel PSD in logarithm scale. (c) The vertical profile of Z$_{HH}$ from the third layer (0.45 km) to the top (dBZ, shaded area) of MRR. (d) The temperature (°C, red line) from nearby AWS (Vaisala WXT520) and Z$_{HH}$ from the third layer of MRR. (e) The**
**velocity ($V_Z^{MRR}$) of MRR measurement (red line) and the Z$_{HH}$ weighted velocity ($V_Z^{\rho bulk}$) calculated from bulk density (blue line).**

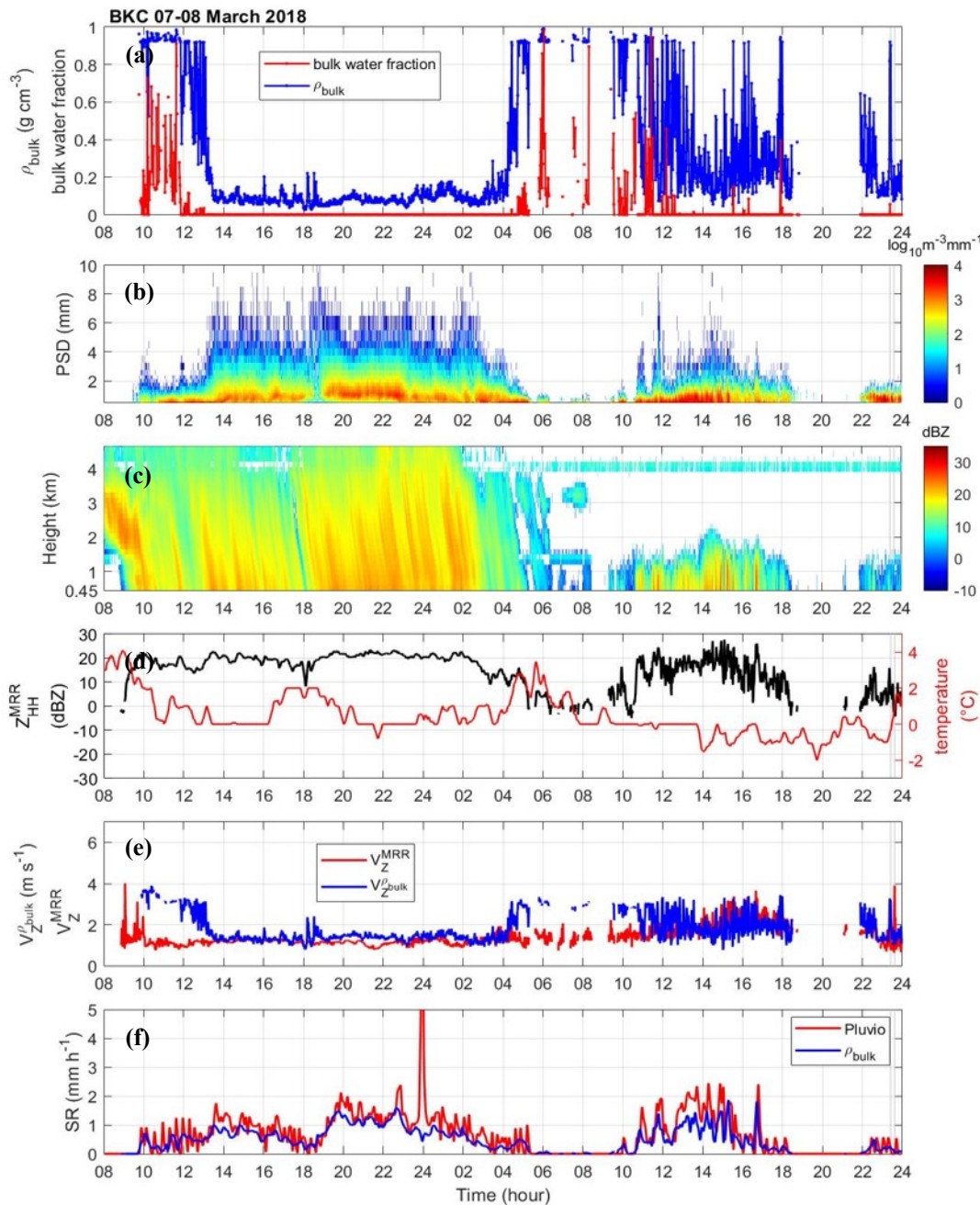

**Figure 10: The data from the BKC site on 7 March 2018. (a) Derived bulk density (g cm⁻³, blue line), and bulk water fraction (red line). (b) Parsivel PSD in logarithm scale. (c) The vertical profile of $Z_{HH}$ from the third layer (0.45 km) to the top (dBZ, shaded area) of MRR. (d) The temperature (℃, red line) from nearby AWS (Vaisala WXT520) and $Z_{HH}$ from the third layer of MRR. (e) The velocity ($V_Z^{MRR}$) of MRR measurement (red line) and the $Z_{HH}$ weighted velocity ($V_Z^{\rho_{bulk}}$) calculated from bulk density (blue line). (f) The liquid-equivalent snow rate (SR) of Pluvio measurement (red line) and bulk density derived SR (blue line).**

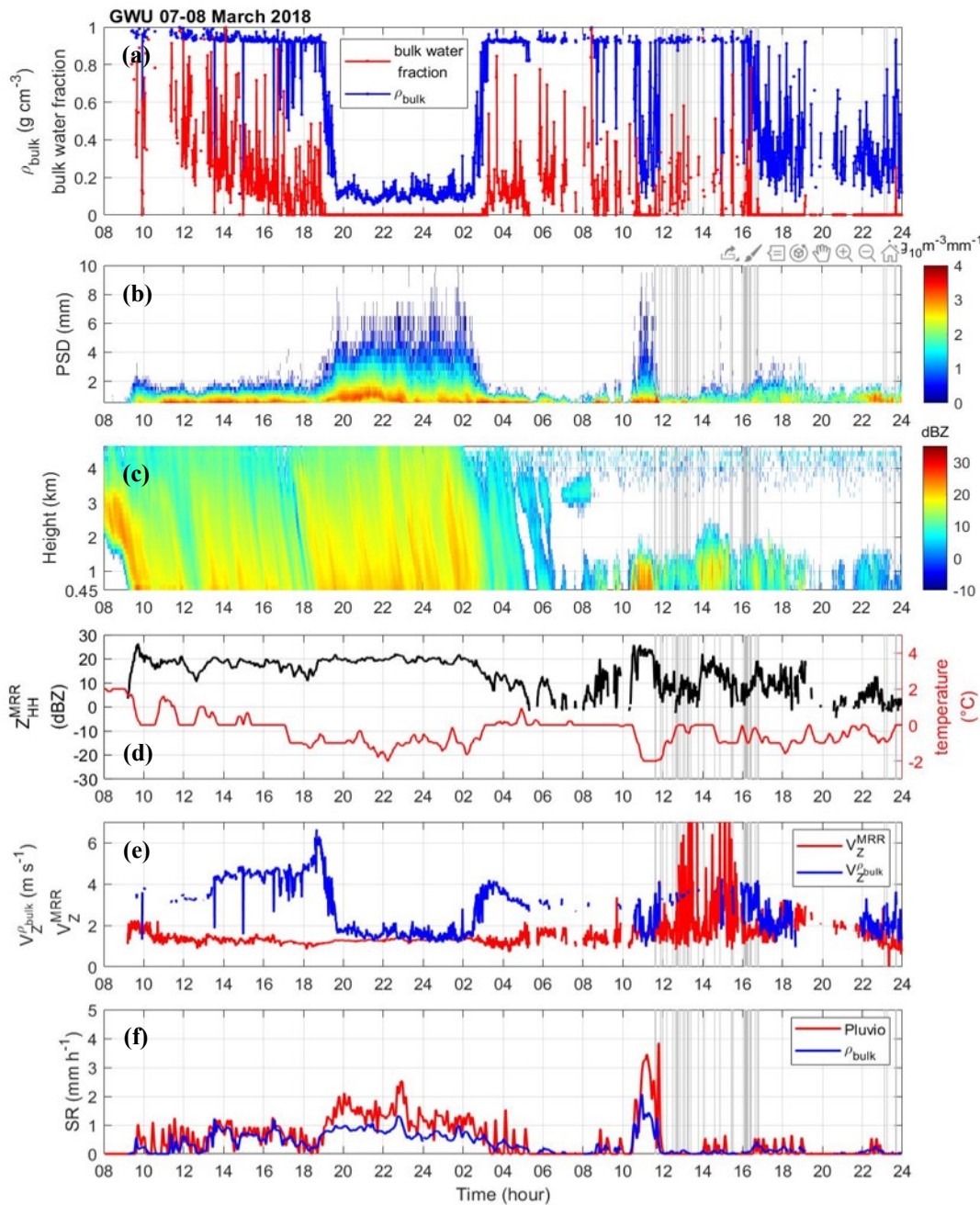

**Figure 11: The data from the GWU site on 7 March 2018. (a) Derived bulk density (g cm$^{-3}$, blue line), and bulk water fraction (red line). (b) Parsivel PSD in logarithm scale. (c) The vertical profile of Z$_{HH}$ from the third layer (0.45 km) to the top (dBZ, shaded area)**
**of MRR. (d) The temperature (℃, red line) from nearby AWS (Vaisala WXT520) and Z$_{HH}$ from the third layer of MRR. (e) The velocity ($V_Z^{MRR}$) of MRR measurement (red line) and the Z$_{HH}$ weighted velocity ($V_Z^{\rho bulk}$) calculated from bulk density (blue line). (f) The liquid-equivalent snow rate (SR) of Pluvio measurement (red line) and bulk density derived SR (blue line).**

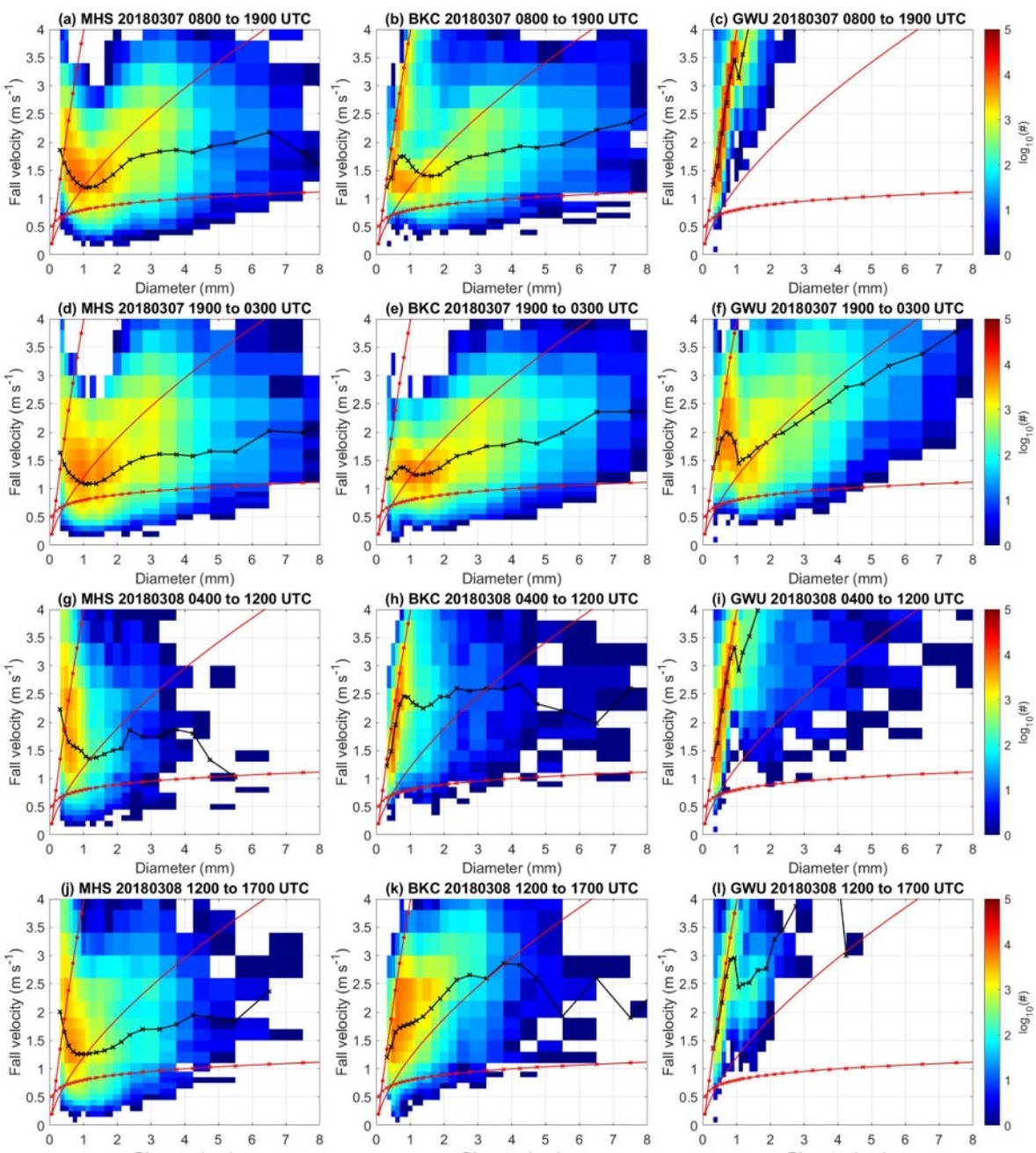

**Figure 12: The number concentration function of fall velocity and diameter from Parsivel (logarithm scale shown in color-shaded) from MHS, BKC, and GWU sites collected from 7-8 March 2018. The average of fall velocity in each diameter bin (m s⁻¹, black line), relations of fall velocity-diameter of rain, graupel, and dry dendrites (from the upper to lower red line) are derived for (a)~(c) 08-19 UTC on 7 March, (d)~(f) 19 UTC on 7 March to 03 UTC on 8 March, (g)~(i) 04-12 UTC on 8 March, and (j)~(l) 12-17 UTC on 8 March in the MHS, BKC, and GWU sites. The relation of rain is from Brandes et al. (2002). The relationships of graupel and dry dendrites are from Locatelli and Hobbs (1974).**

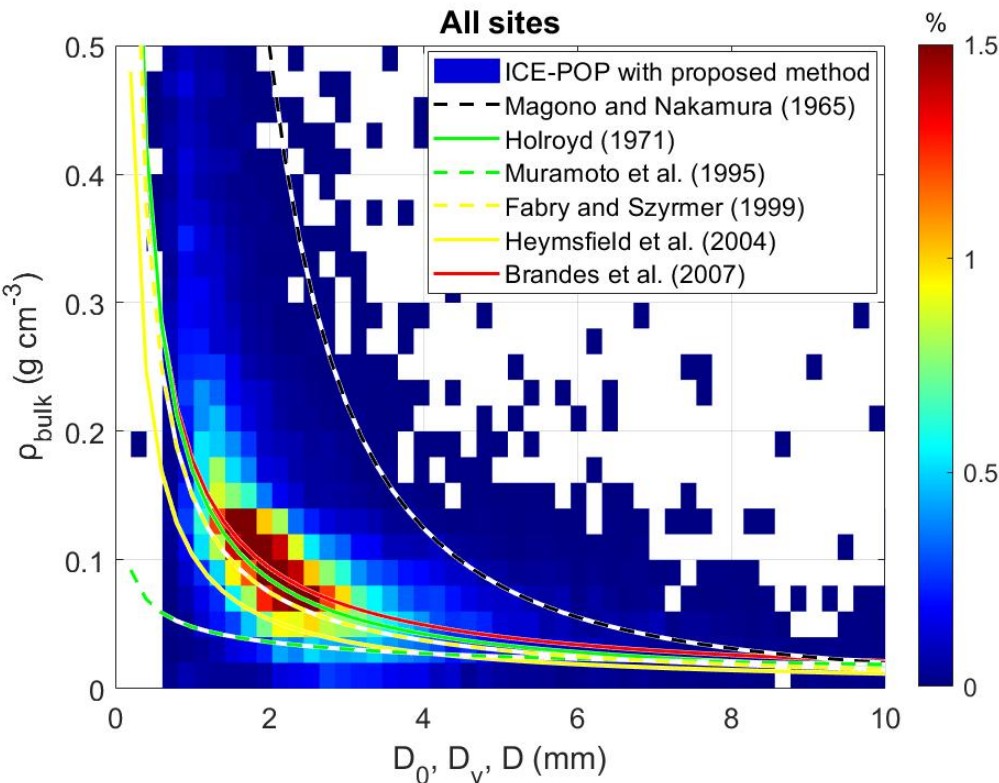

**Figure 13: The number concentration function of retrieved bulk density and observed volume-weighted diameter (Dv) from all sites of the entire ICE-POP 2018 and its pre-campaign is shown in shaded. Various density-particle size relationships are shown, including Brandes et al. (2007), Heymsfield et al. (2004), Magono and Nakamura (1965), Holroyd (1971), Muramoto et al. (1995), and Fabry and Szyrmer (1999). Each study defines particle diameter differently. Please see Table 4 for the particle diameter definition of each study.**

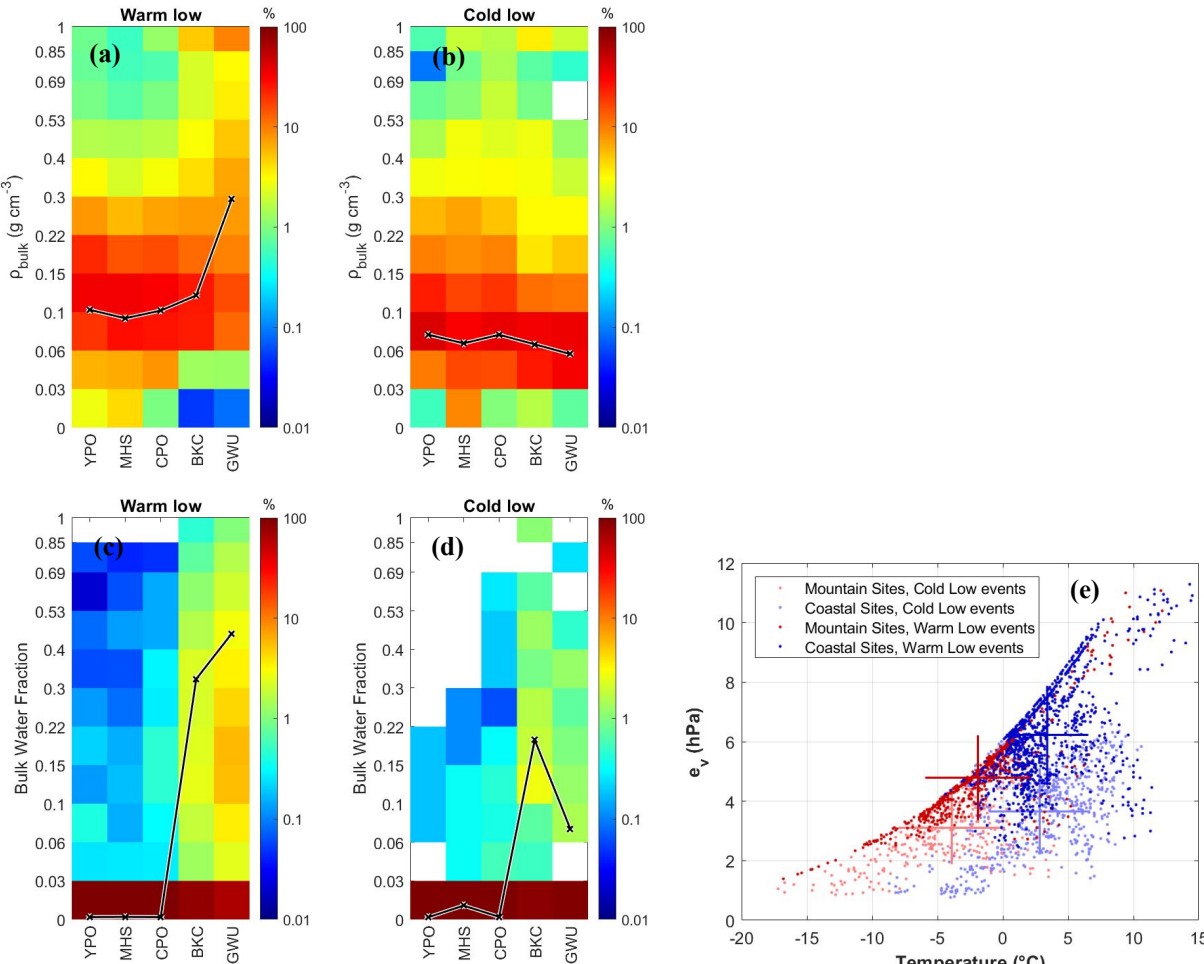

**Figure 14: (a)** The normalized number concentration function of retrieved bulk density of each site for warm-low events. The black line indicates the median bulk density of each site. **(b)** Same as in (a), but for cold-low events. **(c)** The normalized concentration density function of retrieved bulk water fraction of each site for warm-low events. The black line represents the mean values of each site's top 5% bulk water fraction. **(d)** Same as in (c), but for cold-low events. The normalization is individually applied to every site, ensuring that the total sum of the normalized distribution for each site equals 100%. **(e)** The temperature ($^0$C) and water vapor pressure (hPa) measurements from nearby mountain (red and light red dots) and coastal (blue and light blue dots) AWS sites. The blue and red dots are for warm-low events. The light blue and red dots are for cold-low events. The center of the cross is the mean values of temperature ($^0$C) and water vapor pressure (hPa). The standard deviations of temperature ($^0$C) and water vapor pressure (hPa) are shown in bars.

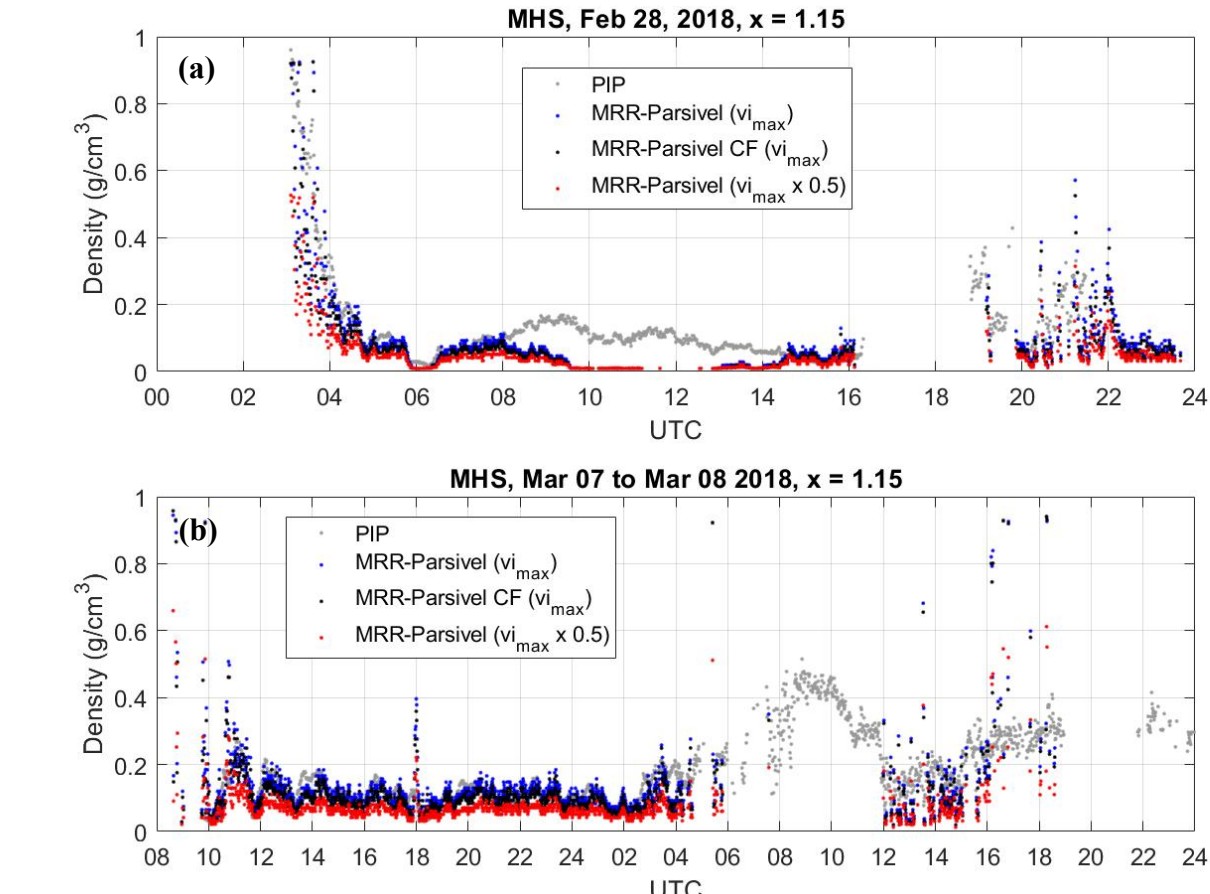

**Figure 15: (a) The retrieved density from PIP (gray dots). Three types of retrieved bulk density from collocated MRR and Parsivel. The blue dots are from the original PSD with maximum ice fraction ($vi_{max}$) assumption. The black dots are retrieved from CF-adjusted PSD with maximum vi assumption. The red dots are retrieved from the original PSD with half-maximum vi ($vi_{max}$ x 0.5) assumption. The case is 28 February 2018. (b) Same as (a), but for case 7 March 2018.**

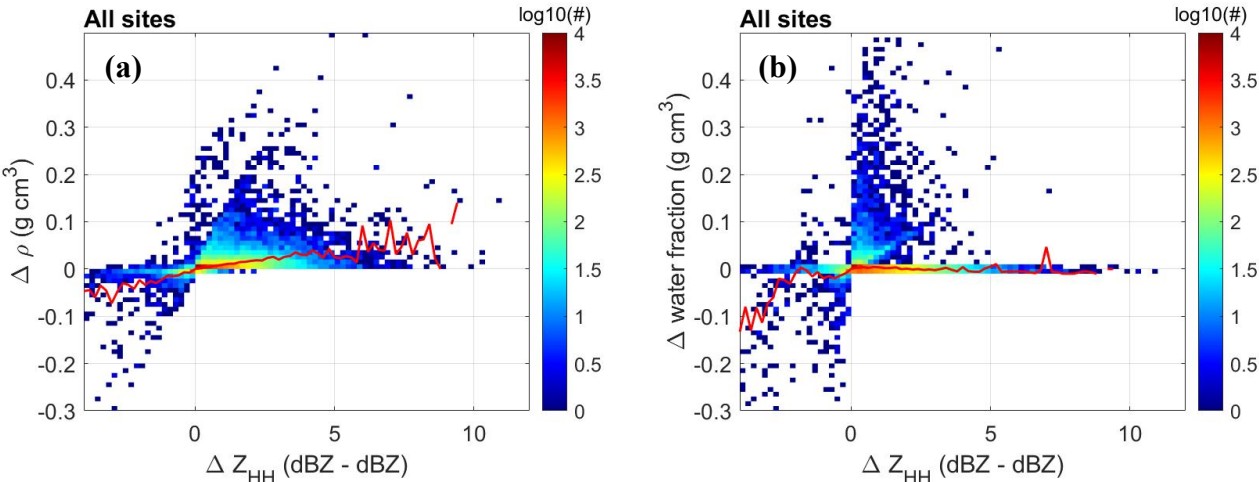

Figure 16: (a) The number concentrations of the bulk density retrieval uncertainty ($\Delta\rho^{bulk}$, g cm-3) to the MRR reflectivity measurement uncertainty ($\Delta Z_{HH}$, dBZ). The positive value of $\Delta Z_{HH}$ indicates MRR reflectivity has a positive error. The positive value of $\Delta\rho^{bulk}$ indicates bulk density has a positive error. The red line indicates the mean bias for uncertainty in the MRR reflectivity measurement. (b) Same as (a), but for bulk water fraction.