# Peer review of "Estimating the Snow Density using Collocated Parsivel and MRR Measurements: A Preliminary Study from ICE-POP 2017/2018"

_EGUsphere, 2023_

## Referee Comment (RC2)

General Comments
################

The authors develop and apply a retrieval that uses measured radar
reflectivities and particle size distributions to estimate bulk particle density
of snowfall and bulk particle liquid fraction. Additional constraint/quality
control is provided by measured radar Doppler velocity. The retrieval is
applied to two snowfall cases with similar synoptic setup that were observed
during the ICE-POP field campaign. The results are compared for the two cases.
Results are further evaluated in terms of terrain effects for the second case.
The retrieval is evaluated by comparing retrieval-derived snowfall rates against
observed snowfall rates.

The topic and retrieval products are very relevant to current meteorological
snowfall research. Understanding the variability of and controls on snowfall
properties is important for remote sensing and for numerical model assessment.

Although relevant, however, I have concerns that the main conclusions of the
study are not well supported by the methodology and the results as they are
presented. I summarize each of the four main conclusions below and describe why
I feel it is not supported.

1. High sensitivity of $Z\_HH$ to the liquid portion of the particle allows for
precise bulk water fraction estimation.

The study provides no evidence that the bulk water (liquid) fraction produced by
the retrieval is consistent with actual bulk liquid water fraction. The claim
that the retrieval can estimate bulk liquid water fraction is supported only by
a brief argument regarding the differing sensitivity of the retrieval forward
model to the water and ice volume fractions.

2. The use of Vz as a filter improves agreement between measured snowfall rates
and snowfall rates estimated from the bulk-density retrievals.

Again, this isn't supported. The study describes retrieval performance using
only data to which the filter has already been applied. It does not show
retrieval results when the filter is not applied, so no judgement can be made
about the effects of the filter.

3. Microphysical similarity between two warm low synoptic events confirms the
dependence of the micro-scale factors on the synoptic conditions.

The results from the study show similarities in the retrieved microphysical
properties for these two events, but that is not sufficient to support the
conclusion. Perhaps other synoptic setups would produce similar microphysical
properties, negating this conclusion.

4. Differences in bulk density and water fraction between mountain sites and
coastal sites are indicative of geographical and synoptic environmental effects
on the distinct microphysical characteristics of winter precipitation systems.

The geographical effects are suggested somewhat by the results from the second
case study, but as noted in regards to conclusion #3, synoptic control can't be
demonstrated using two cases with similar synoptic setup. Further, although the
authors discuss differing meteorological properties at the coastal and mountain
locations, there is no demonstration that the coastal and mountain sites differ
in moisture availability.

I have further significant concerns:

A.  There are no estimates of retrieval uncertainties.  This makes it impossible to determine, for example, if differences between the observed and retrieval-derived snowfall rates are significant.

B.  The description of the methodology is not sufficient.  For example, a Rayleigh reflectivity model is described, but that is not what is used in the radar forward model.  Also, the description of how the ice and liquid volume fractions are determined in the retrieval is unclear and not well justified.

C.  The reflectivity and Doppler velocity data from the MRR require reprocessing to be representative of snowfall.  See Maahn and Kollias (2012).  It's not clear if this or similar reprocessing was performed.

D.  The discussion of results, particularly for the 7-8 March 2018 case, needs to be better organized and cleaned up to more clearly bring focus to the significant patterns in the results.

E.  There are assumptions of spherical particles in both the particle scattering calculations and in the fallspeed calculations, but only limited discussion of whether they are adequate for use with snowflakes and at the MRR's frequency.

F.  The particle "size" measured by the Parsivel is ill-defined for snow particles.  For background, in addition to Battaglia et al., see also Wood et al. (2013).  This makes it difficult to make useful comparisons of bulk particle densities that are determined using different types of disdrometer measurements (e.g., Figure 13).

G.  In further revision, English-language usage and grammar could be improved. I have tried to include some comments in the details below that may be helpful.

Line-by-line comments:
####################

Abstract
* * *
L9:  Does "bulk water fraction" mean bulk *liquid* water fraction?  And is this the volume fraction or mass fraction?

Also, here and in other places, be careful how you use words to describe what you are retrieving.  Here you say "hydrometeor's bulk density and bulk water fraction".  This implies you are retrieving these properties for individual particles, which is not correct.  You are retrieving the bulk density and bulk water fraction for populations of particles.

L13:  The meaning of "The combination of minimum water fraction subsequently determines the bulk density" is not clear.

L15-16:  The meaning of "self-evaluation" in this context is not clear.

L20-21:  Regarding "a similar transition", from what state to what state?

L21:  Again, you are retrieving population properties, not the properties of individual particles.

L28:  Do you actually mean "liquid water content" here, which usually means the concentration (e.g., grams per cubic meter) of liquid water?

Introduction
* * *
L30-31: This doesn't seem to be an example of either of the prior two
statements in this section. What is it intended to exemplify?

L33-34: It's not specifically the increase in liquid phase fraction that
causes the fallspeed of an individual particle to increase. It is the change
in particle aerodynamics, specifically the reduction in particle size and
horizontally-projected area while particle mass stays constant) that causes
the fallspeed to increase.

L38-39: It is not clear what point the authors are making with this
statement. How is the work by Morrison and Milbrandt important to this work?

L40: I think that "inhibits" is not the correct word here.

L47: The results of Brandes et al. (2007) were limited to 52 cases over two
winter seasons and isolated to a particular location. It doesn't seem correct
to me to describe it as "climatological".

L50-51: The meaning of "differentiation of riming degree" is unclear.

L54: Perhaps "above the 2DVD" would be clearer.

L55: I think that it is the difference that is minimized.

L61: I think "disdrometer" is more commonly used.

L64: "transmits" instead of "transmitted".

L65: "scatters" instead of "scattered".

L67: See earlier comment re "distrometer".

L73-74: The meaning of "regarded as the self-evaluation of our result" is not
clear.

Instruments and Data Processing
* * *
L85: Horizontal wind can cause problems with Pluvio and Parsivel
measurements. Was any filtering or correction applied based on ambient wind
speed?

L90-91: What are the elevations of the sites?

L93-95: Is it actually the PSD data that were filtered (PSD data do not
include fall velocities)? Or was it the single-particle size and fallspeed
data that were filtered? How does this filtering affect the calculated PSDs?
Does it reduce particle counts?

L96: What is the vertical resolution of the MRR data? What is the altitude
above ground level of the third gate?

L104-106: "Bias" already implies that an average was taken, no need to say "mean
bias". What are the standard deviations of the differences between the
MRR $Z_{HH}$ and the simulated $Z_{HH}$? This would provide some insight into the uncertainty
of the bias estimates.

Methodology
\*\*\*\*\*\*\*\*\*\*

L107:  In the entirety of the Methodology section, there is no discussion of
how uncertainties are determined for the retrieved properties or properties
derived from the retrieval results.

L112-114:  It is unclear why the Huang et al. study is introduced at the beginning of the
methodology.  The Rayleigh assumption used by Huang et al is clearly not
appropriate for K-band radar (MRR) observing snowfall, so equation 2 is not
applicable.  The actual equation for estimating $Z_{HH}$ using T-matrix
backscatter cross-sections and attenuation is never presented or discussed.

L121:  "modified from Huang et al.", I believe.

L126-127:  More details are needed here.  How were the dielectric properties
of the mixed ice/liquid/air particles determined?  How was the liquid water
assumed to be distributed within a particle?

L128-130:  It's not clear to me why spherical shapes were assumed just because
the snow particles are observed from the bottom.  I believe the particles
would still appear to be non-spherical.  It is not clear that spherical
particle T-matrix calculations are appropriate for modeling snowflake
backscattering a K-band.

L138-144:  Something seems off about the results shown in Figure 2.  The size
distribution in panel (a) shows the size distribution consists of small
particles and that the concentrations of those particles are small.

I would expect the radar reflectivity to be small, in the neighborhood of 0 to
3 dBZ with typical ice densities for snowflakes based on other field
experiment results I've examined with similar size distributions.  Yet
according to the dashed blue line in panel (b) the calculated reflectivity
reaches over 18 dBZ, even with very small ice densities and virtually no
liquid water.

Please check these calculations.

L145-150:  I don't see this recommendation in Huang et al., so I think it is
necessary to explain more fully the reasoning for this approach and to
describe more completely the details of the approach.

Do you mean that given the observed reflectivity, you would just pick the
largest vi that reproduces that reflectivity?  Why?

L149-153:  This part of the methodology also requires more complete
explanation and evidence.  I'm not sure I follow and agree with your argument
here.

Since T-matrix is being used rather than equation (2), it may not be clear to
many readers how the liquid and ice water dielectric factors come into play.
I expect you are using some form of mixing rule (e.g., Maxwell-Garnet?).  I
think explanation needs to be provided about how the particle dielectric
properties are determined for a mixture of ice and liquid water and how this
influences backscattering properties as vi and vw change.

Further, Figure 2b seems to show that there is only a narrow range of the
solution space (vw = 0.015 to 0.1 with vi < 0.5) for which Z might be said to

be moderately more sensitive to vw than to vi due to liquid water's larger dielectric constant.

Also, how is this sensitivity to vw influenced by your method for choosing vi? Clearly, if you pick the maximum vi for this case, there is much weaker sensitivity of Z to vw.

Finally, it is not clear what is meant by "The change of vw can be ... obtained ...".

L153-156: For clarity, I would briefly describe both approaches here, then follow with more detailed descriptions of each one. What is meant by "self-verified"?

L157-165: This is for spherical particles. Do you assert it is appropriate for snow particles? How does this relationship compare with Mitchell and Heymsfield (2005) or Heymsfield and Westbrook (2010)? These newer fallspeed models are more appropriate for snowflakes.

L166-167: This is not a correct statement. Both Vz_MRR and Z_MRR (which is used to constrain the retrieval) are derived from the same basic measurements of Doppler spectra. So they are not independent.

L168-169: But what were this "various issues"?

L170-171: So, my understanding is that, for the data presented in the results, any retrievals with retrieved Vz greater than observed Vz plus one standard deviation are excluded. Is that correct? How does the 1-sigma uncertainty in the obsered Vz compare against the 1-sigma uncertainty in the retrieved Vz?

L176: I think the term on the right of the summation needs to be multiplied by the size bin width (delta_D_i) before summation.

L178: Perhaps "compared against" rather than "examined with".

Results
* * *
L184: The Results contain no assessments of uncertainties in the observations (Z_HH, Vz, PSD, SR) , in the retrieved properties (bulk particle density, bulk liquid water fraction), or in the properties derived from the retrieval results (Z_HH, Vz, SR). How are we to determine if the retrieval results and Vz and SR biases, for example, are significant or not?

Reflectivity-weighted (Vz)
==========================

L197: -0.27 to 0.03 is the range in bias values only, not related to standard deviation.

L199: Clarify that this is the bias and standard deviation for all site results combined.

L200-201: It would be appropriate to acknowledge this limitation earlier in the paper where the method is introduced.

L203: Usually, "mixed-phase".

L203-204: Again, there is a vague reference to "measurement issues", but

there has been no descriptive discussion or quantification of them.

L206-207: This kind of filtering (omitting data from further analysis simply because the data don't give results that match other observations) tends to negate or reduce the believability of the proposed method. This is especially true when the authors cannot point to specific physical conditions that caused the method to fail. How much data was filtered at this stage? How poor are the subsequent results if the data are not filtered?

Liquid-equivalent snowfall rate (SR)
====================================

L215-216: Snow gauges like the Pluvio can have problems with undercatch when surface winds are strong. Were the winds checked and any filtering or corrections applied? The bias in the density-derived SR versus the Pluvio SR might be worse if the Pluvio data are corrected for undercatch.

L216-219: This is the first mention of snow/ice accumulation on the MRR antenna. It would be appropriate to mention that this occurred during the description of the observations earlier in the paper.

L224: Should be "moist air".

L227-228: For the case study of the 28 February event, why is only the MHS site data analyzed?

Case study: 28 February 2018
=============================

L258: Regarding "fall velocity was more significant than 1 m s^-1", I suggest rewording this to avoid confusion with statistical signficance.

L263: Regarding "derivation density", do you mean "derived density"?

L263-264: Are you describing the *maximum* particle sizes?

Case study: 7 March 2018
========================

L268-269: "produced prominent precipitation" and "produced intensive precipitation" sounds like repetition, are both needed?

L273-310: There are a number of locations on these lines that describe bulk water fraction. See my major comments above - I don't think the capability of the retrieval to distinguish and quantify bulk water fraction (or volume fraction of liquid water) has been demonstrated.

L286: Regarding "which are in accord with the distributions of all velocity-diameter relations", it is not clear to me what this means.

L288: Regarding "They gradually dissipated", it is not clear what "They" is referring to.

L293-294: Regarding "Hence, it implies more ... confirm the distribution of fall velocity and diameter". The meaning here is not clear to me.

L296: Regarding "confirmed by the alike contrast", the meaning of "alike

contrast" is not clear.

L298:  Not true, YPO, MHS and CPO, BKC show mostly near-zero bulk water fraction.
For most of this discussion, need to be clear about when only-elevated,
only-coastal, or all sites are being described.

L300:  "Transited" should be "transitioned".

Statistical analysis of bulk density and bulk water fraction
================================================================

L317-320:  What is the basis of the assertion that Brandes et al. (2007)
observations were dominated by "almost spherical aggregates"?  Brandes et al.
appear to have used the equivalent volume diameter as determined by the 2DVD
software, as particle sizes.  These, will be different than the particle size
determined by the Parsivel.  Brandes et al. do use the median volume diameter
to parameterize the bulk density; however it is not evident that the cases in
this study and those of Brandes et al. involved similar meteorological
conditions.  Evidence should be presented for this claim.

L321-325:  The particle sizes used in Heymsfield et al. (2004) are derived
from aircraft particle probes, as you have noted.   These particle sizes
are probably more like the "maximum dimension" of the particle and less like
the "equivalent diameter" determined by a Parsivel.  Additionally, Heymsfield
et al. relate density to mass mean diameter, not to median volume diameter.
So the comparison described here is somewhat an "apples to oranges"
comparison.  It is not surprising there are differences.

L335-344:  As I noted above, I am not convince that this method is capable of
accurately distinguishing and quantifying the liquid and ice volume ratios
and the corresponding bulk water fraction.  Also, although it is asserted that
there are differences in the meteorology of the warm-low and cold-low events
(i.e., "warmer and moister environments" for the warm-low events), no
meteorological data is provided to support this.

L345:  It is probably more appropriate to say that the density of snow varies
with "imposed weather conditions".

Conclusions
* * *
L347-350:  As I've noted, I have concerns about the bulk water fraction
estimates.  I don't believe sufficient proof of the capability has been
provided, and in no way has evidence been provided that the values are
"precise".

L352:  Clarify what is meant by "self-evaluation".

L357:  There's no evidence shown that applying the Vz criteria improves the
consistency of retrieved SR with observed SR.

L359:  Is "all available cases" true?  SR comparison are shown only for two
cases at the sites.

L364-365:  I don't think this statement is supported.  This study has
investigated two cases which have similar synoptic setups and has found
similarity of microphysical characteristics.  But you haven't demonstrated
that different synoptic setups will produce microphysical characteristics
dissimilar to these.

L366:  I would suggest "contrasting" or "dissimilar" rather than "contrastive".

Tables and Figures
* * *
Table 3:  Note previous comment about "mean bias".  Also, why is the Vz criterion for "ALL" shown as "nan"?  To help us understand the significance of the biases and standard deviations, please also include the associated mean values and standard deviations of the observed quantities.

Figure 6:  Is the colorbar axis labeled correctly?  Were there really counts ranging up to 10**50?

Figure 12:  Why does the mountainous MHS site maintain a population of high-fall-velocity small particles throughout the 7-8 March event?

---

## Author Comment (AC1)

Dear Reviewer,

The authors sincerely appreciate your valuable comments and suggestions to help improve the manuscript. We have revised the manuscript titled "Estimating the Snow Density using Collocated Parsivel and MRR Measurements: A Preliminary Study from ICE-POP 2017/2018 ". that was submitted to ACP (Atmospheric Chemistry and Physics) on 3 January, 2024. Based on your suggestions, we have put substantial effort into additional analysis. The manuscript has been thoughtfully revised regarding the comments from all reviewers.

One of the major concerns of the proposed density retrieval algorithm using collocated MRR and Parsivel is lacking the uncertainty analysis. As per the reviewer's suggestion, we have performed substantial investigations of the retrieval uncertainty. The impacts of the measurement uncertainty of the Parsivel and the MRR on the bulk density retrieval are analyzed quantitatively. The measurement issue of Parsivel is also investigated to understand its impact on bulk density retrieval. The results are summarized in the revised manuscript as a Discussion section.

The MRR data quality issue has been examined per the reviewer's suggestion. The post-processed data have replaced entire MRR raw data by applying the algorithm from Maahn and Kollias (2012). All the bulk density, bulk water fraction, and reflectivity-weighted velocity retrievals have been recalculated. The figures have been revised as well.

The original purpose of utilizing reflectivity-weighted velocity to filter adequate retrieval is no longer needed and has been removed in the revised manuscript. The quality of the retrieval results have been greatly improved by applying the post-processed MRR data per the reviewer's suggestion. The low SNR MRR measurement has been removed. The comparison of reflectivity-weighted velocity is mainly used to identify the inadequate retrieval due to the attenuation effect on MRR reflectivity.

The performance of the retrieved bulk density has been validated by the snowfall rate (SR) from collocated Pluvio measurements and reflectivity-weighted fall velocity (Vz) from MRR. In addition to SR and Vz, the performance of the retrieved bulk density has been compared with the precipitation imaging package (PIP), a video disdrometer (Newman et al., 2009; Pettersen et al., 2020). The PIP was also deployed at the MHS site during ICE-POP 2018 (Tokay et al., 2023). The comparison of retrieved bulk density between the proposed algorithm in this study and PIP has shown good agreement with each other. The high consistency further confirms the performance of the retrieved bulk density. Since there is no direct bulk water fraction measurement for validation, the authors consider the validation of bulk density retrieval to PIP and Pluvio as "indirect" evidence to support the bulk water fraction retrieval.

The SR and Vz validation analysis shows that the algorithm can adequately retrieve the bulk density and bulk water fraction. The consistency of the retrieved bulk density to collocated PIP confirms the performance of the proposed algorithm in this study. The advantage of the proposed algorithm is that it utilizes collocated Parsivel and MRR, which are commercially available, commonly used, and robust instruments. The Parsivel and MRR can operate unattentively and need little maintenance. Further application of the proposed algorithm helps derive long-term observation data on snow properties. The authors believe the proposed algorithm can provide an alternative choice if sophisticated instruments (e.g., 2DVD, PIP, SVI, MASC) are unavailable.

The manuscript has also been revised carefully following the reviewer's suggestions on English wording. The authors would like to express our sincere appreciation for the comments. The point-to-point replies to every comment have been prepared. Please see the following replies. The added or modified sentences in the revised manual are in red for your convenience. We would appreciate any feedback on the revisions.

**General comments**

Do the authors see a possibility that the retrieved high values of bulk density and liquid fraction during periods of low reflectivity and precipitation rate could be, at least partly, an artifact from applying the method on observations of very weak precipitation? The retrieval seems somewhat unstable in these conditions which is not surprising considering factors such as low signal to noise ratio due to weak signal. Yet, the authors draw considerable attention to the retrievals of these periods of very weak precipitation. It is worth considering which disciplines would benefit from the microphysical retrievals of weak mixed phase precipitation (<0.5mm/h)? I would suggest either introducing a threshold for minimum reflectivity or precipitation rate where the method is applied, or otherwise critically reviewing the method's performance in weak precipitation, where the precipitation rate falls below the sensitivity of the Pluvios.

**Reply: The retrieved bulk density and bulk water fraction as a function of density are shown in Figure R1. Most retrieved bulk density is obtained from MRR reflectivity from 10 to 30 dBZ. Some retrieval results with high bulk density and low MRR reflectivity can be found, but not frequent.**

**The quality of the retrieval results has been greatly improved by applying the post-processed MRR data as per the reviewer's suggestion. The low SNR MRR measurement has been removed. The authors intend to preserve as much data as possible by only eliminating the data with an attenuation effect on MRR reflectivity. The attenuated MRR reflectivity underestimates the retrieved bulk density and bulk water fraction.**

[Figure]

**Figure R1: (a) The retrieved bulk density as a function of MRR reflectivity. (b) Same as (b), but for bulk water fraction.**

I don't feel that the retrieval of $v_w$ is adequately demonstrated in the case studies since there seem to be no time periods where there would be both a) agreement between derived and measured terminal velocity and b) notable precipitation intensity at the same time. It raises the concern how the method would perform in mixed phase situations with larger particles or higher precipitation rates. In my mind, the best way to address this would be to replace one of the case studies with another one that has significant intensity of wet snow, or discuss the possible limitations of the method's application.

**Reply: The bulk water fraction is derived along with the maximum possible bulk density using the proposed method in this study. If a different assumption is made when selecting possible bulk density, the retrieved bulk water fraction will be different. Therefore, the performance of the retrieved bulk water fraction is linked with bulk density retrieval. Since there are no direct measurements of bulk water fraction, we will compare the retrieved bulk density from the proposed method and PIP. The consistency between retrieved bulk density from the two algorithms confirms that the retrieved bulk water should be reasonable.**

**As shown in Fig. R2(a), the retrieved bulk density values from the proposed algorithm and PIP gradually decrease from nearly 1.0 to 0.1 (g cm⁻³) between 03 and 06 UTC. Both algorithms capture the transition from the mixing-phase to dry snow. Please see Figure R2. The manuscript has been revised to include a discussion of bulk water fraction retrieval. Please see Line 398-417.**

**In Figure 14, the coastal sites (BKC and GWU) were associated with higher retrieved bulk water fraction. The environment parameters, including temperature (⁰C) and vapor pressure (hPa), show that the coastal sites have higher temperatures and more water vapor. Please see Lines 339-357.**

[Figure]

**Figure R2: (a) The retrieved bulk density from collocated MRR and Parsivel. The blue dots are retrieved from CF-adjusted PSD. The red dots are from the original PSD. The gray dots are the retrieval from PIP. The case is 28 February 2018. (b) Same as (a), but for case 7 March 2018.**

In the presented case studies, it seems like non-zero $v_w$ values only occur when bulk density is nearly saturated at over 0.9 g/cm. I'm concerned whether this is physically reasonable especially given the assumption of spherical particles. This raises the question whether, effectively, the liquid fraction would act as a kind of an overflow buffer in the calculations when density alone cannot explain the reflectivity values. This could rise from the assumption of maximum ice volume fraction ($v_i$). Or could it be explained with the drizzle-like nature of the precipitation during the $v_w$ signals? This concern could be dispelled with a counter example.

**Reply: The assumption of the particle shape has been discussed in the revised manuscript. Please see Lines 386-397.**

**As the reply to the previous comment, the bulk water fraction is derived along with the maximum possible bulk density in the proposed method in this study. The performance of the retrieved bulk water fraction is linked with bulk density retrieval. Since there are no direct measurements of bulk water fraction, the consistency between retrieved bulk density from the two algorithms confirms that the retrieved bulk water**

**should be reasonable. Please see Figure R2. The discussion of bulk water retrieval can be found in Lines 398-417 of the revised manuscript.**

The manuscript lacks discussion on the implications of assuming spherical particles. This might be significant consideration given the wide range of different particle habits and their shapes and preferred falling alignments.

**Reply: The discussion of the assuming spherical particles has been included in the revised manuscript. Please see Lines 386-397. The particle shape assumption for falling velocity and reflectivity calculation has been discussed. Please see Lines 218-221.**

Since the manuscript considers the liquid fraction, it would be worth showing or at least mentioning if melting layer signals were detected in the MRR observations.

**Reply: Most cases in this study have surface temperatures near zero or lower than zero degrees. As shown in Figures 5 and 7-11, there is no pronounced bright-band signature from MRR radar data.**

The viewpoint in the manuscript is more technical and focuses less on microphysics. As such, the topic is suitable for ACP but might be even better suited for AMT. This is a possible consideration for resubmission after revisions.

**Reply: The manuscript contains the retrieval technique description and the retrieval result analysis. The retrieval technique is demonstrated by substantial bulk density analysis and water fraction evolution. The features of these bulk properties of warm-low and cold-low events are also investigated.**

**Specific comments**

The title refers to snow density, but retrievals are attempted for snow, mixed phase precipitation and light drizzle. If liquid fraction plays an important role in the revised manuscript, it would be good to be reflected in the title.

**Reply: The majority of the particles in this study are snow. The algorithm estimates bulk water fraction value, and the mixing phase of snow can be subsequently differentiated from dry snow. As the bulk water fraction equals one, thus the particle is considered a drizzle or raindrop. Hence, the main objective is to estimate the snow bulk density, including dry and wet snow.**

The title indicates this is a preliminary study. Are the authors working on a more comprehensive analysis? Worth mentioning in the discussion.

**Reply: The proposed method has been introduced and applied to ICE-POP 2018 data. The analysis also shows that the algorithm can adequately retrieve the bulk density and bulk water fraction. The advantage of the proposed algorithm is that it utilizes collocated Parsivel and MRR, which are commonly used and robust instruments. The Parsivel and MRR can operate unattentively and need little maintenance. Further application of the proposed algorithm helps derive long-term observation data on snow properties. The discussion has been added to the revised manuscript. Please see Lines 441-447.**

L42: This sentence seems to suggest that riming and melting are the only processes affecting snow density. While these are important, one should not forget that, e.g., the primary particle habit and aggregation have great impact, too.

**Reply: The aggregation process has been added to the revised manuscript. Please Lines 31-33.**

L59: As one of the selling points of the new density retrieval method is the use of robust intsruments that require little maintenance, perhaps it would be good to mention studies that use, e.g., PIP or Parsivel for density retrievals. Such studies could be found, for example, by doing citation analysis on Brandes et al. (2007) and Huang et al. (2010), as referred to in the manuscript.

**Reply: Per the reviewer's suggestion, the PIP (precipitation imaging package, Newman et al., 2009; Pettersen et al., 2020) was also deployed at the MHS site during ICE-POP 2018. Tokay et al. (2023) have utilized PIP to investigate the PSD parameters, including mass-weighted diameter and normalized intercept. The bulk density is estimated by Tokay et al. (2023) with various assumptions. The PIP retrieved density was generated from the assumption that $D_{max} = 1.15\ D_{eq}$, and the mass derivation included was based on Bohm (1989). The time series of retrieved bulk density from the proposed algorithm and PIP are shown in Fig. R2. The consistency between retrieved bulk density from the two algorithms confirms that the retrieved bulk water should be reasonable. Please see Lines 398-413 in the revised manuscript.**

**In addition, the MASC (multi-angle snowflake camera) is also introduced in the revised manuscript. Please see Lines 59-62.**

Section 2: Details about the ambient temperature measurements are missing, in particular, the types of the instruments or sensors used. The temperature measurements, while not part of the main retrieval methods presented, should be of great interest for the reader as an indication for melting and other microphysical processes.

**Reply: The temperature measurements were derived from collocated AWS (Vaisala WXT520). The comparison between the temperature measurements from AWS and PARSIVEL has shown that WXT520 has better performance. The Parsivel data had a significant bias in the Parsivel temperature data. The information on the temperature sensor has been added to the revised manuscript. Please see Fig. 5, 7-11.**

Section 2: I would like to see basic information about the Pluvios used such as make, orifice size and shielding.

**Reply: The Pluvios are OTT Pluvio² - Weighing Rain Gauge. All of the Pluvios were equipped with double windshields. The Pluvio at the MHS was within the DFIR (double fence intercomparison reference) in addition to the double shield. All the sites investigated in this study have no taller trees or buildings near the MRR antenna and Parsivel. The environmental conditions of all sites are introduced in the revised manuscript; please see Lines 92-103.**

L130: What is meant by a canting angle when referring to spherical particles?

**Reply: The discussion of non-spherical particles has been introduced in the revised manuscript. Non-spherical and spherical particle measurements are different when MRR looks upward. However, the orientation of the non-spherical particle is assumed to be isotropic and homogeneous. A sensitivity investigation assuming the particle axis ratio of 0.5 has been conducted. The results show that about 1.5 dBZ variation of simulated reflectivity can be induced due to the assumption of particle size.**

**A random error of MRR reflectivity with a standard deviation of 1.2 dB is introduced into the retrieval algorithm to imitate the particle assumption and MRR measurement uncertainty. The overall standard deviation of bulk density retrieval uncertainty is about 0.025 (g cm$^{-3}$) for a given MRR reflectivity uncertainty of 1.2 dB. The bulk water fraction retrieval has the same feature, and the uncertainty is about 0.041. Please see the Discussion section in the revised manuscript.**

L133: Why was this temperature range chosen for the simulations? Is it physically reasonable to simulate mixed-phase precipitation in -10 degrees Celsius, for example?

**Reply: The temperature is set to $0^0$C in the T-matrix simulation. The sensitivity test of temperature is shown in Fig. 1. The results indicate that the reflectivity simulation is insensitive to particle temperature. Please see Lines 150-155.**

L147: I failed to find references to liquid water fraction from Huang et al. (2010). They seem to just assume particles to consist of ice and air. Either I missed it, or this reference could be more accurate.

**Reply: Huang et al. (2010) assumed that a mixture of snow contains only ice and air. Please see page 642 of Huang et al. (2010), "To calculate the backscattering properties of the particles measured by the 2DVD, we consider snow to be a mixture of ice and air.".**

**The manuscript has been revised to improve the clarity. Please see Lines 167-168.**

L263: "The particle size was" to 'Maximum particle size ranged from'

**Reply: The sentence has been revised as per the reviewer's suggestion. Please see Lines 276-277.**

L279: Since there are many sites involved in this study, it could be useful to have a small map showing their locations.

**Reply: This study is part of a series of studies of ICE-POP 2018. Many published papers contain such information. The authors would like to use references to provide such information to keep the manuscript concise. "The instruments were located in nineteen sites across the Gangwon region on the east coast of Korea (see Kim et al. 2021 for detailed information of each site)." Please Lines 93-94.**

L315: "number density function" to 'number concentration'

**Reply: The sentence has been revised as per the reviewer's suggestion. Please see Lines 325-326.**

L331: How is mean bulk density calculated here? Is it integrated over the total volume?

**Reply: The mean bulk density is replaced by the median value of bulk density. The median value is obtained for each site. Please see Line 341.**

L334: Since rain was not excluded from the retrievals, we are now talking about the mean bulk density of mixed precipitation instead of snow. As the densities of rain and snow are quite different, the mean value is easily driven by the fraction of rain, masking the possible signal from snow properties.

**Reply: The number concentration of the retrieved bulk density is shown in Fig. 14. Both dry snow and mixing-phase events are shown in both warm-low and cold-low events.**

**The median values of the retrieved bulk density of each site are shown in Fig. 14a,b. The temperature ($^0$C) and water vapor pressure (hPa) measurements from nearby mountain and coastal AWS sites are collected and summarized in Fig. 14e. The warm-low events have warmer and moister conditions compared to cold-low events. The warm- and cold-low events in the coastal area have similar mean temperature values. On the other hand, the water vapor pressure increases significantly from cold-low to warm-low events. The mountain area has similar features but with higher temperature increments and fewer increments of water vapor pressure. Please see Lines 339-357.**

L345: This sentence seems to suggest that the density of snow has an impact on the weather. It's unclear to me what was meant here. Please rephrase to clarify.

**Reply: The sentence has been revised. Please see Lines 419.**

L526: "The ZHH variation with vw is much less than that with vi". I think, the opposite is true.

**Reply: It should be "On the contours of $Z_{HH}$, the $Z_{HH}$ variation with vw is much less than that with vi." The sentence has been rephrased; please see Lines 628.**

L560: What does the "shaded area" refer to in Fig. 5c?

**Reply: The "shaded area" refers to the MRR reflectivity profile. The sentence has been revised to improve the clarity. "The time series of MRR $Z_{HH}$ vertical profile (dBZ) from the third gate (0.45 km) to the 5 km." Please see Fig. 5 and 7-11.**

Figures 5d, 9d-11d: There seem to be flat parts in the temperature measurements, where the measured value does not seem to change even for a fraction of a degree. These look like a measurement errors, perhaps gaps in the measurements. This raises concerns about the quality of the temperature measurements. The measurements should be checked and erroneous values omitted from the analysis. If there is a cause for concern about the quality of the measurements, it should be discussed in the manuscript.

**Reply: The temperature measurements were derived from collocated AWS (Vaisala WXT520). The comparison between the temperature measurements from AWS and Parsivel has shown that WXT520 has better performance. The Parsivel data had a significant bias in the Parsivel temperature data. The information on the temperature sensor has been added to the revised manuscript. The scale range of the temperature in Fig. 5, 7-11 is also improved for clarity.**

Figure 6: The integration times in these figures are quite long. For example, in Fig. 6b, there seem to be multiple modes in the (D, v) distribution. Because of the long integration time, it is unclear if these modes are co-existing or if the dominating particle type is evolving over time. Have the authors considered analyzing the particle properties such as (D, v) distribution in shorter time intervals?

**Reply: The authors did examine shorter time intervals for (D, v) distribution. The (D, v) distribution showed better differentiation between dry and mixing-phase snow. Both Fig. 6 and 12 can illustrate the transition of the mixing phase to dry snow. Considering the number of the figure and keeping the manuscript concise, we would like to maintain the current format of Fig. 6 and Fig. 12.**

**Technical comments**

L10: Authors should choose between the spellings "disdrometer" and "distrometer". Currently, mixed spelling is used for this word in the manuscript.

**Reply: The "distrometer" has been revised to "disdrometer".**

L27: Since riming refers to a process and not a hydrometeor type, change "riming" to 'rimed particles'.

**Reply: The "riming" has been revised to "rimed particles". Please see Line 27.**

L73-74: The readability of this sentence could be improved by rephrasing.

**Reply: Please see Lines 81-82. The sentence has been revised: Subsequently, the measurement of $Z_{HH}$ weighted fall velocity ($V_Z$) from MRR is compared with the calculated $V_Z$ from the derived bulk density and Parsivel PSD measurement.**

L121: The authors probably meant 'modified FROM Huang et al. (2010)'.

**Reply: The typo has been corrected. Please see Lines 138.**

L142: "as the increasing" to 'with increasing'

**Reply: The sentence has been revised as per the reviewer's suggestion. Please see Line 161.**

L178: "examined with" to 'compared with'

**Reply: The sentence has been revised as per the reviewer's suggestion. Please see Line 193.**

L186: "four with Pluvio" to 'four of them equipped with a Pluvio'

**Reply: The sentence has been revised as per the reviewer's suggestion. Please see Line 200.**

L196: "consistency of" to 'consistent'

**Reply: The sentence has been revised as per the reviewer's suggestion. Please see Line 209.**

L232: Full stop after "shown in Fig. 5"

**Reply: The sentence has been revised as per the reviewer's suggestion. Please see Line 347.**

L267: "relatively weaker" to 'weaker'. There are also other instances of using "relatively" with a comparative in the manuscript. I advice against this.

**Reply: The sentence has been revised as per the reviewer's suggestion. Please see Line 280. Other similar comparatives have been revised as well.**

L274: "precipitation system" to 'precipitation'

**Reply: The sentence has been revised as per the reviewer's suggestion. Please see Line 283.**

L303: "has a more consistent relation" to 'is more consistent with the relation'

**Reply: The sentence has been revised as per the reviewer's suggestion. Please see Line 314.**

L345: "microphysics processes" to 'microphysical processes'

**Reply: The sentence has been revised as per the reviewer's suggestion. Please see Line 419.**

L353: "Consistency" to 'Inconsistency'

**Reply: The sentence has been revised as per the reviewer's suggestion. Please see Line 425.**

L361: The manuscript could be more consistent with how dates are written: with or without ordinal indicators.

**Reply: The sentence has been revised as per the reviewer's suggestion. Please see Line 431.**

L501: "blue lines" to 'blue', as there are also other symbols than lines.

**Reply: The sentence has been revised as per the reviewer's suggestion. Please see Line 603.**

L524: "DSD" to 'PSD'

**Reply: The sentence has been revised as per the reviewer's suggestion. Please see Line 624.**

L569: "derived" to 'shown'

**Reply: The sentence has been revised as per the reviewer's suggestion. Please Line 669.**

Figure 5d: "MRM" to 'MRR' in the y-axis label.

**Reply: The typo has been corrected. Please see Fig. 5d.**

Figure 6c: label (c) missing from figure

**Reply: The typo has been corrected. Please see Fig. 6c.**

---

## Author Comment (AC2)

Dear Reviewer,

The authors sincerely appreciate your valuable comments and suggestions to help improve the manuscript. We have revised the manuscript titled "Estimating the Snow Density using Collocated Parsivel and MRR Measurements: A Preliminary Study from ICE-POP 2017/2018 ". that was submitted to ACP (Atmospheric Chemistry and Physics) on 3 January, 2024. Based on your suggestions, we have put substantial effort into additional analysis. The manuscript has been thoughtfully revised regarding the comments from all reviewers.

One of the major concerns of the proposed density retrieval algorithm using collocated MRR and Parsivel is lacking the uncertainty analysis. As per the reviewer's suggestion, we have performed substantial investigations of the retrieval uncertainty. The impacts of the measurement uncertainty of the Parsivel and the MRR on the bulk density retrieval are analyzed quantitatively. The measurement issue of Parsivel is also investigated to understand its impact on bulk density retrieval. The results are summarized in the revised manuscript as a Discussion section.

The MRR data quality issue has been examined per the reviewer's suggestion. The post-processed data have replaced entire MRR raw data by applying the algorithm from Maahn and Kollias (2012). All the bulk density, bulk water fraction, and reflectivity-weighted velocity retrievals have been recalculated. The figures have been revised as well.

The original purpose of utilizing reflectivity-weighted velocity to filter adequate retrieval is no longer needed and has been removed in the revised manuscript. The quality of the retrieval results have been greatly improved by applying the post-processed MRR data per the reviewer's suggestion. The low SNR MRR measurement has been removed. The comparison of reflectivity-weighted velocity is mainly used to identify the inadequate retrieval due to the attenuation effect on MRR reflectivity.

The performance of the retrieved bulk density has been validated by the snowfall rate (SR) from collocated Pluvio measurements and reflectivity-weighted fall velocity (Vz) from MRR. In addition to SR and Vz, the performance of the retrieved bulk density has been compared with the precipitation imaging package (PIP), a video disdrometer (Newman et al., 2009; Pettersen et al., 2020). The PIP was also deployed at the MHS site during ICE-POP 2018 (Tokay et al., 2023). The comparison of retrieved bulk density between the proposed algorithm in this study and PIP has shown good agreement with each other. The high consistency further confirms the performance of the retrieved bulk density. Since there is no direct bulk water fraction measurement for validation, the authors consider the validation of bulk density retrieval to PIP and Pluvio as "indirect" evidence to support the bulk water fraction retrieval.

The SR and Vz validation analysis shows that the algorithm can adequately retrieve the bulk density and bulk water fraction. The consistency of the retrieved bulk density to collocated PIP confirms the performance of the proposed algorithm in this study. The advantage of the proposed algorithm is that it utilizes collocated Parsivel and MRR, which are commercially available, commonly used, and robust instruments. The Parsivel and MRR can operate unattentively and need little maintenance. Further application of the proposed algorithm helps derive long-term observation data on snow properties. The authors believe the proposed algorithm can provide an alternative choice if sophisticated instruments (e.g., 2DVD, PIP, SVI, MASC) are unavailable.

The manuscript has also been revised carefully following the reviewer's suggestions on English wording. The authors would like to express our sincere appreciation for the comments. The point-to-point replies to every comment have been prepared. Please see the following replies. The added or modified sentences in the revised manual are in red for your convenience. We would appreciate any feedback on the revisions.

General Comments ################

1. High sensitivity of $Z_{HH}$ to the liquid portion of the particle allows for precise bulk water fraction estimation.

The study provides no evidence that the bulk water (liquid) fraction produced by the retrieval is consistent with actual bulk liquid water fraction. The claim that the retrieval can estimate bulk liquid water fraction is supported only by a brief argument regarding the differing sensitivity of the retrieval forward model to the water and ice volume fractions.

**Reply: The bulk water fraction is derived along with the maximum possible bulk density in the proposed method in this study. If different assumption is made in the selection of possible bulk density, the retrieved bulk water fraction will be different. Therefore, the performance of the retrieved bulk water fraction is linked with bulk density retrieval. Since there are no direct measurements of bulk water fraction, we will compare the retrieved bulk density from the proposed method and PIP. The consistency between retrieved bulk density from two algorithm confirms that the retrieved bulk water should be reasonable. Please see Figure R1.**

**As shown in Fig. R1(a), the retrieved bulk density values from the proposed algorithm and PIP gradually decrease from nearly 1.0 to 0.1 (g cm$^{-3}$) between 03 and 06 UTC. Both algorithms capture the transition from the mixing-phase to dry snow. Please see Figure R1 in the previous reply. The manuscript has been revised to include the discussion of the bulk water fraction retrieval. Please see Line 398-413.**

[Figure]

**Figure R1: (a) The retrieved bulk density from collocated MRR and Parsivel. The blue dots are retrieved from CF-adjusted PSD. The red dots are from the original PSD. The gray dots are the retrieval from PIP. The case is 28 February 2018. (b) Same as (a), but for case 7 March 2018.**

2. The use of Vz as a filter improves agreement between measured snowfall rates and snowfall rates estimated from the bulk-density retrievals.

Again, this isn't supported. The study describes retrieval performance using only data to which the filter has already been applied. It does not show retrieval results when the filter is not applied, so no judgement can be made about the effects of the filter.

**Reply: We did calculate the SR before and after Vz filter, and the results did show great improvements. As per the reviewer's suggestion of the data quality of MRR, the MRR data has been replaced by the post-processed data by applying the algorithm from Maahn and Kollias (2012). All of the retrievals of the bulk density, bulk water fraction and reflectivity-weighted velocity have been recalculated. The original noisy retrieval results have been removed due to low SNR. The Vz calculation is no longer used to remove noisy bulk density retrieval. The Vz comparison is derived to verify the overall performance of bulk density retrieval. In addition, the Vz can identify the inadequate bulk density retrieval due to attenuation effect on MRR reflectivity. Please see Line 185-187.**

3. Microphysical similarity between two warm low synoptic events confirms the dependence of the micro-scale factors on the synoptic conditions.

The results from the study show similarities in the retrieved microphysical properties for these two events, but that is not sufficient to support the conclusion. Perhaps other synoptic setups would produce similar microphysical properties, negating this conclusion.

**Reply: The warm low events (28 February 2018 and 7 March 2018) shown in the manuscripts are the events that have the most accumulated snowfall during ICE-POP. The 28 February case is shown to demonstrate that the retrieved snow density is reasonable when compared to Pluvio SR. However, some discrepancy of the calculated Vz from retrieved bulk density and measurements of Vz from MRR can be noticed. The discrepancy is due to the attenuation effect on MRR antenna. The authors do not intend to emphasize the similarity of microphysical properties. These two events are to demonstrate the detail evolution of the bulk density retrieval results. The discussion has been revised. Please see Lines 339-357.**

4. Differences in bulk density and water fraction between mountain sites and coastal sites are indicative of geographical and synoptic environmental effects on the distinct microphysical characteristics of winter precipitation systems.

The geographical effects are suggested somewhat by the results from the second case study, but as noted in regards to conclusion #3, synoptic control can't be demonstrated using two cases with similar synoptic setup. Further, although the authors discuss differing meteorological properties at the coastal and mountain locations, there is no demonstration that the coastal and mountain sites differ in moisture availability.

**Reply: The number concentration of the retrieved bulk density is shown in Fig. 14. Both dry snow and mixing-phase events are shown in both warm-low and cold-low events. The median values of the retrieved bulk density of each site are shown in Fig. 14a,b. The temperature ($^0C$) and water vapor pressure (hPa) measurements from nearby mountain and coastal AWS sites are collected and summarized in Fig. 14e. The warm-low events have warmer and moister conditions compared to cold-low events. The warm- and cold-low events in the coastal area have similar mean temperature values. On the other hand, the water vapor pressure increases significantly from cold-low to warm-low events. The mountain area has similar features but with higher temperature increments and fewer increments of water vapor pressure. Please see Lines 339-357 of the revised manuscript.**

A. There are no estimates of retrieval uncertainties. This makes it impossible to determine, for example, if differences between the observed and retrieval-derived snowfall rates are significant.

**Reply: The retrieval uncertainty is investigated as per the reviewer's suggestion. The retrieval uncertainty analysis is performed by considering the assumption of particle shape, Parsivel measurement uncertainty, and the MRR measurement uncertainty. Please see the Discussion section for detail retrieval uncertainty analysis, Lines 358-417.**

B. The description of the methodology is not sufficient. For example, a Rayleigh reflectivity model is described, but that is not what is used in the radar forward model. Also, the description of how the ice and liquid volume fractions are determined in the retrieval is unclear and not well justified.

**Reply: The description of the methodology has been improved as per the reviewer's suggestion. The description of the calculation of reflectivity has been revised to improve the clarity. The equation 2 is no longer Rayleigh reflectivity model. The backscattering cross-section ($\sigma$) replaces $D^6$ in equation 2. The reference of Bringi and Chandrasekar (2001) is provided to replace Huang et al. (2010). The ice and water are assumed to be evenly distributed within the particle. Please see Lines 125-171 of the revised manuscript.**

C. The reflectivity and Doppler velocity data from the MRR require reprocessing to be representative of snowfall. See Maahn and Kollias (2012). It's not clear if this or similar reprocessing was performed.

**Reply: The MRR data has been post-processed as per the reviewer's suggestion. The values of bulk density and bulk water fraction have been recalculated. The impact of the $Z_{HH}$ difference to the values of bulk density and bulk water fraction can be considered as the retrieval uncertainty due to MRR reflectivity measurement uncertainty. The results are shown in Discussion section. Please see Lines 111-114.**

D. The discussion of results, particularly for the 7-8 March 2018 case, needs to be better organized and cleaned up to more clearly bring focus to the significant patterns in the results.

**Reply: The discussion of the 7-8 March 2018 case has been rewritten to imprve the clarity. Please see Lines 278-322.**

E. There are assumptions of spherical particles in both the particle scattering calculations and in the fallspeed calculations, but only limited discussion of whether they are adequate for use with snowflakes and at the MRR's frequency.

**Reply: The discussion of non-spherical particles has been introduced in the revised manuscript. Non-spherical and spherical particle measurements are different when MRR looks upward. However, the orientation of the non-spherical particle is assumed to be isotropic and homogeneous. A sensitivity investigation assuming the particle axis ratio of 0.5 has been conducted. The results show that about 1.5 dBZ variation of simulated reflectivity can be induced due to the assumption of particle size.**

**A random error of MRR reflectivity with a standard deviation of 1.2 dB is introduced into the retrieval algorithm to imitate the particle assumption and MRR measurement uncertainty. The overall standard deviation of bulk density retrieval uncertainty is about 0.025 (g cm$^{-3}$) for a given MRR reflectivity uncertainty of 1.2 dB. The bulk water fraction retrieval has the same feature, and the uncertainty is about 0.041. Please see the Discussion section in (Lines 358-417) the revised manuscript.**

F. The particle "size" measured by the Parsivel is ill-defined for snow particles. For background, in addition to Battaglia et al., see also Wood et al. (2013). This makes it difficult to make useful comparisons of bulk particle densities that are determined using different types of disdrometer measurements (e.g., Figure 13).

**Reply: In Figure 13, the bulk density comparison among this study, Heymsfield et al. (2004), and Brandes et al. (2007) does not intend to emphasize the difference. We convert D0 to Dm by assuming exponential PSD (Dm = D0*4/3.67). Considering distinct environmental conditions, instrumentations, and retrieval techniques, most of the particles in this study are consistent with the $\rho_{bulk} - D_m$ relation from Heymsfield et al. (2004) and Brandes et al. (2007). These results indicate that the proposed bulk density estimation algorithm can derive accurate retrievals with statistically consistent microphysical characteristics from previous studies. Please see Lines 324-338.**

F. In further revision, English-language usage and grammar could be improved. I have tried to include some comments in the details below that may be helpful.

**Reply: Authors sincerely appreciate the reviewer's help. We have done all the correction as per both reviewer's suggestion. We will further improve the manuscript by asking professional assistance from English editor.**

Line-by-line comments: #######################

Abstract ********

L9: Does "bulk water fraction" mean bulk *liquid* water fraction? And is this the volume fraction or mass fraction?

Also, here and in other places, be careful how you use words to describe what you are retrieving. Here you say "hydrometeor's bulk density and bulk water fraction". This implies you are retrieving these properties for individual particles, which is not correct. You are retrieving the bulk density and bulk water fraction for populations of particles.

**Reply: The sentence has been revised to improve the clarity. "derive bulk density and bulk water fraction of a population of particles …". Please see Line 9. The bulk water fraction is the volume fraction, not the mass fraction. Please see Lines 126-128.**

L13: The meaning of "The combination of minimum water fraction subsequently determines the bulk density" is not clear.

**Reply: The sentence has been revised to improve the clarity. "The combination of minimum water fraction and maximum ice fraction subsequently determines the bulk density ($\rho_{bulk}$)." Please see Lines 13-14.**

L15-16: The meaning of "self-evaluation" in this context is not clear. L20-21: Regarding "a similar transition", from what state to what state?

**Reply: The "self-evaluation" has been removed in the revised manuscript. "The estimated $\rho_{bulk}$ was examined independently by comparison of the liquid-equivalent snowfall rate (SR) of collocated Pluvio." Please see Lines 15-16.**

L21: Again, you are retrieving population properties, not the properties of individual particles.

**Reply: The sentence has been revised to improv the clarity. Please see Lines 21-22.**

L28: Do you actually mean "liquid water content" here, which usually means the concentration (e.g., grams per cubic meter) of liquid water?

**Reply: Yes, authors refer liquid water content (LWC, g cm$_{-3}$). The LWC has been constantly retrieved by dual-polarimetric radar measurements.**

Introduction ************

L30-31: This doesn't seem to be an example of either of the prior two statements in this section. What is it intended to exemplify?

**Reply: The examples intend to show how microphysical parameterizations in numerical forecast models can be improved by validating with the snow property obtained from observational data. The sentence has been revised to improve the clarity. Please see Lines 33-44.**

L33-34: It's not specifically the increase in liquid phase fraction that causes the fallspeed of an individual particle to increase. It is the change in particle aerodynamics, specifically the reduction in particle size and horizontally-projected area while particle mass stays constant) that causes the fallspeed to increase.

**Reply: The sentence has been revised as per the reviewer's suggestion. Please see Line 38-39.**

L38-39: It is not clear what point the authors are making with this statement. How is the work by Morrison and Milbrandt important to this work?

**Reply: The study from Morrison and Milbrandt (2015) proposed a new microphysical scheme that parameterizes the density of hydrometers. A robust density estimation algorithm can evaluate microphysical simulations from numerical models. Please see the revised manuscript, Line 44.**

L40: I think that "inhibits" is not the correct word here.

**Reply: The typo has been corrected. Please see Lines 30.**

L47: The results of Brandes et al. (2007) were limited to 52 cases over two winter seasons and isolated to a particular location. It doesn't seem correct to me to describe it as "climatological".

**Reply: The "climatological" has been removed. Please see Lines 48.**

L50-51: The meaning of "differentiation of riming degree" is unclear.

**Reply: The sentence has been revised to improve the clarity. Please see Lines 51-52.**

L54: Perhaps "above the 2DVD" would be clearer.

**Reply: The sentence has been revised as per the reviewer's suggestion. Please see Line 55.**

L55: I think that it is the difference that is minimized.

**Reply: The sentence has been revised as per the reviewer's suggestion. Please see Line 55.**

L61: I think "disdrometer" is more commonly used.

**Reply: The "distrometer" has been revised to "disdrometer". Please see Line 69.**

L64: "transmits" instead of "transmitted".

**Reply: The "transmitted" has been revised to "transmits". Please see Line 71.**

L65: "scatters" instead of "scattered".

**Reply: The "scattered" has been revised to "scatters". Please see Line 72.**

L67: See earlier comment re "distrometer".

**Reply: The "distrometer" has been revised to "disdrometer". Please see Line 74.**

L73-74: The meaning of "regarded as the self-evaluation of our result" is not clear.

**Reply: The "regarded as the self-evaluation of our result" has been removed to improve the clarity. Please see Lines 81-82.**

Instruments and Data Processing ******************************

L85: Horizontal wind can cause problems with Pluvio and Parsivel measurements. Was any filtering or correction applied based on ambient wind speed?

**Reply: All of the Pluvios were equipped with double windshields. The Pluvio at the MHS was within the DFIR (double fence intercomparison reference) in addition to the double shield. The environmental condition of all sites are introduced in the revised manuscript, please see Lines 99-102.**

L90-91: What are the elevations of the sites?

**Reply: The elevations of each site are, YPO (772 m MSL), MHS (789 m a.m.s.l.), CPO (855 m MSL), BKC (175 m MSL), and GWU (36 m MSL). Each site's detailed layout and information can be found in Kim et al. (2021). The reference is added in the revised manuscript. Please see Lines 96-98.**

L93-95: Is it actually the PSD data that were filtered (PSD data do not include fall velocities)? Or was it the single-particle size and fallspeed data that were filtered? How does this filtering affect the calculated PSDs? Does it reduce particle counts?

**Reply: The Parsivel single-particle size and fallspeed data were filtered. The minute Parsivel data was quality-controlled using the fall velocity filtering technique (Lee et al. 2015). The filtering removes the outlier particles, reducing particle counts. Subsequently, the PSD was calculated from the filtered data. The manuscript has been revised to improve clarity. Please see Lines 107-110.**

L96: What is the vertical resolution of the MRR data? What is the altitude above ground level of the third gate?

**Reply: The MRRs had the same configuration during ICEP-POP; the vertical resolution was 150 m, and there were 31 gates up to 4.65 km. The third gate is 450 m above ground. The vertical resolution information has been added to the revised manuscript. Please see Lines 111-114.**

L104-106: "Bias" already implies that an average was taken, no need to say "mean bias". What are the standard deviations of the differences between the MRR Z_HH and the simulated Z_HH? This would provide some insight into the uncertainty of the bias estimates.

**Reply: The "mean bias" has been revised to "bias" per the reviewer's suggestion. The number concentration plot of MRR measured reflectivity, and Parsivel PSD calculated reflectivity, as shown below in Figure R2. The standard deviation of the differences between them is about 1.1 to 1.3 dB for each site (shown in Table 2 of the revised manuscript). As the reviewer indicates that the standard deviation value can be considered as the MRR reflectivity uncertainty. The standard deviation will be further applied to investigate the bulk density retrieval uncertainty.**

[Figure]

**Figure R2: The number concentration distribution measured of reflectivity from MRR and simulated reflectivity from Parsivel of BKC site. The bias is -2.1 dBZ, and the standard deviation is 1.28 dBZ.**

Methodology ***********

L107: In the entirety of the Methodology section, there is no discussion of how uncertainties are determined for the retrieved properties or properties derived from the retrieval results.

**Reply: As per the revidwer's sggestion, the retrieval uncertainty has been investigated and summarized in the revised manuscript. A Discussion has been added in the repvsed mnauscript. Please See Lines 358-417.**

**With 1.2 dB MRR reflectivity uncertainty, the retrieval bulk density uncertainty is about 0.023 g cm$^{-1}$. An Discussion section has been added for discussing the retrieval uncertainty. Please see Lines 386-397.**

L112-114: It is unclear why the Huang et al. study is introduced at the beginning of the methodology. The Rayleigh assumption used by Huang et al is clearly not appropriate for K-band radar (MRR) observing snowfall, so equation 2 is not applicable. The actual equation for estimating Z_HH using T-matrix backscatter cross-sections and attenuation is never presented or discussed.

**Reply: Authors agree with reviewer's comment. Since T-Matrix simulation is used and the Rayleigh assumption is not used in the retrieval algorithm, the equation 2 has been revised. The reference has also been changed to Bringi and Chandrasekar (2001). The sentence of Rayleigh assumption is removed to improve the manuscript's clarity. Please see Lines 130-134.**

L121: "modified from Huang et al.", I believe.

**Reply: The typo has been corrected. Please see Line 138.**

L126-127: More details are needed here. How were the dielectric properties of the mixed ice/liquid/air particles determined? How was the liquid water assumed to be distributed within a particle?

**Reply: The ice and water are evenly distributed within the particle. The manuscript has been revised to improve the clarity. Please see Lines 144-145.**

L128-130: It's not clear to me why spherical shapes were assumed just because the snow particles are observed from the bottom. I believe the particles would still appear to be non-spherical. It is not clear that spherical particle T-matrix calculations are appropriate for modeling snowflake backscattering a K-band.

**Reply: Non-spherical and spherical particles do appear differently when looking upward. However, the orientation of the non-spherical particle is assumed to be isotropic and homogeneous. A sensitivity investigation assuming the particle axis ratio of 0.5 has been conducted. The results show that about there is about a 1.5 dBZ variation of simulated reflectivity can be induced due to the assumption of particle size.**

**A random error of MRR reflectivity with a standard deviation of 1.2 dB is introduced to imitate the particle assumption and MRR measurement uncertainty to the retrieval uncertainty. The overall standard deviation of bulk density retrieval is about 0.025 (g cm⁻³) for a given MRR reflectivity uncertainty of 1.2 dB. The bulk water fraction retrieval has the same feature and the uncertainty is about 0.041. Please see Discussion section in the revised manuscript. Please See Lines 358-417.**

L138-144: Something seems off about the results shown in Figure 2. The size distribution in panel (a) shows the size distribution consists of small particles and that the concentrations of those particles are small. I would expect the radar reflectivity to be small, in the neighborhood of 0 to 3 dBZ with typical ice densities for snowflakes based on other field experiment results I've examined with similar size distributions. Yet according to the dashed blue line in panel (b) the calculated reflectivity reaches over 18 dBZ, even with very small ice densities and virtually no liquid water.

Please check these calculations.

**Reply: The calculated reflectivity values reaching over 18 dBZ is due to the high water fraction and high dielectric constant. The calculated reflectivity from PSD is ranging from -5 to 35 dBZ, the values do cover 0 to 3 dBZ (low ice and water fraction). The 18 dBZ is the reference reflectivity value from MRR to constrain the retrieved bulk density. We will use the bulk density retrieval from precipitation imaging package (PIP) to verify the calculation of T-Matrix simulation and retrieved density.**

**The PIP, a video disdrometer, provides the PSD, fall speed, density, and snowfall rate of hydrometers (Newman et al., 2009; Pettersen et al., 2020) was also deployed at the MHS site during ICE-POP 2018. Tokay et al. (2023) have utilized PIP to investigate the PSD parameters, including mass-weighted diameter and normalized intercept. The bulk density is estimated by Tokay et al. (2023) with various assumptions. The PIP retrieved**

density was generated from the assumption that $D_{max} = 1.15 D_{eq}$, and the mass derivation included was based on Bohm (1989). The time series of retrieved bulk density from proposed algorithm and PIP are shown in Fig. R1.

We change the data time of Figure 2 to 1559 UTC in the revised manuscript. The PSD has higher concentration of particles, thus the reflectivity values are ranging from –5 to 50 dBZ. The MRR reference reflectivity is 22.23 dB. Please see the Fig. 2 in the revised manuscript. As shown in Fig. R1, the retrieved bulk density from the proposed algorithm (blue and red dots) and the PIP (gray dots) are in good agreements. The bulk density was about 0.07 (g cm$^{-1}$) from both methods. The consistency of retrieved bulk density confirms the calculation of the bulk density and the simulated reflectivity from PSD. The discussion of the consistency between the proposed method and the PIP has been included in the the revised manuscript. Please see Lines 388-413 (the Discussion section).

L145-150: I don't see this recommendation in Huang et al., so I think it is necessary to explain more fully the reasoning for this approach and to describe more completely the details of the approach. Do you mean that given the observed reflectivity, you would just pick the largest vi that reproduces that reflectivity? Why?

Reply: One difference between our algorithm and Huang et al.'s (2010) is the assumption of the particle composition. Huang et al. (2010) assumed that a mixture of snow contains only ice and air. Please see page 642 of Huang et al. (2010), "To calculate the backscattering properties of the particles measured by the 2DVD, we consider snow to be a mixture of ice and air." The assumption of only ice and air from Huang et al. (2010) is the same as the vw (water fraction) equals zero in Fig. 2b. The bottom part of Fig. 2b (e.g., vw = 0) is exactly the same as Huang et al. (2010).

In our proposal algorithm, the reflectivity calculation fully considers water/ice/air fractions. Therefore, various water/ice/air fraction combinations can be derived from the matched reflectivity between MRR measurement and Parsivel calculation. To determine the bulk snow density from these possible combinations of vw/vi, the bulk density with maximum vi (minimum vw) with maximum bulk density is selected, similar to Huang et al. (2010)'s assumption. The manuscript has been revised to improve the clarity, please see Lines 157-171.

L149-153: This part of the methodology also requires more complete explanation and evidence. I'm not sure I follow and agree with your argument here.

Reply: The bulk water fraction is derived along with the maximum possible bulk density in the proposed method in this study. If different assumption is made in the selection of

possible bulk density, the retrieved bulk water fraction will be different. Therefore, the performance of the retrieved bulk water fraction is linked with bulk density retrieval. Since there are no direct measurements of bulk water fraction, we will compare the retrieved bulk density from the proposed method and PIP. The consistency between retrieved bulk density from two algorithm confirms that the retrieved bulk water should be reasonable.

As shown in Fig. R1(a), the retrieved bulk density values from the proposed algorithm and PIP gradually decrease from nearly 1.0 to 0.1 (g cm$^{-3}$) between 03 and 06 UTC. Both algorithms capture the transition from the mixing-phase to dry snow. Please see Figure R1. The manuscript has been revised to include the discussion of the bulk water fraction retrieval. Please see Line 398-413.

Since T-matrix is being used rather than equation (2), it may not be clear to many readers how the liquid and ice water dielectric factors come into play. I expect you are using some form of mixing rule (e.g., Maxwell-Garnet?). I think explanation needs to be provided about how the particle dielectric properties are determined for a mixture of ice and liquid water and how this influences backscattering properties as vi and vw change.

Further, Figure 2b seems to show that there is only a narrow range of the solution space (vw = 0.015 to 0.1 with vi < 0.5) for which Z might be said to be moderately more sensitive to vw than to vi due to liquid water's larger dielectric constant. Also, how is this sensitivity to vw influenced by your method for choosing vi? Clearly, if you pick the maximum vi for this case, there is much weaker sensitivity of Z to vw. Finally, it is not clear what is meant by "The change of vw can be ... obtained ...".

Reply: The mixing rule of Maxwell-Garnet is applied in T-Matrix calculation. The influence of vi/vw composition to the backscattering properties has been discussed in the manuscript. Please Lines 134-135.

As the reply to the previous comment, the performance of the retrieved bulk water fraction is linked with bulk density retrieval. Since there are no direct measurements of bulk water fraction, we will compare the retrieved bulk density from the proposed method and PIP. The retrieved bulk density values from the proposed algorithm and PIP gradually decrease from nearly 1.0 to 0.1 (g cm$^{-3}$) between 03 and 06 UTC in Fig R1(a). Both algorithms capture the transition from the mixing-phase to dry snow. Please see Figure R1. The manuscript has been revised to include the discussion of the bulk water fraction retrieval. Please see Line 398-413.

L153-156: For clarity, I would briefly describe both approaches here, then follow with more detailed descriptions of each one. What is meant by "self-verified"?

**Reply: The description has been revised to improve the clarity. The original purpose of utilizing reflectivity-weighted velocity to filter adequate retrieval is no longer needed and has been removed in the revised manuscript. The quality retrieval results have been greatly improved by applying the post-processed MRR data per the reviewer's suggestion. The low SNR MRR measurement has been removed. The comparison of reflectivity-weighted velocity is mainly used to identify the inadequate retrieval due to the attenuation effect on MRR reflectivity. Please see Lines 185-187.**

L157-165: This is for spherical particles. Do you assert it is appropriate for snow particles? How does this relationship compare with Mitchell and Heymsfield (2005) or Heymsfield and Westbrook (2010)? These newer fallspeed models are more appropriate for snowflakes.

**Reply: As per the reviewer's suggestion, the retrieval results has been greatly improved after applying the post-processed MRR data (Maahn and Kollias 2012). The noisy bulk density has been removed. The original purpose of removing inadequate bulk density retrieval by reflectivity-weighted velocity is no longer needed. The reflectivity-weighted velocity comparison is obtained for two purposes. First, the comparison intends to examine the overall performance of the retrieved bulk density. The overall consistency is shown in Fig. 3. Second, the "bulk density-derived" reflectivity-weighted velocity is obtained to identify antenna attenuation issue as shown in Fig. 5(e). The issue of spherical particle assumption on terminal velocity calculation has been discussed in the revised manuscript. Please see Lines 218-221.**

L166-167: This is not a correct statement. Both Vz_MRR and Z_MRR (which is used to constrain the retrieval) are derived from the same basic measurements of Doppler spectra. So they are not independent.

**Reply: The manuscript has been revised as per the reviewer's suggestion. Please see Lines 185-187.**

L168-169: But what were this "various issues"?

**Reply: The manuscript has been revised as per the reviewer's suggestion. Please see Lines 185-187.**

L170-171: So, my understanding is that, for the data presented in the results, any retrievals with retrieved Vz greater than observed Vz plus one standard deviation are excluded. Is that

correct? How does the 1-sigma uncertainty in the obsered Vz compare against the 1-sigma uncertainty in the retrieved Vz?

**Reply: The V$_Z$ criteria for removing inadequate bulk density retrieval is no longer needed. As per the reviewer's suggestion, the retrieval results have been greatly improved after applying the post-processed MRR data (Maahn and Kollias 2012). The noisy bulk density has been removed. The Vz difference is mainly for identifying the attenuation effect of MRR reflectivity. Please see Lines 185-187.**

L176: I think the term on the right of the summation needs to be multiplied by the size bin width (delta_D_i) before summation.

**Reply: The typo has been corrected. Please see Line 191.**

L178: Perhaps "compared against" rather than "examined with".

**Reply: The sentence has been revised as per the reviewer's suggestion. Please see Line 193.**

Results *******

L184: The Results contain no assessments of uncertainties in the observations (Z_HH, Vz, PSD, SR) , in the retrieved properties (bulk particle density, bulk liquid water fraction), or in the properties derived from the retrieval results (Z_HH, Vz, SR). How are we to determine if the retrieval results and Vz and SR biases, for example, are significant or not?

**Reply: As per the reviewer's suggestion, the discussion of the retrieval uncertainty has been included in the revised manuscript. Please the Discussion section for more detail. (Lines 358-417).**

Reflectivity-weighted (Vz) ===========================

L197: -0.27 to 0.03 is the range in bias values only, not related to standard deviation.

**Reply: The Vz has been recalculated by using post-processed MRR data. The values of standard deviation are provided in the submitted manuscript. The sentence has been revised to improve the clarity. Please see Lines 206-216 in the revised manuscript.**

L199: Clarify that this is the bias and standard deviation for all site results combined.

**Reply: The bias and standard deviation is all site results combined. The sentence has been revised to improve the clarity. Please see Lines 212-213 in the revised manuscript.**

L200-201: It would be appropriate to acknowledge this limitation earlier in the paper where the method is introduced.

**Reply: More discussion has been added in the revised manuscript. Please see Lines 217-221.**

L203: Usually, "mixed-phase".

**Reply: The typo has been corrected. Please see Line 219.**

L203-204: Again, there is a vague reference to "measurement issues", but there has been no descriptive discussion or quantification of them.

**Reply: The discussion has been revised. Please see Lines 214-221.**

L206-207: This kind of filtering (omitting data from further analysis simply because the data don't give results that match other observations) tends to negate or reduce the believability of the proposed method. This is especially true when the authors cannot point to specific physical conditions that caused the method to fail. How much data was filtered at this stage? How poor are the subsequent results if the data are not filtered?

**Reply: The $V_Z$ criteria for removing inadequate bulk density retrieval is no longer needed. As per the reviewer's suggestion, the retrieval results have been greatly improved after applying the post-processed MRR data (Maahn and Kollias 2012). The noisy bulk density due to low SNR has been removed. The post-processed MRR data has been greatly improved its sensitivity. The Vz difference is mainly for identifying the attenuation effect of MRR reflectivity. Please see Lines 215-216 of the revised manuscript.**

Liquid-equivalent snowfall rate (SR) =====================================

L215-216: Snow gauges like the Pluvio can have problems with undercatch when surface winds are strong. Were the winds checked and any filtering or corrections applied? The bias in the density-derived SR versus the Pluvio SR might be worse if the Pluvio data are corrected for undercatch.

**Reply: All of the Pluvios were equipped with double windshields. The Pluvio at the MHS was within the DFIR (double fence intercomparison reference) in addition to the double**

**shield. The environmental condition of all sites are introduced in the revised manuscript, please see Lines 99-102.**

L216-219: This is the first mention of snow/ice accumulation on the MRR antenna. It would be appropriate to mention that this occurred during the description of the observations earlier in the paper.

**Reply: The discussion of the attenuation effect of MRR due to snow/ice accumulation on the antenna has been added in the revised manuscript. Please see Lines 213-216.**

L224: Should be "moist air".

**Reply: The typo has been corrected. Please see Line 238.**

L227-228: For the case study of the 28 February event, why is only the MHS site data analyzed?

**Reply: The 28 February 2018 event is selected to demonstrate the retrieval results. The consistency of SR calculated from retrieved bulk density and measurement from Pluvio indicate that the proposed algorithm performs reasonably well. The pronounced attenuation effect of MRR reflectivity and its impact to underestimate Vz are shown to demonstrate the retrieval uncertainty. The other sites show almost exactly the same evolution of the retrieved properties. In addition, the PIP was deployed at MHS and collocated with MRR and Parsivel. The comparison of retrieved bulk density from our method and PIP is discussed in the revised manuscript. To keep the manuscript concise, only MHS stie of 28 February is shown. More detailed analysis of each site of 7 March 2018 is discussed.**

Case study: 28 February 2018 ===============================

L258: Regarding "fall velocity was more significant than 1 m s^-1", I suggest rewording this to avoid confusion with statistical signficance.

**Reply: The sentence has been revised to improve the clarity. Please see Line 271.**

L263: Regarding "derivation density", do you mean "derived density"?

**Reply: The typo has been corrected. Please see Lines 276.**

L263-264: Are you describing the *maximum* particle sizes?

**Reply: The typo has been corrected. Please see Lines 276.**

Case study: 7 March 2018 ==========================

L268-269: "produced prominent precipitation" and "produced intensive precipitation" sounds like repetition, are both needed?

**Reply: The "produced prominent precipitation" has been removed. Please see Lines 280-281.**

L273-310: There are a number of locations on these lines that describe bulk water fraction. See my major comments above - I don't think the capability of the retrieval to distinguish and quantify bulk water fraction (or volume fraction of liquid water) has been demonstrated.

**Reply: The bulk water fraction is derived along with the maximum possible bulk density in the proposed method in this study. If different assumption is made in the selection of possible bulk density, the retrieved bulk water fraction will be different. Therefore, the performance of the retrieved bulk water fraction is linked with bulk density retrieval. Since there are no direct measurements of bulk water fraction, we will compare the retrieved bulk density from the proposed method and PIP. The consistency between retrieved bulk density from two algorithm confirms that the retrieved bulk water should be reasonable. Please see Figure R1.**

**As shown in Fig. R1(a), the retrieved bulk density values from the proposed algorithm and PIP gradually decrease from nearly 1.0 to 0.1 (g cm$^{-3}$) between 03 and 06 UTC. Both algorithms capture the transition from the mixing-phase to dry snow. Please see Figure R1 in the previous reply. The manuscript has been revised to include the discussion of the bulk water fraction retrieval. Please see Line 398-417.**

L286: Regarding "which are in accord with the distributions of all velocity-diameter relations", it is not clear to me what this means.

**Reply: The sentence has been revised to improve the clarity. Please see Lines 295-296.**

L288: Regarding "They gradually dissipated", it is not clear what "They" is referring to.

**Reply: "They" refers "the precipitation". The sentence has been revised to improve the clarity. Please see Lines 299.**

L293-294: Regarding "Hence, it implies more ... confirm the distribution of fall velocity and diameter". The meaning here is not clear to me.

**Reply: The sentence has been revised to improve the clarity. Please see Lines 302-303.**

L296: Regarding "confirmed by the alike contrast", the meaning of "alike contrast" is not clear.

**Reply: The sentence has been revised to improve the clarity. Please see Lines 306-308.**

L298: Not true, YPO, MHS and CPO, BKC show mostly near-zero bulk water fraction. For most of this discussion, need to be clear about when only-elevated, only-coastal, or all sites are being described.

**Reply: Only BKC and GWU feature high bulk water fractions. The sentence has been revised to improve the clarity. Please see Line 309.**

L300: "Transited" should be "transitioned".

**Reply: The type has been corrected. Please see Line 312.**

Statistical analysis of bulk density and bulk water fraction

================================================================

L317-320: What is the basis of the assertion that Brandes et al. (2007) observations were dominated by "almost spherical aggregates"? Brandes et al. appear to have used the equivalent volume diameter as determined by the 2DVD software, as particle sizes. These, will be different than the particle size determined by the Parsivel. Brandes et al. do use the median volume diameter to parameterize the bulk density; however it is not evident that the cases in this study and those of Brandes et al. involved similar meteorological conditions. Evidence should be presented for this claim.

**Reply: The statement "dominated by almost spherical aggregates" can be found in the abstract of Brandes et al. (2007). Considering distinct environmental conditions, instrumentations, and retrieval techniques, most of the particles in this study are consistent with the $\rho_{bulk} - D_m$ relation from Heymsfield et al. (2004) and $\rho_{bulk} - D_0$ from Brandes et al. (2007). These results indicate that the proposed bulk density estimation algorithm can derive reasonable retrievals with statistically consistent microphysical characteristics from previous studies. The manuscript has been revised to improve the clarity. Please see Lines 324-338.**

L321-325: The particle sizes used in Heymsfield et al. (2004) are derived from aircraft particle probes, as you have noted. These particle sizes are probably more like the "maximum dimension" of the particle and less like the "equivalent diameter" determined by a Parsivel. Additionally, Heymsfield et al. relate density to mass mean diameter, not to median volume

diameter. So the comparison described here is somewhat an "apples to oranges" comparison. It is not surprising there are differences.

**Reply: Since the two papers are using different parameter (Dm and D₀) to present mean size, we convert $D_0$ to Dm by assuming exponential PSD ($Dm = 4D_0/3.67$). Moreover, the bulk density comparison among this study, Heymsfield et al. (2004), and Brandes et al. (2007) does not intend to emphasize the difference. The discussion has been rephrased. The Figure has been revised. Please see Figure 13 and Lines 324-338.**

L335-344: As I noted above, I am not convince that this method is capable of accurately distinguishing and quantifying the liquid and ice volume ratios and the corresponding bulk water fraction. Also, although it is asserted that there are differences in the meteorology of the warm-low and cold-low events (i.e., "warmer and moister environments" for the warm-low events), no meteorological data is provided to support this.

**Reply: As per the previous comments and replies, we have compared the retrieved bulk density from the proposed algorithm and PIP retrieval (see below figures) and the SR with Pluvios. The bulk water fraction is derived along with the maximum possible bulk density in the proposed method in this study. If different assumption is made in the selection of possible bulk density, the retrieved bulk water fraction and bulk density will be different. Therefore, the performance of the retrieved bulk water fraction is linked with bulk density retrieval. Since there are no direct measurements of bulk water fraction, we compared the retrieved bulk density from the proposed method and PIP. The consistency between retrieved bulk density from two algorithm confirms that the retrieved bulk water should be reasonable.**

**As shown in Fig. R1(a), the retrieved bulk density values from the proposed algorithm and PIP gradually decrease from nearly 1.0 to 0.1 (g cm⁻³) between 03 and 06 UTC. Both algorithms capture the transition from the mixing-phase to dry snow. Please see Figure R1 in the previous reply. The manuscript has been revised to include the discussion of the bulk water fraction retrieval. Please see Line 398-417.**

**As the reply to the previous comment. The temperature (⁰C) and water vapor pressure (hPa) measurements from nearby mountain and coastal AWS sites are collected and summarized in Fig. 14e. The warm-low events have warmer and moister conditions compared to cold-low events. The warm- and cold-low events in the coastal area have similar mean temperature values. On the other hand, the water vapor pressure increases significantly from cold-low to warm-low events. The mountain area has similar features but with higher temperature increments and fewer increments of water vapor pressure. Please see Lines 351-357.**

L345: It is probably more appropriate to say that the density of snow varies with "imposed weather conditions".

**Reply: The sentence has been revised to improve the clarity. Please see Lines 419.**

Conclusions \*\*\*\*\*\*\*\*\*\*\*

L347-350: As I've noted, I have concerns about the bulk water fraction estimates. I don't believe sufficient proof of the capability has been provided, and in no way has evidence been provided that the values are "precise". The high sensitivity of $Z_{HH}$ to the liquid portion of the particle led to precise bulk water fraction estimation. It implied better capability of the density variation due to bulk water fraction change (ex. melting) in the proposed method in this study.

**Reply: Thanks reviewer's suggestion. As per the previous comments and replies, we have compared the retrieved bulk density from the proposed algorithm and PIP retrieval (see below figures) and the SR with Pluvios. The bulk water fraction is derived along with the maximum possible bulk density in the proposed method in this study. If different assumption is made in the selection of possible bulk density, the retrieved bulk water fraction will be different. Therefore, the performance of the retrieved bulk water fraction is linked with bulk density retrieval. Since there are no direct measurements of bulk water fraction, we compared the retrieved bulk density from the proposed method and PIP. The consistency between retrieved bulk density from two algorithm confirms that the retrieved bulk water should be reasonable.**

**As shown in Fig. R1(a), the retrieved bulk density values from the proposed algorithm and PIP gradually decrease from nearly 1.0 to 0.1 (g cm$^{-3}$) between 03 and 06 UTC. Both algorithms capture the transition from the mixing-phase to dry snow. Please see Figure R1 in the previous reply. The manuscript has been revised to include the discussion of the bulk water fraction retrieval. Please see Line 398-417.**

L352: Clarify what is meant by "self-evaluation".

**Reply: The sentence has been revised. Please see Lines 424-425.**

L357: There's no evidence shown that applying the Vz criteria improves the consistency of retrieved SR with observed SR.

**Reply: The Vz criteria is no longer needed. Please see the revised manuscript, Lines 424-427.**

L359: Is "all available cases" true? SR comparison are shown only for two cases at the sites.

**Reply: The SR comparisons from the cases listed in Table 1 are summarized in Table 3. Please see the revised manuscript, Lines 428-430.**

L364-365: I don't think this statement is supported. This study has investigated two cases which have similar synoptic setups and has found similarity of microphysical characteristics. But you haven't demonstrated that different synoptic setups will produce microphysical characteristics dissimilar to these.

**Reply: The statement has been revised to avoid confusing and improve clarity. Please see Lines 431-435.**

L366: I would suggest "contrasting" or "dissimilar" rather than "contrastive".

**Reply: The typo has been corrected. Please see Line 436.**

Tables and Figures ******************

Table 3: Note previous comment about "mean bias". Also, why is the Vz criterion for "ALL" shown as "nan"? To help us understand the significance of the biases and standard deviations, please also include the associated mean values and standard deviations of the observed quantities.

**Reply: The "mean bias" has been revised to "bias". The Vz criterion is no longer needed. The mean values of Vz and SR are provided. Please see the Table 3 in revised manuscript.**

Figure 6: Is the colorbar axis labeled correctly? Were there really counts ranging up to 10**50?

**Reply: The typo has been corrected. It should be 10**5. Please see Fig. 6 in the revised manuscript. Fig. 12 also has the same mistake, and also been corrected.**

Figure 12: Why does the mountainous MHS site maintain a population of high-fall-velocity small particles throughout the 7-8 March event?

**Reply: As indicated by the study from Battaglia et al. (2010), Parsivel's fall velocity measurement may not be accurate for a snowflake particle. This is due to the internally assumed relationship between horizontal and vertical snow particle dimensions. Friedrich et al. (2016) indicate that Parsivel can suffer from splashing of particles (observed as a small diameter with large fall velocity when particles fall on the head of the sensor) and margin fallers (observed as a faster velocity than true fall velocity when particles fall through the edge of the sampling area). Yuter et al. (2006), Aikins et al. (2016), and Kim et al. (2021) indicate the splashing and border effects of the diameter of**

**< 1 mm in Parsivel fall velocity measurements. The Parsivel data shown in Figure 12 was quality-controlled, as suggested by Lee et al. (2015). The discussion of fall velocity measurement uncertainty is added in the revised manuscript. Please see Lines 105-109.**

---

## Referee Report (RR1)

Thanks to the authors for their additions and responses to my original comments. The authors have

- performed substantial investigations of the effects of uncertainties,

- applied the Maahn and Kollias post-processing to the MRR data, then recalculated retrieval results,

- removed the filtering based on reflectivity-weighted velocity,

- and added a comparison of retrieved bulk density against PIP estimates of bulk density, finding good agreement.

These modifications address most of the issues from my previous review.

I do still have a substantial concern about the method used to decompose the bulk density into an ice volume fraction ("bulk ice fraction", $vi$) and a liquid volume fraction ("bulk water fraction", $vw$). The authors do not provide a physical basis for the approach they use. Instead, they use what seems to be an ad-hoc requirement to obtain the smallest possible bulk water fraction given the $Z_{HH}$ and the retrieved $\rho_{bulk}$.

It is clear from equation (2) that $Z_{HH}$ is a function of $\rho_{bulk}$, so I think the part of the study related to determining $\rho_{bulk}$ is reasonable. But it is also clear from equation (2) that $Z_{HH}$ provides no information that could be used to distinguish bulk ice and water fractions. From equation (1), the best that can be obtained is a linear relationship between $vi$ and $vw$. An error in $vw$ could be compensated by an offsetting error in $vi$ to give an accurate $\rho_{bulk}$. Thus an accurately-retrieved $\rho_{bulk}$ does not indicate or imply that an estimated $vw$ is correct.

This concern could be addressed if the authors can provide a rational justification for the approach they have taken. Perhaps there are reasons that they have decided to select the solutions that provide minimum $vw$. Why is it desirable to choose the minumum $vw$ solution? Note my comment below for lines 164-171 of the revised article. If they have sound reasons and can elaborate on those in the methodology, that would address this concern.

If the authors are unable to do this, I think the proper approach would be for the authors to de-emphasize their claim of "retrieving" bulk water fraction and instead state that their analyses are for one possible approach to selecting the bulk water fraction.

**Line-by-line comments on new revision**

**L** 28: Do you specifically mean "liquid water content" here? Do you instead mean just "water content," since remote sensors observe both ice and liquid hydrometeors.

**L** 38-39: Since density is not related to the "aerodynamic process," maybe rewrite this as "... induces higher density as well as higher fall velocity by the aerodynamic process."

**L** 107: The word "minute" by itself in English is ofter used to mean "small" or "tiny." I suggest using "one-minute" instead, which has the desired meaning of "a sample of length one minute of time," here and at other locations in the paper.

**L** 111: Is the "ICEP-POP" that is used here intentional, rather than "ICE-POP"?

**L** 144-147: If each hydrometeor is truly "regarded as a symmetric sphere" and ice and water are assumed to be evenly distributed within the particle, canting angle is not relevant - the scattering properties will not change with respect to any rotation of the sphere. Why are canting angles considered? Were the particles not actually symmetric spheres? Please enhance this description to be clear and correct about what is being assumed for the calculation of the scattering properties.

**L** 164-171: This description of the methodology is the point of my most significant concern with the study. To obtain distinct $vi$ and $vw$, the authors make an ad-hoc choice to pick the solution with the maximum $\rho_{bulk}$ and $vi$. The justification they provide is that this is "similar to Huang et al. (2010), which assumes" the particles are only ice and air. My opinion is that this justification is not sufficient to allow a claim that $vw$ is being retrieved.

**L** 172-173 and 188: Per the statements by the authors here, the validation approaches that are being used are to validate the retrieved bulk density, not the bulk water fraction.

**L** 184-185: Reflectivity-weighted velocity (for comparison to radar Doppler velocity) is more often seen calculated from PSDs as
$$V_Z^{\rho_{bulk}} = \frac{\sum_i \sigma_{bk}(D_i)V(D_i)N(D_i)\Delta D_i}{\sum_i \sigma_{bk}(D_i)N(D_i)\Delta D_i}$$
where $\sigma_{bk}(D_i)$ is the backscatter cross-section for particles in size bin $i$. It's not clear here what is meant by $Z(\rho_{bulk}, D)$. Please add some description of $Z(\rho_{bulk}, D)$, how it is calculated and whether your formula gives results that are the same as this more typical formula. If not the same, the comparisons of $V_Z^{\rho_{bulk}}$ and $V_Z^{MRR}$ may be of concern.

**L** 196: Per the reference, Kim et al. (2021), equation (6) gives the volume-weighted mean diameter, not the mass-weighted mean diameter. The Kim et al. statement seems correct, since equation (6) gives the ratio of the fourth moment of the PSD to its third moment.

**L** 211: I think this should be "... and CPO are slightly lower."

**L** 220: Regarding "various measurement issues" that induce inconsistency, please be more explicit by stating what are these issues.

**L** 264: Please check Figure 5. I do not see a gray area.

**L** 291-292 and 295-296: I think these lines overstate the interpretation of Figure 12 somewhat. I agree that the distributions in Figure 12 do show changes in fall velocity-diameter relationships. It is probably OK to say that the particular changes in the relationships are consistent with increases in $\rho_{bulk}$ which could be associated with increases in bulk water fraction resulting, for example, from melting of particles. But I believe it is an overstatement to say that "gradual increases in density, as well as the bulk water fraction" can be *found* in the $V(D)$ distributions in Figure 12 (L291-292) or that the retrieved bulk density and bulk water fraction *reveal* distinct $V(D)$ relations (L295-296).

**L** 303: I'd suggest "transitioned" rather than "transited".

**L** 304: Should this be "at other sites" instead of "as other sites"?

**L** 304-305: See my earlier comment regarding L 291-292. Saying that the $V(D)$ relation "is consistent with" the bulk water fraction seems more appropriate.

**L** 326: See my earlier comment regarding L 196 and what is actually calculated by equation (6).

**L** 334: Where does this relationship between $D_m$ and $D_0$ come from? Is there a reference?

**L** 359: See my opening comments and concerns along with the related line-by-line comments regarding the ability of the retrieval to determine bulk water fraction. This statement also falls under that concern and should be addressed.

**L** 366: While Battaglia et al. do discuss Parsivel fallspeed errors, I don't believe they are discussed in Wood et al.

**L** 376: Especially since this approach of using CF is from personal communications and not from a published reference, the values of the particle size-dependent CF and the method by which its values are determined should be documented here, to allow the results to be reproduced.

**L** 410-411: No, I don't think it is justified to say that since the bulk densities are in agreement with those from the PIP, the bulk water fractions are confirmed. $Z_{HH}$ is dependent on bulk density in a way that makes it not possible to discriminate the contributions of $vi$ and $vw$. Since bulk density depends on both $vi$ and $vw$, offsetting errors in $vi$ and $vw$ could still give a correct $\rho_{bulk}$.

**L** 413: Usually "mixed-phase" rather than "mixing-phase".

**L** 416-417: See comment regarding L 410-411.

**L** 433: There appears to be an incomplete sentence here: "The retrieved bulk density."

**L** 441: Again, see and address my overall comments regarding retrieval of bulk water fraction.

**L** 444: Usually "unattended" rather than "unattentively".

**General comments**

Does EGU have a policy on including information about where to obtain the input datasets used for the study presented in the paper? Is a data availability statement required? In the acknowledgements, I note that the source of the PIP data is not mentioned.

---

## Author Response (AR3)

Dear Reviewer,

The authors sincerely appreciate your valuable comments and suggestions to help improve the manuscript. We have revised the manuscript titled "Estimating the Snow Density using Collocated Parsivel and MRR Measurements: A Preliminary Study from ICE-POP 2017/2018 ". that was submitted to ACP (Atmospheric Chemistry and Physics) on 3 January, 2024. Based on your suggestions, we have put substantial effort into additional analysis. The manuscript has been thoughtfully revised regarding the comments from all reviewers.

One of the major concerns of the proposed density retrieval algorithm using collocated MRR and Parsivel is lacking the uncertainty analysis. As per the reviewer's suggestion, we have performed substantial investigations of the retrieval uncertainty. The impacts of the measurement uncertainty of the Parsivel and the MRR on the bulk density retrieval are analyzed quantitatively. The measurement issue of Parsivel is also investigated to understand its impact on bulk density retrieval. The results are summarized in the revised manuscript as a Discussion section.

The MRR data quality issue has been examined per the reviewer's suggestion. The post-processed data have replaced the entire MRR raw data by applying the algorithm from Maahn and Kollias (2012). All the bulk density, bulk water fraction, and reflectivity-weighted velocity retrievals have been recalculated. The figures have been revised as well.

The original purpose of utilizing reflectivity-weighted velocity to filter adequate retrieval is no longer needed and has been removed in the revised manuscript. The quality of the retrieval results has been greatly improved by applying the post-processed MRR data per the reviewer's suggestion. The low SNR MRR measurement has been removed. The comparison of reflectivity-weighted velocity is mainly used to identify the inadequate retrieval due to the attenuation effect on MRR reflectivity.

The performance of the retrieved bulk density has been validated by the snowfall rate (SR) from collocated Pluvio measurements and reflectivity-weighted fall velocity (Vz) from MRR. In addition to SR and Vz, the performance of the retrieved bulk density has been compared with the precipitation imaging package (PIP), a video disdrometer (Newman et al., 2009; Pettersen et al., 2020). The PIP was also deployed at the MHS site during ICE-POP 2018 (Tokay et al., 2023). The comparison of retrieved bulk density between the proposed algorithm in this study and PIP has shown good agreement with each other. The high consistency further confirms the performance of the retrieved bulk density. Since there is no direct bulk water fraction measurement for validation, the authors consider the validation of bulk density retrieval to PIP and Pluvio as "indirect" evidence to support the bulk water fraction retrieval.

The SR and Vz validation analysis shows that the algorithm can adequately retrieve the bulk density and bulk water fraction. The consistency of the retrieved bulk density to collocated PIP confirms the performance of the proposed algorithm in this study. The advantage of the proposed algorithm is that it utilizes collocated Parsivel and MRR, which are commercially available, commonly used, and robust instruments. The Parsivel and MRR can operate unattentively and need little maintenance. Further application of the proposed algorithm helps derive long-term observation data on snow properties. The authors believe the proposed algorithm can provide an alternative choice if sophisticated instruments (e.g., 2DVD, PIP, SVI, MASC) are unavailable.

The manuscript has also been revised carefully following the reviewer's suggestions on English wording. The authors would like to express our sincere appreciation for the comments. The added or modified sentences in the revised manual are in red for your convenience.

The point-to-point replies to every comment have been prepared in the following. For your convenience, the reply is arranged as follows,

Reviewer's comments

**Response**

Revisions in the manuscript

We would appreciate any feedback on the revisions.

**General comments**

Do the authors see a possibility that the retrieved high values of bulk density and liquid fraction during periods of low reflectivity and precipitation rate could be, at least partly, an artifact from applying the method on observations of very weak precipitation? The retrieval seems somewhat unstable in these conditions which is not surprising considering factors such as low signal to noise ratio due to weak signal. Yet, the authors draw considerable attention to the retrievals of these periods of very weak precipitation. It is worth considering which disciplines would benefit from the microphysical retrievals of weak mixed phase precipitation (<0.5mm/h)? I would suggest either introducing a threshold for minimum reflectivity or precipitation rate where the method is applied, or otherwise critically reviewing the method's performance in weak precipitation, where the precipitation rate falls below the sensitivity of the Pluvios.

**Reply: The retrieved bulk density and bulk water fraction as a function of density are shown in Figure R1. Most retrieved bulk density is obtained from MRR reflectivity from 10 to 30 dBZ. Some retrieval results with high bulk density and low MRR reflectivity can be found, but not frequent.**

**The quality of the retrieval results has been greatly improved by applying the post-processed MRR data as per the reviewer's suggestion. The low SNR MRR measurement has been removed. The authors intend to preserve as much data as possible by only eliminating the data with an attenuation effect on MRR reflectivity. The attenuated MRR reflectivity underestimates the retrieved bulk density and bulk water fraction.**

[Figure]

**Figure R1: (a) The retrieved bulk density as a function of MRR reflectivity. (b) Same as (b), but for bulk water fraction.**

I don't feel that the retrieval of $v_w$ is adequately demonstrated in the case studies since there seem to be no time periods where there would be both a) agreement between derived and measured terminal velocity and b) notable precipitation intensity at the same time. It raises the concern how the method would perform in mixed phase situations with larger particles or higher precipitation rates. In my mind, the best way to address this would be to replace one of the case studies with another one that has significant intensity of wet snow, or discuss the possible limitations of the method's application.

**Reply: The bulk water fraction is derived along with the maximum possible bulk density using the proposed method in this study. If a different assumption is made when selecting possible bulk density, the retrieved bulk water fraction will be different. Therefore, the performance of the retrieved bulk water fraction is linked with bulk density retrieval. Since there is no direct measurement of bulk water fraction, we compare the retrieved bulk density from the proposed method and PIP. The consistency between retrieved bulk density from the two algorithms strengthens the reliability of the retrieved bulk water, which should be reasonable.**

As shown in Fig. R2(a), the retrieved bulk density values from the proposed algorithm and PIP gradually decrease from nearly 1.0 to 0.1 (g cm$^{-3}$) between 03 and 06 UTC. Both algorithms capture the transition from the mixing phase to dry snow. Please see Figure R2. The manuscript has been revised to include a bulk water fraction retrieval discussion.

[revised manuscript text omitted]

In the presented case studies, it seems like non-zero $v_w$ values only occur when bulk density is nearly saturated at over 0.9 g/cm. I'm concerned whether this is physically reasonable especially given the assumption of spherical particles. This raises the question whether, effectively, the liquid fraction would act as a kind of an overflow buffer in the calculations when density alone cannot explain the reflectivity values. This could rise from the assumption of maximum ice volume fraction ($v_i$). Or could it be explained with the drizzle-like nature of the precipitation during the $v_w$ signals? This concern could be dispelled with a counter example.

**Reply: The assumption of the particle shape has been discussed in the revised manuscript. Please see Lines 386-397 in the revised manuscript or the following.**

[revised manuscript text omitted]

The manuscript lacks discussion on the implications of assuming spherical particles. This might be significant consideration given the wide range of different particle habits and their shapes and preferred falling alignments.

**Reply: The discussion of the assuming spherical particles has been included in the revised manuscript. Please see Lines 386-397. Or the reply of the previous question.**

**The particle shape assumption for falling velocity and reflectivity calculation has been discussed. Please see Lines 218-221 in the revised manuscript. Or see the following.**

The various shapes aerodynamically complicate the falling behaviors of ice-phase and mixed-phase particles (Mitchell and Heymsfield 2005; Heymsfield and Westbrook 2010). Moreover, various measurement issues of MRR and Parsivel also induce some inconsistency. Nevertheless, the overall consistency of the $V_Z^{MRR}$ and $V_Z^{\rho_{bulk}}$ suggests that the retrieved bulk density is an adequately reasonable value.

Since the manuscript considers the liquid fraction, it would be worth showing or at least mentioning if melting layer signals were detected in the MRR observations.

**Reply: Most cases in this study have surface temperatures near zero or lower than zero degrees. As shown in Figures 5 and 7-11, there is no pronounced bright-band signature from MRR radar data.**

The viewpoint in the manuscript is more technical and focuses less on microphysics. As such, the topic is suitable for ACP but might be even better suited for AMT. This is a possible consideration for resubmission after revisions.

**Reply: The manuscript contains the retrieval technique description and the retrieval result analysis. The retrieval technique is demonstrated by substantial bulk density analysis and water fraction evolution. The features of these bulk properties of warm-low and cold-low events are also investigated.**

**Specific comments**

The title refers to snow density, but retrievals are attempted for snow, mixed phase precipitation and light drizzle. If liquid fraction plays an important role in the revised manuscript, it would be good to be reflected in the title.

**Reply: The majority of the particles in this study are snow. The algorithm estimates bulk water fraction value, and the mixing phase of snow can be subsequently differentiated from dry snow. As the bulk water fraction equals one, thus the particle is considered a drizzle or raindrop. Hence, the main objective is to estimate the snow bulk density, including dry and wet snow.**

The title indicates this is a preliminary study. Are the authors working on a more comprehensive analysis? Worth mentioning in the discussion.

**Reply: The proposed method has been introduced and applied to ICE-POP 2018 data. The analysis also shows that the algorithm can adequately retrieve the bulk density and bulk water fraction. The advantage of the proposed algorithm is that it utilizes collocated Parsivel and MRR, which are commonly used and robust instruments. The Parsivel and MRR can operate unattentively and need little maintenance. Further application of the proposed algorithm helps derive long-term observation data on snow properties. The discussion has been added to the revised manuscript. Please see Lines 441-447 in the revised manuscript. Or see the following.**

The SR and Vz validation analysis shows that the algorithm can adequately retrieve the bulk density and bulk water fraction. The consistency of the retrieved bulk density to collocated PIP confirms the performance of the proposed algorithm in this study. The advantage of the proposed algorithm is that it utilizes collocated Parsivel and MRR, which are commercially available, commonly used, and robust instruments. The Parsivel and MRR can operate unattentively and need little maintenance. The proposed algorithm provides an alternative choice if a sophisticated instrument (e.g., 2DVD, PIP, SVI, MASC, etc.) is unavailable. Further application of the proposed algorithm helps derive long-term observation data on snow properties.

L42: This sentence seems to suggest that riming and melting are the only processes affecting snow density. While these are important, one should not forget that, e.g., the primary particle habit and aggregation have great impact, too.

**Reply: The aggregation process has been added to the revised manuscript. Please see Lines 31-32 in the revised manuscript. Or see the following.**

The snow density caused by various degrees of riming, melting, and aggregation processes is essential to derive the Ze-SR relation (Huang et al. 2014).

L59: As one of the selling points of the new density retrieval method is the use of robust intsruments that require little maintenance, perhaps it would be good to mention studies that use, e.g., PIP or Parsivel for density retrievals. Such studies could be found, for example, by doing citation analysis on Brandes et al. (2007) and Huang et al. (2010), as referred to in the manuscript.

**Reply: Per the reviewer's suggestion, the PIP (precipitation imaging package, Newman et al., 2009; Pettersen et al., 2020) was also deployed at the MHS site during ICE-POP 2018. Tokay et al. (2023) have utilized PIP to investigate the PSD parameters, including mass-weighted diameter and normalized intercept. The bulk density is estimated by Tokay et al. (2023) with various assumptions. The PIP retrieved density was generated from the assumption that $D_{max} = 1.15 \, D_{eq}$, and the mass derivation included was based on Bohm (1989). The time series of retrieved bulk density from the proposed algorithm and PIP are shown in Fig. R2. The consistency between retrieved bulk density from the two algorithms confirms that the retrieved bulk water should be reasonable. Please see Lines 398-413 in the revised manuscript. Or see the following.**

5.3 The bulk density comparison with collocated PIP

   The precipitation imaging package (PIP), a video disdrometer, provides the PSD, fall speed, density, and snowfall rate of hydrometers (Newman et al., 2009; Pettersen et al., 2020) was also deployed at the MHS site during ICE-POP 2018. Tokay et al. (2023) have utilized PIP to investigate the PSD parameters, including mass-weighted diameter and normalized intercept. The bulk density is estimated by Tokay et al. (2023) with various assumptions. The PIP retrieved density was generated from the assumption that $D_{max} = 1.15 \, D_{eq}$, and the mass derivation included was based on Bohm (1989). As shown in Fig. 15, the retrieved bulk density from the proposed algorithm in this study and PIP have high consistency. Both retrieved bulk densities are highly correlated to each other, except for the period of 08 to 15UTC on 28 February due to the attenuation effect of the accumulated snow on the MRR antenna (Fig. 5e).

      The bulk water fraction is derived along with the maximum possible bulk density using the proposed method in this study. The retrieved bulk water fraction will differ if a different assumption is made when selecting possible bulk density. Therefore, the performance of the retrieved bulk water fraction is linked with bulk density retrieval. Since there are no direct measurements of bulk water fraction, the consistency between retrieved bulk density from two

algorithms "indirectly" confirms that the retrieved bulk water should be reasonable. As shown in Fig. 15a, the retrieved bulk density values from the proposed algorithm and PIP gradually decrease from nearly 1.0 to 0.1 (g cm$^{-3}$) between 03 and 06 UTC. Both algorithms capture the fast transition from the mixing-phase to dry snow.

**In addition, the MASC (multi-angle snowflake camera) is also introduced in the revised manuscript. Please see Lines 59-62 in the revised manuscript. Or see the following.**

Other sophisticated instrumentations are developed to investigate the microphysical characteristics of snow particles. The precipitation imaging package (PIP), a video disdrometer, provides the PSD, fall speed, density, and snowfall rate of hydrometers (Newman et al., 2009; Pettersen et al., 2020). The Multi-Angle Snowflake Camera (MASC) captures high-resolution photographs of hydrometeors from three angles while simultaneously measuring their fall speed (Garrett et al. 2012).

Section 2: Details about the ambient temperature measurements are missing, in particular, the types of the instruments or sensors used. The temperature measurements, while not part of the main retrieval methods presented, should be of great interest for the reader as an indication for melting and other microphysical processes.

**Reply: The temperature measurements were derived from collocated AWS (Vaisala WXT520). The comparison between the temperature measurements from AWS and PARSIVEL has shown that WXT520 has better performance. The Parsivel data had a significant bias in the Parsivel temperature data. The information on the temperature sensor has been added to the revised manuscript. Please see Fig. 5, 7-11 in the revised manuscript. Or see the following.**

The temperature (°C, red line) from nearby AWS (Vaisala WXT520) and $Z_{HH}$ from the third layer of MRR.

Section 2: I would like to see basic information about the Pluvios used such as make, orifice size and shielding.

**Reply: The Pluvios are OTT Pluvio² - Weighing Rain Gauge. All of the Pluvios were equipped with double windshields. The Pluvio at the MHS was within the DFIR (double fence intercomparison reference) in addition to the double shield. All the sites investigated in this study have no taller trees or buildings near the MRR antenna and Parsivel. The environmental conditions of all sites are introduced in the revised manuscript. Please see Lines 92-103 in the revised manuscript. Or see the following.**

The data of MRR, Parsivel, and Pluvio (OTT Pluvio² - Weighing Rain Gauge) were collected during the ICE-POP 2018 (2017/2018 winter) and the pre-ICE-POP campaign (2016/2017 winter). The instruments were located in nineteen sites across the Gangwon region on the east coast of Korea (see Kim et al. 2021 for detailed information of each site). Five sites with collocated MRR and Parsivel were available for this study. These sites aligned across the Taebaek Mountains from mountain to coast are YPO (YongPyong Observatory, 772 m MSL), MHS (MayHills Supersite, 789 m MSL), CPO (Cloud Physics Observatory, 855 m MSL), BKC (BoKwang 1-ri Community Center, 175 m MSL), and GWU (Gangneung-Wonju National University, 36 m MSL), respectively. The YPO, MHS, and CPO sites are in the mountainous region, while GWU and BKC sites are in the coastal area (Kim et al. 2021). All of the Pluvios were equipped with double windshields. The Pluvio at the MHS was within the DFIR (double fence intercomparison reference) in addition to the double shield. All the sites investigated in this study have no taller trees or buildings near the MRR antenna and Parsivel. Each site's detailed layout and information can be found in Kim et al. (2021).

L130: What is meant by a canting angle when referring to spherical particles?

**Reply: The discussion of non-spherical particles has been introduced in the revised manuscript. Non-spherical and spherical particle measurements are different when MRR looks upward. However, the orientation of the non-spherical particle is assumed to be isotropic and homogeneous. A sensitivity investigation assuming the particle axis ratio of 0.5 has been conducted. The results show that about 1.5 dBZ variation of simulated reflectivity can be induced due to the assumption of particle size.**

**A random error of MRR reflectivity with a standard deviation of 1.2 dB is introduced into the retrieval algorithm to imitate the particle assumption and MRR measurement uncertainty. The overall standard deviation of bulk density retrieval uncertainty is about 0.025 (g cm$^{-3}$) for a given MRR reflectivity uncertainty of 1.2 dB. The bulk water fraction retrieval has the same feature, and the uncertainty is about 0.041. Please see the Discussion section in the revised manuscript. Or pages 6-8 of this reply.**

L133: Why was this temperature range chosen for the simulations? Is it physically reasonable to simulate mixed-phase precipitation in -10 degrees Celsius, for example?

**Reply: The temperature is set to $0^0$C in the T-matrix simulation. The sensitivity test of temperature is shown in Fig. 1. The results indicate that the reflectivity simulation is insensitive to particle temperature. Please see Lines 150-155 in the revised manuscript. Or see the following.**

An example of simulated $Z_{HH}$ from Parsivel observed snow PSD via T-matrix simulation with different combination of vi/vw and temperature is shown in Fig. 1. The results indicate that the simulated $Z_{HH}$ values remain nearly identical when varying the temperature from -10 to 0 $^0$C. On the other hand, the simulated $Z_{HH}$ varies significantly when altering the composition of vi/vw. The lowest (highest) value of $Z_{HH}$ was from the combination of vi/vw of 1.0/0.0 (0.0/1.0), which was pure ice (rain) with a density of 0.92 (1.0) g cm$^{-3}$. The particle temperature was consequently assumed to have a constant value of 0℃ in the following $Z_{HH}$ T-matrix simulation.

L147: I failed to find references to liquid water fraction from Huang et al. (2010). They seem to just assume particles to consist of ice and air. Either I missed it, or this reference could be more accurate.

**Reply: Huang et al. (2010) assumed that a mixture of snow contains only ice and air. Please see page 642 of Huang et al. (2010), "To calculate the backscattering properties of the particles measured by the 2DVD, we consider snow to be a mixture of ice and air.".**

   **The manuscript has been revised to improve the clarity. Please see Lines 167-168 in the revised manuscript. Or see the following.**

   This assumption is similar to Huang et al. (2010), which assumes that a mixture of snow contains only ice and air. The water fraction is not considered in Huang et al. (2010).

L263: "The particle size was" to 'Maximum particle size ranged from'

**Reply: The sentence has been revised as per the reviewer's suggestion. Please see Lines 276-277.**

L279: Since there are many sites involved in this study, it could be useful to have a small map showing their locations.

**Reply: This study is part of a series of studies of ICE-POP 2018. Many published papers contain such information. The authors would like to use references to provide such information to keep the manuscript concise. "The instruments were located in nineteen sites across the Gangwon region on the east coast of Korea (see Kim et al. 2021 for detailed information of each site)." Please Lines 93-94.**

L315: "number density function" to 'number concentration'

**Reply: The sentence has been revised as per the reviewer's suggestion. Please see Lines 325-326.**

L331: How is mean bulk density calculated here? Is it integrated over the total volume?

**Reply: The mean bulk density is replaced by the median value of bulk density. The median value is obtained for each site. Please see Line 341.**

L334: Since rain was not excluded from the retrievals, we are now talking about the mean bulk density of mixed precipitation instead of snow. As the densities of rain and snow are quite different, the mean value is easily driven by the fraction of rain, masking the possible signal from snow properties.

**Reply: The number concentration of the retrieved bulk density is shown in Fig. 14. Both the numbers of the dry snow and mixing-phase events are shown in both warm-low and cold-low events. The median values of the retrieved bulk density of each site are shown in Fig. 14a,b. The temperature ($^0$C) and water vapor pressure (hPa) measurements from nearby mountain and coastal AWS sites are collected and summarized in Fig. 14e. The warm-low events have warmer and moister conditions compared to cold-low events. The warm- and cold-low events in the coastal area have similar mean temperature values. On the other hand, the water vapor pressure increases significantly from cold-low to warm-low events. The mountain area has similar features but with higher temperature increments and fewer increments of water vapor pressure. Please see Lines 339-357.**

L345: This sentence seems to suggest that the density of snow has an impact on the weather. It's unclear to me what was meant here. Please rephrase to clarify.

**Reply: The sentence has been revised. Please see Lines 419.**

L526: "The ZHH variation with vw is much less than that with vi". I think, the opposite is true.

**Reply: It should be "On the contours of $Z_{HH}$, the $Z_{HH}$ variation with vw is much less than that with vi." The sentence has been rephrased; please see Lines 628.**

L560: What does the "shaded area" refer to in Fig. 5c?

**Reply: The "shaded area" refers to the MRR reflectivity profile. The sentence has been revised to improve the clarity. "The time series of MRR $Z_{HH}$ vertical profile (dBZ) from the third gate (0.45 km) to the 5 km." Please see Fig. 5 and 7-11.**

Figures 5d, 9d-11d: There seem to be flat parts in the temperature measurements, where the measured value does not seem to change even for a fraction of a degree. These look like a measurement errors, perhaps gaps in the measurements. This raises concerns about the quality

of the temperature measurements. The measurements should be checked and erroneous values omitted from the analysis. If there is a cause for concern about the quality of the measurements, it should be discussed in the manuscript.

**Reply: The temperature measurements were derived from collocated AWS (Vaisala WXT520). The comparison between the temperature measurements from AWS and Parsivel has shown that WXT520 has better performance. The Parsivel data had a significant bias in the Parsivel temperature data. The information on the temperature sensor has been added to the revised manuscript. The scale range of the temperature in Fig. 5, 7-11 is also improved for clarity.**

Figure 6: The integration times in these figures are quite long. For example, in Fig. 6b, there seem to be multiple modes in the (D, v) distribution. Because of the long integration time, it is unclear if these modes are co-existing or if the dominating particle type is evolving over time. Have the authors considered analyzing the particle properties such as (D, v) distribution in shorter time intervals?

**Reply: The authors did examine shorter time intervals for (D, v) distribution. The (D, v) distribution showed better differentiation between dry and mixing-phase snow. Both Fig. 6 and 12 can illustrate the transition of the mixing phase to dry snow. Considering the number of the figure and keeping the manuscript concise, we would like to maintain the current format of Fig. 6 and Fig. 12.**

**Technical comments**

L10: Authors should choose between the spellings "disdrometer" and "distrometer". Currently, mixed spelling is used for this word in the manuscript.

**Reply: The "distrometer" has been revised to "disdrometer".**

L27: Since riming refers to a process and not a hydrometeor type, change "riming" to 'rimed particles'.

**Reply: The "riming" has been revised to "rimed particles". Please see Line 27.**

L73-74: The readability of this sentence could be improved by rephrasing.

**Reply: Please see Lines 81-82. The sentence has been revised:** Subsequently, the measurement of $Z_{HH}$ weighted fall velocity ($V_Z$) from MRR is compared with the calculated $V_Z$ from the derived bulk density and Parsivel PSD measurement.

L121: The authors probably meant 'modified FROM Huang et al. (2010)'.

**Reply: The typo has been corrected. Please see Lines 138.**

L142: "as the increasing" to 'with increasing'

**Reply: The sentence has been revised as per the reviewer's suggestion. Please see Line 161.**

L178: "examined with" to 'compared with'

**Reply: The sentence has been revised as per the reviewer's suggestion. Please see Line 193.**

L186: "four with Pluvio" to 'four of them equipped with a Pluvio'

**Reply: The sentence has been revised as per the reviewer's suggestion. Please see Line 200.**

L196: "consistency of" to 'consistent'

**Reply: The sentence has been revised as per the reviewer's suggestion. Please see Line 209.**

L232: Full stop after "shown in Fig. 5"

**Reply: The sentence has been revised as per the reviewer's suggestion. Please see Line 347.**

L267: "relatively weaker" to 'weaker'. There are also other instances of using "relatively" with a comparative in the manuscript. I advice against this.

**Reply: The sentence has been revised as per the reviewer's suggestion. Please see Line 280. Other similar comparatives have been revised as well.**

L274: "precipitation system" to 'precipitation'

**Reply: The sentence has been revised as per the reviewer's suggestion. Please see Line 283.**

L303: "has a more consistent relation" to 'is more consistent with the relation'

**Reply: The sentence has been revised as per the reviewer's suggestion. Please see Line 314.**

L345: "microphysics processes" to 'microphysical processes'

**Reply: The sentence has been revised as per the reviewer's suggestion. Please see Line 419.**

L353: "Consistency" to 'Inconsistency'

**Reply: The sentence has been revised as per the reviewer's suggestion. Please see Line 425.**

L361: The manuscript could be more consistent with how dates are written: with or without ordinal indicators.

**Reply: The sentence has been revised as per the reviewer's suggestion. Please see Line 431.**

L501: "blue lines" to 'blue', as there are also other symbols than lines.

**Reply: The sentence has been revised as per the reviewer's suggestion. Please see Line 603.**

L524: "DSD" to 'PSD'

**Reply: The sentence has been revised as per the reviewer's suggestion. Please see Line 624.**

L569: "derived" to 'shown'

**Reply: The sentence has been revised as per the reviewer's suggestion. Please Line 669.**

Figure 5d: "MRM" to 'MRR' in the y-axis label.

**Reply: The typo has been corrected. Please see Fig. 5d.**

Figure 6c: label (c) missing from figure

**Reply: The typo has been corrected. Please see Fig. 6c.**

Dear Reviewer,

The authors sincerely appreciate your valuable comments and suggestions to help improve the manuscript. We have revised the manuscript titled "Estimating the Snow Density using Collocated Parsivel and MRR Measurements: A Preliminary Study from ICE-POP 2017/2018 ". that was submitted to ACP (Atmospheric Chemistry and Physics) on 3 January, 2024. Based on your suggestions, we have put substantial effort into additional analysis. The manuscript has been thoughtfully revised regarding the comments from all reviewers.

One of the major concerns of the proposed density retrieval algorithm using collocated MRR and Parsivel is lacking the uncertainty analysis. As per the reviewer's suggestion, we have performed substantial investigations of the retrieval uncertainty. The impacts of the measurement uncertainty of the Parsivel and the MRR on the bulk density retrieval are analyzed quantitatively. The measurement issue of Parsivel is also investigated to understand its impact on bulk density retrieval. The results are summarized in the revised manuscript as a Discussion section.

The MRR data quality issue has been examined per the reviewer's suggestion. The post-processed data have replaced the entire MRR raw data by applying the algorithm from Maahn and Kollias (2012). All the bulk density, bulk water fraction, and reflectivity-weighted velocity retrievals have been recalculated. The figures have been revised as well.

The original purpose of utilizing reflectivity-weighted velocity to filter adequate retrieval is no longer needed and has been removed in the revised manuscript. The quality of the retrieval results has been greatly improved by applying the post-processed MRR data per the reviewer's suggestion. The low SNR MRR measurement has been removed. The comparison of reflectivity-weighted velocity is mainly used to identify the inadequate retrieval due to the attenuation effect on MRR reflectivity.

The performance of the retrieved bulk density has been validated by the snowfall rate (SR) from collocated Pluvio measurements and reflectivity-weighted fall velocity (Vz) from MRR. In addition to SR and Vz, the performance of the retrieved bulk density has been compared with the precipitation imaging package (PIP), a video disdrometer (Newman et al., 2009; Pettersen et al., 2020). The PIP was also deployed at the MHS site during ICE-POP 2018 (Tokay et al., 2023). The comparison of retrieved bulk density between the proposed algorithm in this study and PIP has shown good agreement with each other. The high consistency further confirms the performance of the retrieved bulk density. Since there is no direct bulk water fraction measurement for validation, the authors consider the validation of bulk density retrieval to PIP and Pluvio as "indirect" evidence to support the bulk water fraction retrieval.

The SR and Vz validation analysis shows that the algorithm can adequately retrieve the bulk density and bulk water fraction. The consistency of the retrieved bulk density to collocated PIP confirms the performance of the proposed algorithm in this study. The advantage of the proposed algorithm is that it utilizes collocated Parsivel and MRR, which are commercially available, commonly used, and robust instruments. The Parsivel and MRR can operate unattentively and need little maintenance. Further application of the proposed algorithm helps derive long-term observation data on snow properties. The authors believe the proposed algorithm can provide an alternative choice if sophisticated instruments (e.g., 2DVD, PIP, SVI, MASC) are unavailable.

The manuscript has also been revised carefully following the reviewer's suggestions on English wording. The authors would like to express our sincere appreciation for the comments. The added or modified sentences in the revised manual are in red for your convenience.

The point-to-point replies to every comment have been prepared in the following. For your convenience, the reply is arranged as follows,

Reviewer's comments

**Response**

Revisions in the manuscript

We would appreciate any feedback on the revisions.

General Comments ################

1. High sensitivity of $Z_{HH}$ to the liquid portion of the particle allows for precise bulk water fraction estimation.

The study provides no evidence that the bulk water (liquid) fraction produced by the retrieval is consistent with actual bulk liquid water fraction. The claim that the retrieval can estimate bulk liquid water fraction is supported only by a brief argument regarding the differing sensitivity of the retrieval forward model to the water and ice volume fractions.

**Reply: The bulk water fraction is derived along with the maximum possible bulk density in the proposed method in this study. If a different assumption is made when selecting possible bulk density, the retrieved bulk water fraction will be different. Therefore, the**

performance of the retrieved bulk water fraction is linked with bulk density retrieval. Since there is no direct measurement of bulk water fraction, we compare the retrieved bulk density from the proposed method and PIP. The consistency between retrieved bulk density from the two algorithms strengthens the reliability of the retrieved bulk water, which should be reasonable. Please see Figure R1.

As shown in Fig. R1(a), the retrieved bulk density values from the proposed algorithm and PIP gradually decrease from nearly 1.0 to 0.1 (g cm$^{-3}$) between 03 and 06 UTC. Both algorithms capture the transition from the mixing-phase to dry snow.

[Figure]

Figure R1: (a) The retrieved bulk density from collocated MRR and Parsivel. The blue dots are retrieved from CF-adjusted PSD. The red dots are from the original PSD. The gray dots are the retrieval from PIP. The case is 28 February 2018. (b) Same as (a), but for case 7 March 2018.

The manuscript has been revised to include the discussion of the bulk water fraction retrieval. **Please see Lines 398-413 in the revised manuscript. Or see the following.**

5.3 The bulk density comparison with collocated PIP

The precipitation imaging package (PIP), a video disdrometer, provides the PSD, fall speed, density, and snowfall rate of hydrometers (Newman et al., 2009; Pettersen et al., 2020) was

also deployed at the MHS site during ICE-POP 2018. Tokay et al. (2023) have utilized PIP to investigate the PSD parameters, including mass-weighted diameter and normalized intercept. The bulk density is estimated by Tokay et al. (2023) with various assumptions. The PIP retrieved density was generated from the assumption that $D_{max}$ = 1.15 $D_{eq}$, and the mass derivation included was based on Bohm (1989). As shown in Fig. 15, the retrieved bulk density from the proposed algorithm in this study and PIP have high consistency. Both retrieved bulk densities are highly correlated to each other, except for the period of 08 to 15UTC on 28 February due to the attenuation effect of the accumulated snow on the MRR antenna (Fig. 5e).

The bulk water fraction is derived along with the maximum possible bulk density using the proposed method in this study. The retrieved bulk water fraction will differ if a different assumption is made when selecting possible bulk density. Therefore, the performance of the retrieved bulk water fraction is linked with bulk density retrieval. Since there are no direct measurements of bulk water fraction, the consistency between retrieved bulk density from two algorithms "indirectly" confirms that the retrieved bulk water should be reasonable. As shown in Fig. 15a, the retrieved bulk density values from the proposed algorithm and PIP gradually decrease from nearly 1.0 to 0.1 (g cm$^{-3}$) between 03 and 06 UTC. Both algorithms capture the fast transition from the mixing-phase to dry snow.

Despite the various possible factors that can degrade the performance of bulk density retrieval from collocated MRR and Parsivel, the uncertainty study has shown that the retrieval performance has an acceptable low impact by the Parsivel and MRR observational error. The agreement between this study and PIP retrieval further confirms that the proposed algorithm can robustly retrieve the bulk density and water fraction from collocated MRR and Parsivel.

2. The use of Vz as a filter improves agreement between measured snowfall rates and snowfall rates estimated from the bulk-density retrievals.

Again, this isn't supported. The study describes retrieval performance using only data to which the filter has already been applied. It does not show retrieval results when the filter is not applied, so no judgement can be made about the effects of the filter.

**Reply: We did calculate the SR before and after the Vz filter, and the results did show great improvements. As per the reviewer's suggestion regarding the data quality of MRR, the MRR data has been replaced by the post-processed data, which has been applied to the algorithm by Maahn and Kollias (2012). All the bulk density, bulk water fraction, and reflectivity-weighted velocity retrievals have been recalculated. The original noisy retrieval results have been removed due to low SNR. The Vz calculation is no longer used to remove noisy bulk density retrieval. The Vz comparison is derived to verify the overall performance of bulk density retrieval. In addition, the Vz can identify the inadequate**

**bulk density retrieval due to the attenuation effect on MRR reflectivity. Please see Lines 185-187 in the revised manuscript. Or see the following.**

The comparison of $V_Z^{MRR}$ and $V_Z^{\rho bulk}$ is considered an overall validation of the retrieved bulk density. In addition, the inconsistency between the $V_Z^{\rho bulk}$ and the $V_Z^{MRR}$ can identify inadequate bulk density retrieval. For example, the attenuation effect can lead to underestimating the MRR reflectivity measurement and, thus, underestimating the retrieved bulk density.

3. Microphysical similarity between two warm low synoptic events confirms the dependence of the micro-scale factors on the synoptic conditions.

The results from the study show similarities in the retrieved microphysical properties for these two events, but that is not sufficient to support the conclusion. Perhaps other synoptic setups would produce similar microphysical properties, negating this conclusion.

**Reply: The warm-low events (28 February 2018 and 7 March 2018) shown in the manuscripts are the events that have the most accumulated snowfall during ICE-POP. The 28 February case demonstrates that the retrieved snow density is reasonable compared to Pluvio SR. However, some discrepancies between the calculated Vz from retrieved bulk density and measurements of Vz from MRR can be noticed. The discrepancy is due to the attenuation effect on the MRR antenna. The authors do not intend to emphasize the similarity of microphysical properties. These two events are selected to demonstrate the evolution of the bulk density retrieval results in detail. The discussion has been revised. Please see Lines 339-357 in the revised manuscript. Or see the following.**

To further understand the microphysical characteristics of winter precipitation, each site's retrieved bulk density and bulk water fractions are divided into warm-low (nine cases) and cold-low (five cases) events according to the synoptic condition (Gehring et al. 2020; Kim et al. 2021). As shown in Fig. 14a, the median values of bulk density of warm-low events from the mountain site (YPO) to the coastal site (GWU) are about 0.10 to 0.29 g cm$^{-3}$. The GWU site has the highest bulk density. On the other hand, the median values of bulk density of cold-low events from YPO to GWU are about 0.07 to 0.05 g cm$^{-3}$ (Fig. 14b). The overall bulk density values are lower in cold-low events than in warm-low events.

In Fig. 14c and d, more than 90% of bulk water fractions are less than 0.03 for warm- and cold-low events. The YPO site has the lowest bulk water fraction, especially the cold-low events that remain lower than 0.22. The mean value of the top 5% of the bulk water fraction of each site is obtained for further investigation. The values of bulk water fraction gradually increase from the mountain site (YPO) to the coastal site (GWU) for both warm- and cold-low

events. The mean values of the top 5% bulk water fraction of YPO, MHS, and CPO sites for warm-low events are 0.0015, and the BKC and GWU are about 0.32 to 0.45. The cold-low events are 0.0013 to 0.19 for each site.

The temperature ($^0$C) and water vapor pressure (hPa) measurements from nearby mountain and coastal AWS sites are collected and summarized in Fig. 14e. Warm-low events have warmer and moister conditions than cold-low events. The coastal area's warm- and cold-low events have similar mean temperature values. On the other hand, the water vapor pressure increases significantly from cold-low to warm-low events in the coastal region. The mountain area has similar features, but higher temperature increments and fewer increments of water vapor pressure. These results indicate that the winter precipitation systems of coastal sites with warmer and moister environments have higher bulk density and bulk water fraction than mountain sites.

4. Differences in bulk density and water fraction between mountain sites and coastal sites are indicative of geographical and synoptic environmental effects on the distinct microphysical characteristics of winter precipitation systems.

The geographical effects are suggested somewhat by the results from the second case study, but as noted in regards to conclusion #3, synoptic control can't be demonstrated using two cases with similar synoptic setup. Further, although the authors discuss differing meteorological properties at the coastal and mountain locations, there is no demonstration that the coastal and mountain sites differ in moisture availability.

**Reply: The number concentration of the retrieved bulk density is shown in Fig. 14. Both dry snow and mixing-phase events are shown in both warm-low and cold-low events. The median values of the retrieved bulk density of each site are shown in Fig. 14a,b. The temperature ($^0$C) and water vapor pressure (hPa) measurements from nearby mountain and coastal AWS sites are collected and summarized in Fig. 14e. The warm-low events have warmer and moister conditions compared to cold-low events. The coastal area's warm- and cold-low events have similar mean temperature values. On the other hand, the water vapor pressure increases significantly from cold-low to warm-low events. The mountain area has similar features but with higher temperature increments and fewer increments of water vapor pressure. Please see Lines 339-357 of the revised manuscript. Or see the previous reply.**

A. There are no estimates of retrieval uncertainties. This makes it impossible to determine, for example, if differences between the observed and retrieval-derived snowfall rates are significant.

**Reply: The retrieval uncertainty is investigated as per the reviewer's suggestion. The retrieval uncertainty analysis is performed by considering the assumption of particle shape, Parsivel measurement uncertainty, and the MRR measurement uncertainty. The detailed retrieval uncertainty analysis has been summarized in the newly added Discussion section. Please see Lines 358-417 in the revised manuscript. Or see the following.**

[revised manuscript text omitted]

B. The description of the methodology is not sufficient. For example, a Rayleigh reflectivity model is described, but that is not what is used in the radar forward model. Also, the description of how the ice and liquid volume fractions are determined in the retrieval is unclear and not well justified.

**Reply: The description of the methodology has been improved per the reviewer's suggestion. The description of the reflectivity calculation has been revised to improve the clarity. The equation 2 is no longer the Rayleigh reflectivity model. The backscattering cross-section ($\sigma$) replaces D$^6$ in equation 2. The reference of Bringi and Chandrasekar (2001) is provided to replace Huang et al. (2010). The ice and water are assumed to be evenly distributed within the particle. Please see Lines 125-171 of the revised manuscript. Or see the following.**

3 Methodology

Hydrometeor is composed of particles with combinations of solid ice and liquid water with a density of 0.92 ($\rho_{ice}$) and 1.0 ($\rho_{water}$) g cm$^{-3}$, respectively. Therefore, the hydrometeor bulk density ($\rho_{bulk}$) can be determined by its volume ratio of solid ice (vi) and liquid water (vw) as follows,

$$\rho_{bulk} = \text{vi} \times 0.92 + \text{vw}; \text{ g } cm^{-3}. \tag{1}$$

The sum of the values of vi and vw equals one or less than one if it contains air in the particle. Thus, the reflectivity factor (Z$_{HH}$) can be calculated as follows (Bringi and Chandrasekar 2001),

[revised manuscript text omitted]

C. The reflectivity and Doppler velocity data from the MRR require reprocessing to be representative of snowfall. See Maahn and Kollias (2012). It's not clear if this or similar reprocessing was performed.

**Reply: The MRR data has been post-processed per the reviewer's suggestion. The values of bulk density and bulk water fraction have been recalculated. The impact of the $Z_{HH}$ difference on the values of bulk density and bulk water fraction can be considered as the retrieval uncertainty due to MRR reflectivity measurement uncertainty. The results are shown in the Discussion section. Please see Lines 111-114 in the revised manuscript. Or see the following.**

The MRRs had the same configuration during ICEP-POP; the vertical resolution was 150 m, and there were 31 gates up to 4.65 km. The MRR data was post-processed using the algorithm from Maahn and Kollias (2012). The sensitivity of MRR has been enhanced, and the Doppler velocity has been dealiased. The third gate (450 m above ground) of $Z_{HH}$ data from MRR was selected to retrieve snow density since the first two gates contain some clutter contamination.

D. The discussion of results, particularly for the 7-8 March 2018 case, needs to be better organized and cleaned up to more clearly bring focus to the significant patterns in the results.

**Reply: The discussion of the 7-8 March 2018 case has been rewritten to improve the clarity. Please see Lines 278-322 in the revised manuscript. Or see the following.**

4.4 Case study: 7 March 2018

[revised manuscript text omitted]

E. There are assumptions of spherical particles in both the particle scattering calculations and in the fallspeed calculations, but only limited discussion of whether they are adequate for use with snowflakes and at the MRR's frequency.

**Reply: The discussion of non-spherical particles has been introduced in the revised manuscript. Non-spherical and spherical particle measurements are different when MRR looks upward. However, the orientation of the non-spherical particle is assumed to be isotropic and homogeneous. A sensitivity investigation assuming the particle axis ratio of 0.5 has been conducted. The results show that about 1.5 dBZ variation of simulated reflectivity can be induced due to the assumption of particle size.**

**A random error of MRR reflectivity with a standard deviation of 1.2 dB is introduced into the retrieval algorithm to imitate the particle assumption and MRR measurement uncertainty. The overall standard deviation of bulk density retrieval uncertainty is about 0.025 (g cm$^{-3}$) for a given MRR reflectivity uncertainty of 1.2 dB. The bulk water fraction retrieval has the same feature, and the uncertainty is about 0.041.**

**Please see the Discussion section (Lines 358-417) in the revised manuscript. Or see the reply to "comment A" shown previously, pages 7-9.**

F. The particle "size" measured by the Parsivel is ill-defined for snow particles. For background, in addition to Battaglia et al., see also Wood et al. (2013). This makes it difficult to make useful comparisons of bulk particle densities that are determined using different types of disdrometer measurements (e.g., Figure 13).

**Reply: In Figure 13, the bulk density comparison among this study, Heymsfield et al. (2004), and Brandes et al. (2007) does not intend to emphasize the difference. We convert D0 to Dm by assuming exponential PSD (Dm = D0*4/3.67). Considering distinct environmental conditions, instrumentations, and retrieval techniques, most of the particles in this study are consistent with the $\rho_{bulk} - D_m$ relation from Heymsfield et al. (2004) and Brandes et al. (2007). These results indicate that the proposed bulk density estimation algorithm can derive accurate retrievals with statistically consistent microphysical characteristics from previous studies. Please see Lines 324-338 in the revised manuscript. Or see the following.**

The retrieved bulk density and bulk water fraction are investigated statistically to understand the microphysical characteristics of the winter precipitation systems from ICE-POP 2018 and its pre-campaign. Fig. 13 shows the number concentration of retrieved bulk density and observed mass-weighted diameter ($D_m$) from PSD of all sites. The bulk density decreases exponentially as $D_m$ increases. Heymsfield et al. (2004) utilized the aircraft data collected from two field programs, namely the Atmospheric Radiation Measurement (ARM) program, Cirrus Regional Study of Tropical Anvils and Cirrus Layers (CRYSTAL) Florida Area Cirrus Experiment (FACE) in southern Florida during July 2002. The ARM data is mostly ice clouds formed primarily through large-scale ascent, and the CRYSTAL observations are mainly from convectively generated cirrus anvils. Brandes et al. (2007) utilized the data of 52 storm days from the Front Range in eastern Colorado during October–April 2003 to 2005 of a ground-based 2DVD. The data of Brandes et al. (2007) is dominated by almost spherical aggregates having near-exponential or superexponential size distributions. The $\rho_{bulk} - D_0$ relation ($D_0$, median volume diameter) from Brandes et al. (2007) is replaced by $D_m$ in Fig. 13 for comparison, assuming the exponential PSD and $D_m = 4D_0/3.67$.

Despite distinct environmental conditions, instrumentations, and retrieval techniques, most of the particles in this study are consistent with the $\rho_{bulk} - D_m$ relation from Heymsfield et al. (2004) and Brandes et al. (2007). These results indicate that the proposed bulk density estimation algorithm can derive reasonable retrievals with statistically consistent microphysical characteristics from previous studies.

F.  In further revision, English-language usage and grammar could be improved. I have tried to include some comments in the details below that may be helpful.

**Reply: Authors sincerely appreciate the reviewer's help. We have made all the corrections per both reviewers' suggestions. We will further improve the manuscript by asking for professional assistance from the English editor.**

Line-by-line comments: #######################

Abstract ********

L9: Does "bulk water fraction" mean bulk *liquid* water fraction? And is this the volume fraction or mass fraction?

Also, here and in other places, be careful how you use words to describe what you are retrieving. Here you say "hydrometeor's bulk density and bulk water fraction". This implies you are retrieving these properties for individual particles, which is not correct. You are retrieving the bulk density and bulk water fraction for populations of particles.

**Reply: The sentence has been revised to improve the clarity. "derive bulk density and bulk water fraction of a population of particles …". Please see Line 9.**

**The bulk water fraction is the volume fraction, not the mass fraction. Please see Lines 126-128 in the revised manuscript. Or see the following.**

Hydrometeor is composed of particles with combinations of solid ice and liquid water with a density of 0.92 ($\rho_{ice}$) and 1.0 ($\rho_{water}$) g cm$^{-3}$, respectively. Therefore, the hydrometeor bulk density ($\rho_{bulk}$) can be determined by its volume ratio of solid ice (vi) and liquid water (vw) as follows,

L13: The meaning of "The combination of minimum water fraction subsequently determines the bulk density" is not clear.

**Reply: The sentence has been revised to improve the clarity. "The combination of minimum water fraction and maximum ice fraction subsequently determines the bulk density ($\rho_{bulk}$)." Please see Lines 13-14 in the revised manuscript.**

L15-16: The meaning of "self-evaluation" in this context is not clear. L20-21: Regarding "a similar transition", from what state to what state?

**Reply: The "self-evaluation" has been removed in the revised manuscript. "The estimated $\rho_{bulk}$ was examined independently by comparison of the liquid-equivalent snowfall rate (SR) of collocated Pluvio." Please see Lines 15-16 in the revised manuscript.**

L21: Again, you are retrieving population properties, not the properties of individual particles.

**Reply: The sentence has been revised to improve the clarity. Please see Lines 21-22.**

L28: Do you actually mean "liquid water content" here, which usually means the concentration (e.g., grams per cubic meter) of liquid water?

**Reply: Yes, the authors refer to liquid water content (LWC, g cm$_{-3}$). The LWC has been constantly retrieved by dual-polarimetric radar measurements.**

Introduction ************

L30-31: This doesn't seem to be an example of either of the prior two statements in this section. What is it intended to exemplify?

**Reply: The examples intend to show how microphysical parameterizations in numerical forecast models can be improved by validating with the snow property obtained from observational data. The sentence has been revised to improve the clarity. Please see Lines 33-44 in the revised manuscript. Or see the following.**

These physical properties, including terminal fall speed, shape, composition, and density, are also crucial to verifying and improving microphysical parameterizations in numerical forecast models (Yuter et al., 2006; Kim et al., 2021). Simulating proper riming, freezing, and aggregation processes is challenging in numerical models. The riming processes led to snowfall velocity and density diversity in the same particle diameter size (Zhang et al. 2021). The supercooled liquid water freezes on snow particles and fills in the holes of snow; therefore, snow's mass and fall velocity increase while size has little changes (Heymsfield 1982; Moisseev et al. 2018). On the other hand, the melting process induces higher fall velocity and density by the aerodynamics process in a warm environment. The dispersed fall velocity caused by particles melted from dry snow to rain leads to higher collision efficiency and facilitates the aggregation and accretion processes (Yuter et al. 2006). Higher snow density is associated with steeper fall velocity-diameter (V-D) relations, which result from stronger riming (Lee et al. 2015) or melting processes (Yuter et al. 2006). To emulate the diverse physical properties of hydrometers, Morrison and Milbrandt (2015) proposed a new bulk method to parameterize ice-phase particles with evolvable density to study the role of density in numerical simulation. A

robust density estimation algorithm can evaluate microphysical simulations from numerical models.

L33-34: It's not specifically the increase in liquid phase fraction that causes the fallspeed of an individual particle to increase. It is the change in particle aerodynamics, specifically the reduction in particle size and horizontally-projected area while particle mass stays constant) that causes the fallspeed to increase.

**Reply: The sentence has been revised as per the reviewer's suggestion. "On the other hand, the melting process in a warm environment induces higher fall velocity and density by the aerodynamics process." Please see Lines 38-39.**

L38-39: It is not clear what point the authors are making with this statement. How is the work by Morrison and Milbrandt important to this work?

**Reply: The study from Morrison and Milbrandt (2015) proposed a new microphysical scheme that parameterizes the density of hydrometers. A robust density estimation algorithm can evaluate microphysical simulations from numerical models. Please see the revised manuscript, Line 44.**

L40: I think that "inhibits" is not the correct word here.

**Reply: The typo has been corrected. Please see Lines 30.**

L47: The results of Brandes et al. (2007) were limited to 52 cases over two winter seasons and isolated to a particular location. It doesn't seem correct to me to describe it as "climatological".

**Reply: The "climatological" has been removed. Please see Lines 48.**

L50-51: The meaning of "differentiation of riming degree" is unclear.

**Reply: The sentence has been revised to improve the clarity. Please see Lines 51-52.**

L54: Perhaps "above the 2DVD" would be clearer.

**Reply: The sentence has been revised as per the reviewer's suggestion. Please see Line 55.**

L55: I think that it is the difference that is minimized.

**Reply: The sentence has been revised as per the reviewer's suggestion. Please see Line 55.**

L61: I think "disdrometer" is more commonly used.

**Reply: The "distrometer" has been revised to "disdrometer". Please see Line 69.**

L64: "transmits" instead of "transmitted".

**Reply: The "transmitted" has been revised to "transmits". Please see Line 71.**

L65: "scatters" instead of "scattered".

**Reply: The "scattered" has been revised to "scatters". Please see Line 72.**

L67: See earlier comment re "distrometer".

**Reply: The "distrometer" has been revised to "disdrometer". Please see Line 74.**

L73-74: The meaning of "regarded as the self-evaluation of our result" is not clear.

**Reply: The "regarded as the self-evaluation of our result" has been removed to improve the clarity. Please see Lines 81-82 in the revised manuscript. Or see the following.**

Subsequently, the measurement of $Z_{HH}$ weighted fall velocity ($V_Z$) from MRR is compared with the calculated $V_Z$ from the derived bulk density and Parsivel PSD measurement.

Instruments and Data Processing *****************************

L85: Horizontal wind can cause problems with Pluvio and Parsivel measurements. Was any filtering or correction applied based on ambient wind speed?

**Reply: All of the Pluvios were equipped with double windshields. The Pluvio at the MHS was within the DFIR (double fence intercomparison reference) in addition to the double shield. The environmental conditions of all sites are introduced in the revised manuscript, please see Lines 99-102 in the revised manuscript. Or see the following.**

All of the Pluvios were equipped with double windshields. The Pluvio at the MHS was within the DFIR (double fence intercomparison reference) in addition to the double shield. All the sites investigated in this study have no taller trees or buildings near the MRR antenna and Parsivel. Each site's detailed layout and information can be found in Kim et al. (2021).

L90-91: What are the elevations of the sites?

**Reply: The elevations of each site are, YPO (772 m MSL), MHS (789 m a.m.s.l.), CPO (855 m MSL), BKC (175 m MSL), and GWU (36 m MSL). Each site's detailed layout and**

**information can be found in Kim et al. (2021). The reference has been added to the revised manuscript. Please see Lines 96-98.**

L93-95: Is it actually the PSD data that were filtered (PSD data do not include fall velocities)? Or was it the single-particle size and fallspeed data that were filtered? How does this filtering affect the calculated PSDs? Does it reduce particle counts?

**Reply: The Parsivel single-particle size and fallspeed data were filtered. The minute Parsivel data was quality-controlled using the fall velocity filtering technique (Lee et al. 2015). The filtering removes the outlier particles, reducing particle counts. Subsequently, the PSD was calculated from the filtered data. The manuscript has been revised to improve clarity. Please see Lines 107-110 in the revised manuscript. Or see the following.**

The minute Parsivel data was quality-controlled using the fall velocity filtering technique (Lee et al. 2015). The mean fall velocity and standard deviation ($\sigma$) for a given diameter were calculated, and the particles that deviate from the mean fall velocity of more than one standard deviation were filtered. The quality-controlled Parsivel data was subsequently processed to derive the PSD.

L96: What is the vertical resolution of the MRR data? What is the altitude above ground level of the third gate?

**Reply: The MRRs had the same configuration during ICEP-POP; the vertical resolution was 150 m, and there were 31 gates up to 4.65 km. The third gate is 450 m above ground. The vertical resolution information has been added to the revised manuscript. Please see Lines 111-114 in the revised manuscript. Or see the following.**

The MRRs had the same configuration during ICEP-POP; the vertical resolution was 150 m, and there were 31 gates up to 4.65 km. The MRR data was post-processed using the algorithm from Maahn and Kollias (2012). The sensitivity of MRR has been enhanced, and the Doppler velocity has been dealiased. The third gate (450 m above ground) of $Z_{HH}$ data from MRR was selected to retrieve snow density since the first two gates contain some clutter contamination.

L104-106: "Bias" already implies that an average was taken, no need to say "mean bias". What are the standard deviations of the differences between the MRR Z_HH and the simulated Z_HH? This would provide some insight into the uncertainty of the bias estimates.

**Reply: The "mean bias" has been revised to "bias" per the reviewer's suggestion. The number concentration plot of MRR measured reflectivity, and Parsivel PSD calculated reflectivity, as shown below in Figure R2. The standard deviation of the differences**

between them is about 1.1 to 1.3 dB for each site (shown in Table 2 of the revised manuscript). As the reviewer indicates, the standard deviation value can be considered the MRR reflectivity uncertainty. The standard deviation will be further applied to investigate the bulk density retrieval uncertainty.

[Figure]

**Figure R2: The number concentration distribution of measured reflectivity from MRR and simulated reflectivity from Parsivel of BKC site. The bias is -2.1 dBZ, and the standard deviation is 1.28 dBZ.**

Methodology ***********

L107: In the entirety of the Methodology section, there is no discussion of how uncertainties are determined for the retrieved properties or properties derived from the retrieval results.

**Reply: As per the reviewer's suggestion, the retrieval uncertainty has been investigated and summarized in the revised manuscript. With 1.2 dB MRR reflectivity uncertainty, the retrieval bulk density uncertainty is about 0.023 g cm⁻¹. A Discussion section has been added to the revised manuscript. Please see Lines 358-417 in the revised manuscript. Or see the reply to "comment A" shown previously, pages 7-9.**

L112-114: It is unclear why the Huang et al. study is introduced at the beginning of the methodology. The Rayleigh assumption used by Huang et al. is clearly not appropriate for K-band radar (MRR) observing snowfall, so equation 2 is not applicable. The actual equation for estimating $Z\_HH$ using T-matrix backscatter cross-sections and attenuation is never presented or discussed.

**Reply: Authors agree with the reviewer's comment. Since T-matrix simulation is used and the Rayleigh assumption is not used in the retrieval algorithm, equation 2 has been revised. The reference has also been changed to Bringi and Chandrasekar (2001). The sentence of Rayleigh assumption is removed to improve the manuscript's clarity. Please see Lines 130-134 in the revised manuscript. Or see the following.**

The sum of the values of vi and vw equals one or less than one if it contains air in the particle. Thus, the reflectivity factor ($Z_{HH}$) can be calculated as follows (Bringi and Chandrasekar 2001),

$$Z_{HH} = \left(\frac{\rho_{bulk}}{\rho_{ice}}\right)^2 \frac{|K_{ice}|^2}{|K_w|^2} \frac{\lambda^4}{\pi^5} \int \sigma(D)N(D)dD; \; mm^6 \, m^{-3}. \qquad (2)$$

The $K_{ice}$ and $K_w$ are the dielectric factors of solid ice and liquid water, respectively. $\sigma$ is the backscattering cross-section, $D$ is the particle size, and $N(D)$ is the particle size distribution. As shown in (1) and (2), the $Z_{HH}$ is positively correlated to $\rho_{bulk}$.

L121: "modified from Huang et al.", I believe.

**Reply: The typo has been corrected. Please see Line 138.**

L126-127: More details are needed here. How were the dielectric properties of the mixed ice/liquid/air particles determined? How was the liquid water assumed to be distributed within a particle?

**Reply: The ice and water are evenly distributed within the particle. The manuscript has been revised to improve the clarity. The sentence "The ice and water are assumed to be evenly distributed within the particle." has been added, please see Lines 144-145.**

L128-130: It's not clear to me why spherical shapes were assumed just because the snow particles are observed from the bottom. I believe the particles would still appear to be non-spherical. It is not clear that spherical particle T-matrix calculations are appropriate for modeling snowflake backscattering a K-band.

**Reply: Non-spherical and spherical particles do appear differently when looking upward. However, the orientation of the non-spherical particle is assumed to be isotropic and homogeneous. A sensitivity investigation assuming the particle axis ratio of 0.5 has been conducted. The results show that about 1.5 dBZ variation of simulated reflectivity can be induced due to the assumption of particle size.**

**A random error of MRR reflectivity with a standard deviation of 1.2 dB is introduced to imitate the particle assumption and MRR measurement uncertainty to the**

**retrieval uncertainty. The overall standard deviation of bulk density retrieval is about 0.025 (g cm$^{-3}$) for a given MRR reflectivity uncertainty of 1.2 dB. The bulk water fraction retrieval has the same feature, and the uncertainty is about 0.041. Please see the Discussion section in the revised manuscript, Lines 358-417. Or see the reply to "comment A" shown previously, pages 7-9.**

L138-144: Something seems off about the results shown in Figure 2. The size distribution in panel (a) shows the size distribution consists of small particles and that the concentrations of those particles are small. I would expect the radar reflectivity to be small, in the neighborhood of 0 to 3 dBZ with typical ice densities for snowflakes based on other field experiment results I've examined with similar size distributions. Yet according to the dashed blue line in panel (b) the calculated reflectivity reaches over 18 dBZ, even with very small ice densities and virtually no liquid water.

Please check these calculations.

**Reply: The calculated reflectivity values reaching over 18 dBZ are due to the high water fraction and dielectric constant. The calculated reflectivity from PSD ranges from -5 to 35 dBZ, and the values cover 0 to 3 dBZ (low ice and water fraction). The 18 dBZ is the reference reflectivity value from MRR to constrain the retrieved bulk density. We will use the bulk density retrieval from the precipitation imaging package (PIP) to verify the calculation of T-Matrix simulation and retrieved density.**

**The PIP, a video disdrometer, provides the PSD, fall speed, density, and snowfall rate of hydrometers (Newman et al., 2009; Pettersen et al., 2020) was also deployed at the MHS site during ICE-POP 2018. Tokay et al. (2023) have utilized PIP to investigate the PSD parameters, including mass-weighted diameter and normalized intercept. The bulk density is estimated by Tokay et al. (2023) with various assumptions. The PIP retrieved density was generated from the assumption that $D_{max} = 1.15 D_{eq}$, and the mass derivation included was based on Bohm (1989). The time series of retrieved bulk density from the proposed algorithm and PIP are shown in Fig. R1.**

**We change the data time of Figure 2 to 1559 UTC in the revised manuscript. The PSD has a higher concentration of particles, thus the reflectivity values range from – 5 to 50 dBZ. The MRR reference reflectivity is 22.23 dB. Please see the Fig. 2 in the revised manuscript. As shown in Fig. R1, the retrieved bulk density from the proposed algorithm (blue and red dots) and the PIP (gray dots) are in good agreement. The bulk density was about 0.07 (g cm$^{-1}$) from both methods. The consistency of retrieved bulk density confirms the calculation of the bulk density and the simulated reflectivity from PSD. The discussion of the consistency between the proposed method and the PIP has been included in the**

revised manuscript. **Please see Lines 398-413 (the Discussion, 5.3 section) in the revised manuscript. Or see the reply to "comment A" shown previously, pages 8-9.**

L145-150: I don't see this recommendation in Huang et al., so I think it is necessary to explain more fully the reasoning for this approach and to describe more completely the details of the approach. Do you mean that given the observed reflectivity, you would just pick the largest vi that reproduces that reflectivity? Why?

**Reply: One difference between our algorithm and Huang et al.'s (2010) is the assumption of the particle composition. Huang et al. (2010) assumed that a mixture of snow contains only ice and air. Please see page 642 of Huang et al. (2010), "To calculate the backscattering properties of the particles measured by the 2DVD, we consider snow to be a mixture of ice and air." The assumption of only ice and air from Huang et al. (2010) is the same as the vw (water fraction) equals zero in Fig. 2b. The bottom part of Fig. 2b (e.g., vw = 0) is exactly the same as Huang et al. (2010).**

**In our proposal algorithm, the reflectivity calculation fully considers water/ice/air fractions. Therefore, various water/ice/air fraction combinations can be derived from the matched reflectivity between MRR measurement and Parsivel calculation. The bulk density with maximum vi (minimum vw) with maximum bulk density is selected to determine the bulk snow density from these possible combinations of vw/vi. Assuming the minimum water fraction is similar to Huang et al. (2010)'s assumption. The manuscript has been revised to improve the clarity, please see Lines 157-171. Or see the following.**

A selected example of the simulated $Z_{HH}$ from observed PSD with various combinations of vi/vw is shown in Figure 2. The observed PSD from Parsivel (Fig. 2a) was applied to the T-Matrix backscattering simulation. All possible combinations of vi/vw were applied to calculate simulated $Z_{HH}$ (Fig. 2b). The corresponding bulk density was derived via (1) and shown as contour dash lines in Fig. 2b. The values of simulated $Z_{HH}$ vary from 0 to 50 dBZ (shaded color in Fig. 2b). The simulated $Z_{HH}$ values increase with increasing of the bulk snow density. In this selected case, the observed $Z_{HH}$ from MRR was 22.23 dBZ (dashed blue line in Fig. 2b). The observed $Z_{HH}$ from MRR was thus applied to constrain the possible combination of vi/vw and bulk density. The possible ranges of vi/vw are 0.0/0.009 to 0.08/0.0, shown as a dashed blue line in Fig. 2b. The corresponding bulk densities are 0.009 and 0.074 (g cm$^{-3}$), respectively. The higher the fraction of ice (e.g., vi), the higher bulk snow density values can be found in Fig. 2b. To determine the bulk snow density from possible combinations of vi/vw, the maximum bulk density with maximum vi is selected. Choosing the bulk density with maximum vi ensures the minimum value of vw. This assumption is similar to Huang et al. (2010), which assumes that a mixture of snow contains only ice and air. The water fraction is not considered

in Huang et al. (2010). Therefore, the contour's maximum density in Fig. 2(b), 0.074 g cm$^{-3}$, is determined by assuming $v_i$ and $v_w$ are 0.08 and 0.0, respectively. Subsequently, the $v_w$ is regarded as "bulk water fraction," which can also be estimated in the proposed method, in addition to the bulk density, and will be analyzed in the following section.

L149-153: This part of the methodology also requires more complete explanation and evidence. I'm not sure I follow and agree with your argument here.

**Reply: The bulk water fraction is derived along with the maximum possible bulk density in the proposed method in this study. If a different assumption is made when selecting possible bulk density, the retrieved bulk water fraction will be different. Therefore, the performance of the retrieved bulk water fraction is linked with bulk density retrieval. Since there are no direct measurements of bulk water fraction, we will compare the retrieved bulk density from the proposed method and PIP. The consistency between retrieved bulk density from the two algorithms confirms that the retrieved bulk water should be reasonable.**

**As shown in Fig. R1(a), the retrieved bulk density values from the proposed algorithm and PIP gradually decrease from nearly 1.0 to 0.1 (g cm$^{-3}$) between 03 and 06 UTC. Both algorithms capture the transition from the mixing-phase to dry snow. Please see Figure R1. The manuscript has been revised to include the discussion of the bulk water fraction retrieval. Please see Lines 398-413 (the Discussion, 5.3 section) in the revised manuscript. Or see the reply to "comment A" shown previously, pages 8-9.**

Since T-matrix is being used rather than equation (2), it may not be clear to many readers how the liquid and ice water dielectric factors come into play. I expect you are using some form of mixing rule (e.g., Maxwell-Garnet?). I think explanation needs to be provided about how the particle dielectric properties are determined for a mixture of ice and liquid water and how this influences backscattering properties as $v_i$ and $v_w$ change.

Further, Figure 2b seems to show that there is only a narrow range of the solution space ($v_w$ = 0.015 to 0.1 with $v_i$ < 0.5) for which Z might be said to be moderately more sensitive to $v_w$ than to $v_i$ due to liquid water's larger dielectric constant. Also, how is this sensitivity to $v_w$ influenced by your method for choosing $v_i$? Clearly, if you pick the maximum $v_i$ for this case, there is much weaker sensitivity of Z to $v_w$. Finally, it is not clear what is meant by "The change of $v_w$ can be ... obtained ...".

**Reply: In reply to the previous comment, the performance of the retrieved bulk water fraction is linked with bulk density retrieval. Since there are no direct measurements of bulk water fraction, we will compare the retrieved bulk density from the proposed**

**method and PIP. The retrieved bulk density values from the proposed algorithm and PIP gradually decrease from nearly 1.0 to 0.1 (g cm⁻³) between 03 and 06 UTC in Fig R1(a). Both algorithms capture the transition from the mixing-phase to dry snow. Please see Figure R1. The manuscript has been revised to include the discussion of the bulk water fraction retrieval. Please see Lines 398-413 (the Discussion, 5.3 section) in the revised manuscript. Or see the reply to "comment A" shown previously, pages 8-9.**

**The mixing rule of Maxwell-Garnet is applied to the T-Matrix calculation. The influence of vi/vw composition on the backscattering properties has been discussed in the manuscript. Please see Lines 133-135 in the revised manuscript. Or see the following.**

The $K_{ice}$ and $K_w$ are the dielectric factors of solid ice and liquid water, respectively. $\sigma$ is the backscattering cross-section, $D$ is the particle size, and $N(D)$ is the particle size distribution. As shown in (1) and (2), the $Z_{HH}$ is positively correlated to $\rho_{bulk}$. The higher hydrometer bulk density has a higher value of $Z_{HH}$ for a given PSD.

L153-156: For clarity, I would briefly describe both approaches here, then follow with more detailed descriptions of each one. What is meant by "self-verified"?

**Reply: The description has been revised to improve the clarity. The original purpose of utilizing reflectivity-weighted velocity to filter adequate retrieval is no longer needed and has been removed in the revised manuscript. The quality retrieval results have been greatly improved by applying the post-processed MRR data per the reviewer's suggestion. The low SNR MRR measurement has been removed. The comparison of reflectivity-weighted velocity is mainly used to identify the inadequate retrieval due to the attenuation effect on MRR reflectivity. Please see Lines 185-187 in the revised manuscript. Or see the following.**

The comparison of $V_Z^{MRR}$ and $V_Z^{\rho bulk}$ is considered an overall validation of the retrieved bulk density. In addition, the inconsistency between the $V_Z^{\rho bulk}$ and the $V_Z^{MRR}$ can identify inadequate bulk density retrieval. For example, the attenuation effect can lead to underestimating the MRR reflectivity measurement and, thus, underestimating the retrieved bulk density.

L157-165: This is for spherical particles. Do you assert it is appropriate for snow particles? How does this relationship compare with Mitchell and Heymsfield (2005) or Heymsfield and Westbrook (2010)? These newer fallspeed models are more appropriate for snowflakes.

**Reply: Per the reviewer's suggestion, the retrieval results have significantly improved after applying the post-processed MRR data (Maahn and Kollias 2012). The noisy bulk**

**density has been removed. The original purpose of removing inadequate bulk density retrieval by reflectivity-weighted velocity is no longer needed. The reflectivity-weighted velocity comparison is obtained for two purposes. First, the comparison intends to examine the overall performance of the retrieved bulk density. The overall consistency is shown in Fig. 3. Second, the "bulk density-derived" reflectivity-weighted velocity is obtained to identify antenna attenuation issues as shown in Fig. 5(e). The issue of spherical particle assumption on terminal velocity calculation has been discussed in the revised manuscript.** **Please see Lines 218-221 in the revised manuscript. Or see the following.**

The various shapes aerodynamically complicate the falling behaviors of ice-phase and mixed-phase particles (Mitchell and Heymsfield 2005; Heymsfield and Westbrook 2010). Moreover, various measurement issues of MRR and Parsivel also induce some inconsistency. Nevertheless, the overall consistency of the $V_Z^{MRR}$ and $V_Z^{\rho_{bulk}}$ suggests that the retrieved bulk density is an adequately reasonable value.

L166-167: This is not a correct statement. Both Vz_MRR and Z_MRR (which is used to constrain the retrieval) are derived from the same basic measurements of Doppler spectra. So they are not independent.

**Reply: The manuscript has been revised as per the reviewer's suggestion. Please see Lines 185-187.**

L168-169: But what were this "various issues"?

**Reply: The manuscript has been revised as per the reviewer's suggestion. Please see Lines 185-187.**

L170-171: So, my understanding is that, for the data presented in the results, any retrievals with retrieved Vz greater than observed Vz plus one standard deviation are excluded. Is that correct? How does the 1-sigma uncertainty in the obsered Vz compare against the 1-sigma uncertainty in the retrieved Vz?

**Reply: The V$_Z$ criteria for removing inadequate bulk density retrieval is no longer needed. As per the reviewer's suggestion, the retrieval results have significantly improved after applying the post-processed MRR data (Maahn and Kollias 2012). The noisy bulk density has been removed. The Vz difference is mainly used to identify the attenuation effect of MRR reflectivity.** **Please see Lines 185-187. Or the reply to "L153-156" on page 10.**

L176: I think the term on the right of the summation needs to be multiplied by the size bin width (delta_D_i) before summation.

**Reply: The typo has been corrected. Please see Line 191.**

L178: Perhaps "compared against" rather than "examined with".

**Reply: The sentence has been revised as per the reviewer's suggestion. Please see Line 193.**

Results *******

L184: The Results contain no assessments of uncertainties in the observations (Z_HH, Vz, PSD, SR) , in the retrieved properties (bulk particle density, bulk liquid water fraction), or in the properties derived from the retrieval results (Z_HH, Vz, SR). How are we to determine if the retrieval results and Vz and SR biases, for example, are significant or not?

**Reply: As per the reviewer's suggestion, the discussion of the retrieval uncertainty has been included in the revised manuscript. Please see the Discussion section for more details. (Lines 358-417). Or see the reply to "comment A" shown previously, pages 7-9.**

Reflectivity-weighted (Vz) ============================

L197: -0.27 to 0.03 is the range in bias values only, not related to standard deviation.

**Reply: The Vz has been recalculated by using post-processed MRR data. The values of standard deviation are provided in the submitted manuscript. The sentence has been revised to improve the clarity. Please see Lines 206-216 in the revised manuscript. Or see the following.**

4.1 Reflectivity-weighted ($V_Z$)

The normalized number density function of measured $V_Z^{MRR}$ from MRR and "density-calculated" $V_Z^{\rho bulk}$ from Parsivel PSD of five sites are shown in Fig. 3. The $V_Z^{MRR}$ and $V_Z^{\rho bulk}$ are in agreement with each other. The majority of the data show reasonably consistent values. The GWU site had the most consistent velocity. On the other hand, YPO, MHS, and CPO sites had second peak values of about 2.0-3.0 m s$^{-1}$ of $V_Z^{MRR}$ and 1.0-2.0 m s$^{-1}$ of $V_Z^{\rho bulk}$. In general, the $V_Z^{\rho bulk}$ values of YPO, MHS, CPO, and are slightly lower than $V_Z^{MRR}$. The bias values (Table 3) are about -0.81 ms$^{-1}$ to 0.01 ms$^{-1}$. The standard deviation values are about 1.02 to 1.88 ms$^{-1}$, respectively. All sites combined mean bias and standard deviation values are -0.46 ms$^{-1}$ and 1.35 ms$^{-1}$, respectively. It is postulated that the lower values of $V_Z^{\rho bulk}$ than of $V_Z^{MRR}$ is

caused by the attenuation effect on the MRR reflectivity. The attenuated reflectivity leads to underestimation of retrieved bulk density and "density-calculated" $V_Z^{\rho bulk}$. An example of attenuated reflectivity will be discussed in the case study of 28 February 2018. The retrieved bulk density influenced by the attenuation effect will be identified and removed by visual examination of Vz comparison.

L199: Clarify that this is the bias and standard deviation for all site results combined.

**Reply: The bias and standard deviation are all site results combined. The sentence has been revised to improve the clarity. Please see Lines 212-213 in the revised manuscript.**

L200-201: It would be appropriate to acknowledge this limitation earlier in the paper where the method is introduced.

**Reply: More discussion has been added to the revised manuscript. Please see Lines 217-221.**

L203: Usually, "mixed-phase".

**Reply: The typo has been corrected. Please see Line 219.**

L203-204: Again, there is a vague reference to "measurement issues", but there has been no descriptive discussion or quantification of them.

**Reply: The discussion has been revised. Please see Lines 214-221.**

L206-207: This kind of filtering (omitting data from further analysis simply because the data don't give results that match other observations) tends to negate or reduce the believability of the proposed method. This is especially true when the authors cannot point to specific physical conditions that caused the method to fail. How much data was filtered at this stage? How poor are the subsequent results if the data are not filtered?

**Reply: The V$_Z$ criteria for removing inadequate bulk density retrieval is no longer needed. Per the reviewer's suggestion, the retrieval results have significantly improved after applying the post-processed MRR data (Maahn and Kollias 2012). The noisy bulk density due to low SNR has been removed. The post-processed MRR data has greatly improved its sensitivity. The Vz difference is mainly used to identify the attenuation effect of MRR reflectivity. Please see Lines 215-216 of the revised manuscript. Or see the following.**

The retrieved bulk density influenced by the attenuation effect will be identified and removed by visual examination of Vz comparison.

Liquid-equivalent snowfall rate (SR) =====================================

L215-216: Snow gauges like the Pluvio can have problems with undercatch when surface winds are strong. Were the winds checked and any filtering or corrections applied? The bias in the density-derived SR versus the Pluvio SR might be worse if the Pluvio data are corrected for undercatch.

**Reply: All of the Pluvios were equipped with double windshields. The Pluvio at the MHS was within the DFIR (double fence intercomparison reference) in addition to the double shield. The environmental conditions of all sites are introduced in the revised manuscript, please see Lines 99-102. Or see the following.**

All of the Pluvios were equipped with double windshields. The Pluvio at the MHS was within the DFIR (double fence intercomparison reference) in addition to the double shield. All the sites investigated in this study have no taller trees or buildings near the MRR antenna and Parsivel. Each site's detailed layout and information can be found in Kim et al. (2021).

L216-219: This is the first mention of snow/ice accumulation on the MRR antenna. It would be appropriate to mention that this occurred during the description of the observations earlier in the paper.

**Reply: The revised manuscript has added a discussion of the attenuation effect of MRR due to snow/ice accumulation on the antenna. Please see Lines 213-216 in the revised manuscript. Or see the following.**

It is postulated that the lower values of $V_Z^{\rho bulk}$ than of $V_Z^{MRR}$ is caused by the attenuation effect on the MRR reflectivity. The attenuated reflectivity leads to underestimation of retrieved bulk density and "density-calculated" $V_Z^{\rho bulk}$. An example of attenuated reflectivity will be discussed in the case study of 28 February 2018. The retrieved bulk density influenced by the attenuation effect will be identified and removed by visual examination of Vz comparison.

L224: Should be "moist air".

**Reply: The typo has been corrected. Please see Line 238.**

L227-228: For the case study of the 28 February event, why is only the MHS site data analyzed?

**Reply: The 28 February 2018 event is selected to demonstrate the retrieval results. The consistency of SR calculated from retrieved bulk density and measurement from Pluvio indicate that the proposed algorithm performs reasonably well. The pronounced attenuation effect of MRR reflectivity and its impact on underestimating Vz are shown**

to demonstrate the retrieval uncertainty. The other sites show almost the same evolution of the retrieved properties. In addition, the PIP was deployed at MHS and collocated with MRR and Parsivel. The comparison of retrieved bulk density from our method and PIP is discussed in the revised manuscript. Only the MHS site of 28 February is shown to keep the manuscript concise. A more detailed analysis of each site of 7 March 2018 is discussed.

Case study: 28 February 2018 ==============================

L258: Regarding "fall velocity was more significant than 1 m s^-1", I suggest rewording this to avoid confusion with statistical significance.

**Reply: The sentence has been revised to improve the clarity. Please see Line 271.**

L263: Regarding "derivation density", do you mean "derived density"?

**Reply: The typo has been corrected. Please see Lines 276.**

L263-264: Are you describing the *maximum* particle sizes?

**Reply: The typo has been corrected. Please see Lines 276.**

Case study: 7 March 2018 =========================

L268-269: "produced prominent precipitation" and "produced intensive precipitation" sounds like repetition, are both needed?

**Reply: The "produced prominent precipitation" has been removed. Please see Lines 280-281.**

L273-310: There are a number of locations on these lines that describe bulk water fraction. See my major comments above - I don't think the capability of the retrieval to distinguish and quantify bulk water fraction (or volume fraction of liquid water) has been demonstrated.

**Reply: The bulk water fraction is derived along with the maximum possible bulk density in the proposed method in this study. If a different assumption is made when selecting possible bulk density, the retrieved bulk water fraction will be different. Therefore, the performance of the retrieved bulk water fraction is linked with bulk density retrieval. Since there are no direct measurements of bulk water fraction, we will compare the retrieved bulk density from the proposed method and PIP. The consistency between retrieved bulk density from the two algorithms confirms that the retrieved bulk water should be reasonable. Please see Figure R1.**

As shown in Fig. R1(a), the retrieved bulk density values from the proposed algorithm and PIP gradually decrease from nearly 1.0 to 0.1 (g cm$^{-3}$) between 03 and 06 UTC. Both algorithms capture the transition from the mixing-phase to dry snow. Please see Figure R1 in the previous reply. The manuscript has been revised to include the discussion of the bulk water fraction retrieval. **Please see Lines 398-417 in the revised manuscript. Or see the reply to "comment A" shown previously, pages 7-9.**

L286: Regarding "which are in accord with the distributions of all velocity-diameter relations", it is not clear to me what this means.

**Reply: The sentence has been revised to "Overall, the retrieved bulk density and bulk water fraction successfully reveal distinct fall velocity-diameter relations of each site due to the different synoptic environments." to improve clarity. Please see Lines 295-296.**

L288: Regarding "They gradually dissipated", it is not clear what "They" is referring to.

**Reply: "They" refers "the precipitation". The sentence has been revised to improve the clarity. Please see Lines 299.**

L293-294: Regarding "Hence, it implies more ... confirm the distribution of fall velocity and diameter". The meaning here is not clear to me.

**Reply: The sentence has been revised to "The high bulk density (about 0.9 g cm$^{-3}$) period corresponds to a higher bulk water fraction from 0.4 to 0.8." to improve clarity. Please see Lines 302-303.**

L296: Regarding "confirmed by the alike contrast", the meaning of "alike contrast" is not clear.

**Reply: The sentence has been revised to "The higher density initially in MHS and BKC sites are also confirmed by the same contrast between 08 to 19 UTC on 7 March (Figs. 12a, b) than 19 UTC to 03 UTC on 8 March (Figs. 12d, e)." to improve the clarity. Please see Lines 306-308.**

L298: Not true, YPO, MHS and CPO, BKC show mostly near-zero bulk water fraction. For most of this discussion, need to be clear about when only-elevated, only-coastal, or all sites are being described.

**Reply: Only BKC and GWU feature high bulk water fractions. The sentence has been revised to improve the clarity. The sentence has been revised to "the BKC and GWU sites featured high bulk density and bulk water fraction (Figs. 10a-11a)." to improve clarity. Please see Line 309.**

L300: "Transited" should be "transitioned".

**Reply: The type has been corrected. Please see Line 312.**

Statistical analysis of bulk density and bulk water fraction

=====================================================================

L317-320: What is the basis of the assertion that Brandes et al. (2007) observations were dominated by "almost spherical aggregates"? Brandes et al. appear to have used the equivalent volume diameter as determined by the 2DVD software, as particle sizes. These, will be different than the particle size determined by the Parsivel. Brandes et al. do use the median volume diameter to parameterize the bulk density; however it is not evident that the cases in this study and those of Brandes et al. involved similar meteorological conditions. Evidence should be presented for this claim.

**Reply: The statement "dominated by almost spherical aggregates" can be found in the abstract of Brandes et al. (2007). Considering distinct environmental conditions, instrumentations, and retrieval techniques, most of the particles in this study are consistent with the $\rho_{bulk} - D_m$ relation from Heymsfield et al. (2004) and $\rho_{bulk} - D_0$ from Brandes et al. (2007). These results indicate that the proposed bulk density estimation algorithm can derive reasonable retrievals with statistically consistent microphysical characteristics from previous studies. The manuscript has been revised to improve the clarity. Please see Lines 324-338 in the revised manuscript. Or see the following.**

The retrieved bulk density and bulk water fraction are investigated statistically to understand the microphysical characteristics of the winter precipitation systems from ICE-POP 2018 and its pre-campaign. Fig. 13 shows the number concentration of retrieved bulk density and observed mass-weighted diameter ($D_m$) from PSD of all sites. The bulk density decreases exponentially as $D_m$ increases. Heymsfield et al. (2004) utilized the aircraft data collected from two field programs, namely the Atmospheric Radiation Measurement (ARM) program, Cirrus Regional Study of Tropical Anvils and Cirrus Layers (CRYSTAL) Florida Area Cirrus Experiment (FACE) in southern Florida during July 2002. The ARM data is mostly ice clouds formed primarily through large-scale ascent, and the CRYSTAL observations are mainly from convectively generated cirrus anvils. Brandes et al. (2007) utilized the data of 52 storm days from the Front Range in eastern Colorado during October–April 2003 to 2005 of a ground-based 2DVD. The data of Brandes et al. (2007) is dominated by almost spherical aggregates having near-exponential or superexponential size distributions. The $\rho_{bulk} - D_0$ relation ($D_0$, median volume diameter) from Brandes et al. (2007) is replaced by $D_m$ in Fig. 13 for

comparison, assuming the exponential PSD and $D_m = 4D_0/3.67$.

Despite distinct environmental conditions, instrumentations, and retrieval techniques, most of the particles in this study are consistent with the $\rho_{bulk} - D_m$ relation from Heymsfield et al. (2004) and Brandes et al. (2007). These results indicate that the proposed bulk density estimation algorithm can derive reasonable retrievals with statistically consistent microphysical characteristics from previous studies.

L321-325: The particle sizes used in Heymsfield et al. (2004) are derived from aircraft particle probes, as you have noted. These particle sizes are probably more like the "maximum dimension" of the particle and less like the "equivalent diameter" determined by a Parsivel. Additionally, Heymsfield et al. relate density to mass mean diameter, not to median volume diameter. So the comparison described here is somewhat an "apples to oranges" comparison. It is not surprising there are differences.

**Reply: Since the two papers use different parameters (Dm and D₀) to present mean size, we convert D₀ to Dm by assuming exponential PSD (Dm = 4D₀/3.67). Moreover, the bulk density comparison among this study, Heymsfield et al. (2004), and Brandes et al. (2007) does not intend to emphasize the difference. The discussion has been rephrased. The Figure has been revised. Please see Figure 13 and Lines 324-338. Or see the reply to previous comments.**

L335-344: As I noted above, I am not convince that this method is capable of accurately distinguishing and quantifying the liquid and ice volume ratios and the corresponding bulk water fraction. Also, although it is asserted that there are differences in the meteorology of the warm-low and cold-low events (i.e., "warmer and moister environments" for the warm-low events), no meteorological data is provided to support this.

**Reply: As per the previous comments and replies, we have compared the retrieved bulk density from the proposed algorithm and PIP retrieval (see below figures) and the SR with Pluvios. The bulk water fraction is derived along with the maximum possible bulk density in the proposed method in this study. If a different assumption is made when selecting possible bulk density, the retrieved bulk water fraction and density will be different. Therefore, the performance of the retrieved bulk water fraction is linked with bulk density retrieval. Since there are no direct measurements of bulk water fraction, we compared the retrieved bulk density from the proposed method and PIP. The consistency between retrieved bulk density from the two algorithms confirms that the retrieved bulk water should be reasonable.**

**As shown in Fig. R1(a), the retrieved bulk density values from the proposed**

**algorithm and PIP gradually decrease from nearly 1.0 to 0.1 (g cm⁻³) between 03 and 06 UTC. Both algorithms capture the transition from the mixing-phase to dry snow. Please see Figure R1 in the previous reply. The manuscript has been revised to include the discussion of the bulk water fraction retrieval.** **Please see Line 398-417. Or see the reply to "comment A" shown previously, pages 8-9.**

**In reply to the previous comment, the temperature ($^0$C) and water vapor pressure (hPa) measurements from nearby mountain and coastal AWS sites are collected and summarized in Fig. 14e. The warm-low events have warmer and moister conditions compared to cold-low events. The coastal area's warm- and cold-low events have similar mean temperature values. On the other hand, the water vapor pressure increases significantly from cold-low to warm-low events. The mountain area has similar features but with higher temperature increments and fewer increments of water vapor pressure.** **Please see Lines 351-357 in the revised manuscript. Or see the following.**

The temperature ($^0$C) and water vapor pressure (hPa) measurements from nearby mountain and coastal AWS sites are collected and summarized in Fig. 14e. Warm-low events have warmer and moister conditions than cold-low events. The coastal area's warm- and cold-low events have similar mean temperature values. On the other hand, the water vapor pressure increases significantly from cold-low to warm-low events in the coastal region. The mountain area has similar features, but higher temperature increments and fewer increments of water vapor pressure. These results indicate that the winter precipitation systems of coastal sites with warmer and moister environments have higher bulk density and bulk water fraction than mountain sites.

L345: It is probably more appropriate to say that the density of snow varies with "imposed weather conditions".

**Reply: The sentence has been revised to improve the clarity. Please see Lines 419.**

Conclusions ***********

L347-350: As I've noted, I have concerns about the bulk water fraction estimates. I don't believe sufficient proof of the capability has been provided, and in no way has evidence been provided that the values are "precise". The high sensitivity of $Z_{HH}$ to the liquid portion of the particle led to precise bulk water fraction estimation. It implied better capability of the density variation due to bulk water fraction change (ex. melting) in the proposed method in this study.

**Reply: Thanks reviewer's suggestion. As per the previous comments and replies, we have compared the retrieved bulk density from the proposed algorithm and PIP retrieval (see**

below figures) and the SR with Pluvios. The bulk water fraction is derived along with the maximum possible bulk density in the proposed method in this study. If a different assumption is made when selecting possible bulk density, the retrieved bulk water fraction will be different. Therefore, the performance of the retrieved bulk water fraction is linked with bulk density retrieval. Since there are no direct measurements of bulk water fraction, we compared the retrieved bulk density from the proposed method and PIP. The consistency between retrieved bulk density from the two algorithms confirms that the retrieved bulk water should be reasonable.

As shown in Fig. R1(a), the retrieved bulk density values from the proposed algorithm and PIP gradually decrease from nearly 1.0 to 0.1 (g cm$^{-3}$) between 03 and 06 UTC. Both algorithms capture the transition from the mixing-phase to dry snow. Please see Figure R1 in the previous reply. The manuscript has been revised to include the discussion of the bulk water fraction retrieval. **Please see Line 398-417. Or see the reply to "comment A" shown previously, pages 8-9.**

L352: Clarify what is meant by "self-evaluation".

**Reply: The sentence has been revised to "The reflectivity-weighted fall velocity ($V_Z$) of MRR is applied to evaluate the retrieved bulk density and water fraction.". Please see Line 424 in the revised manuscript.**

L357: There's no evidence shown that applying the Vz criteria improves the consistency of retrieved SR with observed SR.

**Reply: The Vz criteria are no longer needed. Please see the revised manuscript, Lines 424-427. Or see the following.**

The reflectivity-weighted fall velocity ($V_Z$) of MRR is applied to evaluate the retrieved bulk density and water fraction. Inconsistency of measured Vz from MRR and calculated Vz from retrieved bulk density from 09 to 15 UTC on 28 February 2018 is noticed. It is postulated that the attenuation effect mainly causes the Vz discrepancy due to the accumulated snow on the MRR antenna.

L359: Is "all available cases" true? SR comparison are shown only for two cases at the sites.

**Reply: The SR comparisons from the cases listed in Table 1 are summarized in Table 3. Please see the revised manuscript, Lines 428-430. Or see the following.**

General consistency between the measured and the bulk density-calculated SR was found in

all available cases (Table 1) of the four sites during the ICE-POP 2018 campaign, as summarized in Table 3.

L364-365: I don't think this statement is supported. This study has investigated two cases which have similar synoptic setups and has found similarity of microphysical characteristics. But you haven't demonstrated that different synoptic setups will produce microphysical characteristics dissimilar to these.

**Reply: The statement has been revised to avoid confusion and improve clarity. "The bulk density and bulk water fraction of two events with warm low synoptic patterns (28 February and 7 March 2018) were investigated." Please see Lines 431-435.**

L366: I would suggest "contrasting" or "dissimilar" rather than "contrastive".

**Reply: The typo has been corrected. Please see Line 436.**

Tables and Figures ******************

Table 3: Note previous comment about "mean bias". Also, why is the Vz criterion for "ALL" shown as "nan"? To help us understand the significance of the biases and standard deviations, please also include the associated mean values and standard deviations of the observed quantities.

**Reply: The "mean bias" has been revised to "bias". The Vz criterion is no longer needed. The mean values of Vz and SR are provided. Please see Table 3 in the revised manuscript.**

Figure 6: Is the colorbar axis labeled correctly? Were there really counts ranging up to 10**50?

**Reply: The typo has been corrected. It should be 10**5. Please see Fig. 6 in the revised manuscript. Fig. 12 also has the same mistake and has also been corrected.**

Figure 12: Why does the mountainous MHS site maintain a population of high-fall-velocity small particles throughout the 7-8 March event?

**Reply: As indicated by the study from Battaglia et al. (2010), Parsivel's fall velocity measurement may not be accurate for a snowflake particle. This is due to the internally assumed relationship between horizontal and vertical snow particle dimensions. Friedrich et al. (2016) indicate that Parsivel can suffer from splashing of particles (observed as a small diameter with large fall velocity when particles fall on the head of the sensor) and margin fallers (observed as a faster velocity than true fall velocity when particles fall through the edge of the sampling area). Yuter et al. (2006), Aikins et al.**

**(2016), and Kim et al. (2021) indicate the splashing and border effects of the diameter of < 1 mm in Parsivel fall velocity measurements. The Parsivel data shown in Figure 12 was quality-controlled, as suggested by Lee et al. (2015). The discussion of fall velocity measurement uncertainty is added in the revised manuscript. Please see Lines 105-109. Or see the following.**

Friedrich et al. (2016) indicate that Parsivel can suffer from splashing of particles (observed as a small diameter with large fall velocity when particles fall on the head of the sensor) and margin fallers (observed as a faster velocity than true fall velocity when particles fall through the edge of the sampling area). The minute Parsivel data was quality-controlled using the fall velocity filtering technique (Lee et al. 2015). The mean fall velocity and standard deviation ($\sigma$) for a given diameter were calculated, and the particles that deviate from the mean fall velocity of more than one standard deviation were filtered.

---

## Author Response (AR4)

Dear Reviewer 1,

The authors sincerely appreciate your valuable comments and suggestions to help improve the manuscript for the second time. We have revised the manuscript titled "Estimating the Snow Density using Collocated Parsivel and MRR Measurements: A Preliminary Study from ICE-POP 2017/2018 ". that was submitted to ACP (Atmospheric Chemistry and Physics) on 3 January 2024. Based on your suggestions, we have put substantial effort into additional analysis. The manuscript has been thoughtfully revised regarding the comments from all reviewers.

Both reviewers are concerned about the maximum ice fraction assumption and the bulk water fraction retrieval. Both reviewers suggest providing more investigations of assuming the maximum ice fraction and deemphasizing (or not overstating) the bulk water fraction retrieval. A sensitivity study of half-maximum ice fraction assumption has been applied and compared with PIP to address reviewers' concerns about bulk water fraction retrieval. The results indicate that the bulk density retrieved from the half-maximum ice fraction assumption is consistently lower than PIP retrievals. On the other hand, the bulk density retrievals from the maximum ice fraction assumption perform better in agreements with PIP than those from the half-maximum ice fraction assumption. The authors also deemphasize the performance of the retrieved bulk water fraction and discuss its uncertainty more in the revised manuscript by adding a discussion paragraph.

Moreover, the discussion of the retrieved density-particle size relationship has been revised in the manuscript. Each study's particle diameter definitions vary due to diverse measuring instruments and principles. The discussion does not intend to emphasize the difference in the density-particle size relationship. Instead of converting various particle diameter definitions, the particle diameter remains as proposed in each study. The definitions of particle size of each study are summarized in Table 4 of the revised manuscript as well.

The manuscript has also been revised carefully following the reviewer's suggestions on English wording and typos. The authors would like to express our sincere appreciation for the comments. The added or modified sentences in the revised manuscript are in red for your convenience. The point-to-point replies to every comment have been prepared in the following. For your convenience, the reply is arranged as follows,

Reviewer's comments

**Response**

Revisions in the manuscript

We would appreciate any feedback on the revisions.

**General comments**

G1: In the first round of reviews, both reviewers raised the issue on the credibility of the method's ability to quantify the liquid and ice volume ratios. While the density retrieval comparison with PIP brings confidence to the presented methods density retrievals that are linked with the liquid water fraction analysis, the choice of maximum vi is not evident but remains somewhat arbitrary in my view. The assumption of dry snow in Huang et al. (2010) is reasonable since they analysed falling snow in temperatures well below freezing. Clearly, the same assumption is not valid in mixed-phase conditions or rain. It is not evident that a "similar" assumption would be valid near melting temperatures that are present in some of the case studies presented in the manuscript. While the novel density retrieval method is well demonstrated, I suggest the authors would be careful with the confidence of the wording when presenting the results of the liquid fraction retrievals. In my view, the validity of the assumption of maximum ice fraction is less than "confirmed", at present. It is not demonstrated in which conditions such assumption would be valid and if the validity would break, e.g., near some temperature threshold. This should be at least noted in the discussion. Further, I suggest adding alternative density retrievals to Fig. 15 using a different assumption on liquid-solid volume fractions to demonstrate the effect on the method's performance.

**Reply: To address reviewers' concerns about the maximum ice fraction assumption and the bulk water fraction retrieval, the authors intend to provide a rational justification for the maximum ice fraction assumption. A half-maximum ice fraction sensitivity study has been applied and compared with PIP. The concept of "retrieving" bulk water fraction has also been deemphasized in the revised manuscript per the reviewer's suggestion.**

**There are two reasons for choosing the maximum ice fraction in the proposed bulk density retrieval algorithm. The first reason is to ensure the "maximum bulk density" is derived. As shown in Fig. 2b, the maximum ice fraction is associated with maximum bulk density. Choosing a lower value of ice fraction subsequently obtains a lower bulk density value. The second reason is that the maximum bulk density with maximum ice fraction assumption has much better agreements of the density-derived SR to the Pluvio observed SR.**

**Moreover, the retrieved bulk density is in good agreement with PIP retrieval. As per the reviewer's suggestion, a sensitivity study of the half-maximum ice fraction assumption is conducted in the revised manuscript. The results indicate that the retrieved bulk density from half-maximum ice fraction is consistently lower than PIP retrieval. On the other hand, the bulk density retrievals from the maximum ice fraction assumption perform better in agreements with PIP than those from the half-maximum ice fraction assumption. The quantitative consistency of retrieved bulk density from maximum ice fraction to PIP gives authors more confidence in the maximum ice fraction assumption. The sensitivity study results are summarized in section 5.2 of the revised manuscript as shown in the following.**

**Please see Lines 389-402 of the revised manuscript.**

5.2 The sensitivity of maximum ice fraction assumption to the bulk density retrieval

The bulk snow density is determined from possible combinations of vi/vw, and the maximum bulk density with maximum ice fraction (vi) is selected in the proposed algorithm. This maximum ice fraction assumption has been applied to the entire ICE-POP data.  As shown in Fig. 2b, choosing the bulk density with minimum ice fraction (e.g., vi=0) leads to the maximum value of vw and minimum bulk density. However, the particle is unlikely to be composed only of water and air. A sensitivity study of selecting a different water/air/ice combination is conducted. The half-maximum ice fraction is selected to derive the retrieved bulk density. As shown in Fig. 15, the retrieved bulk density from the half-maximum ice fraction (red dots) has systematically lower values than the maximum ice fraction (blue dots). The bulk density retrievals from the half-maximum ice fraction have significant discrepancies compared to PIP retrievals.

On the other hand, the density retrieval from the maximum ice fraction assumption has good agreements with PIP retrievals (gray dots). The consistency of retrieved bulk density from the maximum ice fraction to PIP provides more confidence in the assumption of maximum ice fraction. However, the maximum ice fraction assumption may not be valid in a mixed-phased condition when the ice particles melt at a nearly freezing temperature environment. Further investigation is needed in the future study.

**As both reviewers suggest, the revised manuscript also deemphasized the performance of the retrieved bulk water fraction. An additional discussion section (section 5.5) on its uncertainty is added in the revised manuscript. The following discussion has been added.**

**Please see Lines 440-457 of the revised manuscript.**

5.5 The retrieval uncertainty of bulk water fraction

The performance of retrieved bulk density has been quantitatively validated by comparing collocated Pluvio-derived SR and PIP-derived bulk density. On the other hand, quantitative validation of retrieved bulk water fraction is not available due to the limitation of instrumentation. No instrument is capable of directly measuring the bulk water fraction. This study's retrieved bulk water fraction is considered qualitatively reasonable according to the case studies of the 28 February and 7 March 2018 events and the statistical analysis of warm-/cold-low events over coastal and mountain sites (section 4.3–4.5). The distinct bulk density and bulk water fraction retrievals of coastal and mountain sites are revealed. The results indicate that the winter precipitation systems of coastal sites with warmer and moister

environments have higher bulk density and bulk water fraction than mountain sites.

The composition of water/ice/air fraction determines the bulk density. The retrieved bulk water fraction will differ if a different assumption is made when selecting possible bulk density. Therefore, the performance of the retrieved bulk water fraction is partially linked with bulk density retrieval. As shown in Fig. 15a, both the proposed algorithm and PIP capture the fast transition from the mixed-phase ($\rho_{bulk} \approx 1.0$ g cm$^{-3}$) to dry snow ($\rho_{bulk} \approx 0.1$ g cm$^{-3}$). Given the absence of direct measurements of bulk water fraction, the consistency between the retrieved bulk density from the two algorithms is indirect evidence of the qualitative reasonableness of the retrieved bulk water fraction. Combining multiple sophisticated instruments (e.g., 2DVD, PIP, SVI, MASC) and developing a more comprehensive technique can improve our understanding of the critical microphysical characteristics of particles. Further investigation of the particle composition ratio of air/ice/water fraction in different environments is needed.

**In the conclusion, the performance of the proposed method is deemphasized as well. Please see Lines 481-482 of the revised manuscript.**

The consistency of the retrieved bulk density to collocated PIP suggests that the proposed algorithm performs decently in this study.

G2: Referring to discussion raised by Referee #2's questions on compensating for gauge undercatch due to wind, it is recommended adjust for undercatch even with shielded gauges when measuring snow. See e.g. Kochendorfer et al. (2018). I recommend either applying a correction function or simply stating that no correction was applied. It could induce a bias in the order of 10%. Since the precipitation rate is provided only for reference, I consider this only a minor issue.

**Reply: The undercatch adjustment is not applied to the Pluvio data. We have stated that "no under-catch adjustment is applied" in the revised manuscript. Please see Lines 192-193.**

**Specific and technical comments**

L38-39: Suggest "On the other hand, in a warm environment, the melting process induces higher fall velocity due to increased density and aerodynamic effects."

**Reply: The sentence has been revised as per the reviewer's suggestion. Please see Lines 38-39.**

L99: Does "double windshields" refer to double-Alter windshields?

**Reply: The information on the double windshields of each site has been provided in the revised manuscript. Please see Lines 100-102 of the revised manuscript.**

The Pluvios at MHS, BKC, and GWU were equipped with a double windshield with inner Tretyakov and outer Alter shield. The Pluvio at YPO was equipped with a Belfort double alter windshield.

L215-216: I'm worried about the lack of transparency in how much data was omitted from the analysis by this visual examination. Could you provide a statistic on this? Further, I suggest marking the omitted data and analysis results in Figures 5, 7-11, e.g., with a grey mask.

**Reply: The retrieved bulk density subjected to the attenuation effect is shown by a grey mask in the revised manuscript. Please see Figures 5, 7-11 of the revised manuscript.**

L148: As I understand it, in the T-matrix simulations, you assume a standard deviation of 20° for the canting angle of spherical particles. Referring to my comment on the previous round of reviews, I still fail to understand what is the definition of a canting angle of a spherical particle and how is it relevant?

**Reply: As introduced in the manuscript, the proposed algorithm assumes a symmetric sphere particle. The definition of a spherical particle's canting angle is irrelevant. The values of the canting angle do not affect the reflectivity from the T-matrix simulation of a symmetric sphere particle. The canting angle affects the reflectivity simulation as the axis ratio 0.5 is applied in the discussion section (section 5.4). The description has been revised to improve the clarity.**

**Please see Lines 148-150 of the revised manuscript.**

The shape of the hydrometeor is regarded as a symmetric sphere since the $Z_{HH}$ measurement of the hydrometer was observed from the bottom of the snow particle by vertical pointing MRR. No canting angle is considered.

**Please see Lines 425-426 of the revised manuscript.**

A sensitivity investigation assuming the particle axis ratio of 0.5 and the mean and standard

deviation of the canting angle are 0° and 20°, shows that about 1.5 dBZ variation of MRR reflectivity can be induced.

L288: Suggest "relatively lower" to 'lower'. Comparison is always relative.

**Reply: The "relatively lower" has been revised to "lower" per the reviewer's suggestion. Please see Line 295.**

L296, L438: It is incorrect to say that the different sites would represent differrent synoptic environments. Synoptic scale is in the order of 1000km, while the sites seem to be within 100km of each other. The term "mesoclimate" would be more fitting.

**Reply: The term "synoptic" has been revised to "mesoclimate" per the reviewer's suggestion. Please Lines 303 and 478.**

L306-308: Not completely sure if I understood this sentence. Did you mean that the finding of decreased bulk density is supported by the decrease in average fall velocity?

**Reply: Yes, the manuscript intends to indicate that decreased bulk density is supported by the decrease in average fall velocity. In order to improve the manuscript's clarity, the sentence has been revised. Please see Lines 313-315.**

The decrease of averaged fall velocity between 08 to 19 UTC on 7 March (Figs. 12a, b) and 19 UTC to 03 UTC on 8 March (Figs. 12d, e) is consistent with the decreasing density in MHS and BKC sites.

L408: It's unclear to me what is meant here. Do you mean that the agreement with density derived from PIP would be worse with a different assumption?

**Reply: Choosing the maximum ice fraction in the retrieval algorithm ensures the maximum bulk density is derived. As shown in Fig. 2b, the maximum ice fraction is associated with maximum bulk density. Choosing a lower value of ice fraction subsequently obtains a lower bulk density value. The maximum bulk density with maximum ice fraction assumption has much better agreements of the density-derived SR to the Pluvio observed SR. Moreover, the retrieved bulk density has good agreements with PIP retrieval. As per the reviewer's suggestion, a sensitivity study of the ice fraction assumption is conducted in the revised manuscript. The results indicate that the retrieved bulk density from half-maximum ice fraction is consistently lower than PIP retrieval. The**

**consistency of retrieved bulk density from the maximum ice fraction to PIP provides more confidence in the assumption of maximum ice fraction.**

**Please also see the reply to General Comment 1(G1).**

L409-411: Suggest "Given the absence of direct measurements of bulk water fraction, the consistency between the retrieved bulk density from the two algorithms serves as indirect evidence of the reasonableness of the retrieved bulk water fraction."

**Reply: The sentence has been revised as per the reviewer's suggestion. Please see Lines 452-454 of the revised manuscript.**

L413: "mixing-phase" to 'mixed-phase'.

**Reply: The "mixing-phase" has been revised to "mixed-phase" per the reviewer's suggestion. Please see Line 385.**

L414-416: Suggest "Despite various potential factors that could compromise bulk density retrieval from collocated MRR and Parsivel instruments, the uncertainty study indicates that observational errors have a reasonably low effect, maintaining acceptable performance."

**Reply: The sentence has been revised as per the reviewer's suggestion. Please see Lines 436-438 of the revised manuscript.**

Table 3: How is "Mean of Pluvio" defined? What are the units?

**Reply: "Mean of Pluvio" indicates the mean values of SR from available comparison data for each site. Please see Line 462 of the revised manuscript.**

Figure 2, L628: By definition, the value of Z_HH does not change on (along) its contours. Please rephrase.

**Reply: The sentence has been removed to avoid confusion. Please see the Lines 486-487 of the revised manuscript.**

Figures 5d, 9d-11d: Why does the measured temperature sometimes remain constant for extended periods of several hours? E.g., in Fig. 5d, between 13 and 17 UTC, the temperature reading seems to be stuck at -1°C without any fluctuation. Could you give a short explanation

for this behavior? If it is a measurement error, remove the erroneous parts from the figures.

**Reply: During ICE-POP, only the MHS site is equipped with WXT520. The temperature data of other sites are from the collocated POSS (Precipitation Occurrence Sensor System). In the previous manuscript, incorrect temperature data was utilized in Fig. 5. The correct temperature data is used in Fig. 5 of the revised manuscript. In Figs. 9-11, some constant temperatures can be seen. The trend of the temperature helps interpolate the density change. The authors would like to keep it.**

Figure 14: Is the percentage on the color bar integrated over time or particle volume?

**Reply: The percentage on the colorbar is integrated over time.**

Figure 14, L731-732: In my understanding, "number density function" refers to the function that describes also the shape of the PSD. Hence, I suggest using the term "number concentration" here, instead.

**Reply: The "number density function" has been revised to "number concentration" in the revised manuscript.**

Dear Reviewer 2,

The authors sincerely appreciate your valuable comments and suggestions to help improve the manuscript for the second time. We have revised the manuscript titled "Estimating the Snow Density using Collocated Parsivel and MRR Measurements: A Preliminary Study from ICE-POP 2017/2018 ". that was submitted to ACP (Atmospheric Chemistry and Physics) on 3 January 2024. Based on your suggestions, we have put substantial effort into additional analysis. The manuscript has been thoughtfully revised regarding the comments from all reviewers.

Both reviewers are concerned about the maximum ice fraction assumption and the bulk water fraction retrieval. Both reviewers suggest providing more investigations of assuming the maximum ice fraction and deemphasizing (or not overstating) the bulk water fraction retrieval. A sensitivity study of half-maximum ice fraction assumption has been applied and compared with PIP to address reviewers' concerns about bulk water fraction retrieval. The results indicate that the bulk density retrieved from the half-maximum ice fraction assumption is consistently lower than PIP retrievals. On the other hand, the bulk density retrievals from the maximum ice fraction assumption perform better in agreements with PIP than those from the half-maximum ice fraction assumption. The authors also deemphasize the performance of the retrieved bulk water fraction and discuss its uncertainty more in the revised manuscript by adding a discussion paragraph.

Moreover, the discussion of the retrieved density-particle size relationship has been revised in the manuscript. Each study's particle diameter definitions vary due to diverse measuring instruments and principles. The discussion does not intend to emphasize the difference in the density-particle size relationship. Instead of converting various particle diameter definitions, the particle diameter remains as proposed in each study. The definitions of particle size of each study are summarized in Table 4 of the revised manuscript as well.

The manuscript has also been revised carefully following the reviewer's suggestions on English wording and typos. The authors would like to express our sincere appreciation for the comments. The added or modified sentences in the revised manuscript are in red for your convenience. The point-to-point replies to every comment have been prepared in the following. For your convenience, the reply is arranged as follows,

Reviewer's comments

**Response**

Revisions in the manuscript

We would appreciate any feedback on the revisions.

############# Opening comments #############

I do still have a substantial concern about the method used to decompose the bulk density into an ice volume fraction ("bulk ice fraction", vi) and a liquid volume fraction ("bulk water fraction", vw). The authors do not provide a physical basis for the approach they use. Instead, they use what seems to be an ad-hoc requirement to obtain the smallest possible bulk water fraction given the ZHH and the retrieved $\rho_{bulk}$.

It is clear from equation (2) that ZHH is a function of $\rho_{bulk}$, so I think the part of the study related to determining $\rho_{bulk}$ is reasonable. But it is also clear from equation (2) that ZHH provides no information that could be used to distinguish bulk ice and water fractions. From equation (1), the best that can be obtained is a linear relationship between vi and vw. An error in vw could be compensated by an offsetting error in vi to give an accurate $\rho_{bulk}$. Thus an accurately-retrieved $\rho_{bulk}$ does not indicate or imply that an estimated vw is correct.

This concern could be addressed if the authors can provide a rational justification for the approach they have taken. Perhaps there are reasons that they have decided to select the solutions that provide minimum vw. Why is it desirable to choose the minimum vw solution? Note my comment below for lines 164-171 of the revised article. If they have sound reasons and can elaborate on those in the methodology, that would address this concern.

If the authors are unable to do this, I think the proper approach would be for the authors to deemphasize their claim of "retrieving" bulk water fraction and instead state that their analyses are for one possible approach to selecting the bulk water fraction.

**Reply: To address reviewers' concerns about the maximum ice fraction assumption and the bulk water fraction retrieval, the authors intend to provide a rational justification for the maximum ice fraction assumption. A half-maximum ice fraction sensitivity study has been applied and compared with PIP. The concept of "retrieving" bulk water fraction has also been deemphasized in the revised manuscript per the reviewer's suggestion.**

**There are two reasons for choosing the maximum ice fraction in the proposed bulk density retrieval algorithm. The first reason is to ensure the "maximum bulk density" is derived. As shown in Fig. 2b, the maximum ice fraction is associated with maximum bulk density. Choosing a lower value of ice fraction subsequently obtains a lower bulk density value. The second reason is that the maximum bulk density with maximum ice fraction assumption has much better agreements of the density-derived SR to the Pluvio observed SR.**

**Moreover, the retrieved bulk density is in good agreement with PIP retrieval. As per the reviewer's suggestion, a sensitivity study of the half-maximum ice fraction assumption is conducted in the revised manuscript. The results indicate that the retrieved bulk density from half-maximum ice fraction is consistently lower than PIP retrieval. On the other hand, the bulk density retrievals from the maximum ice fraction assumption perform better in agreements with PIP than those from the half-maximum ice fraction assumption. The quantitative consistency of retrieved bulk density from maximum ice fraction to PIP gives authors more confidence in the maximum ice fraction assumption. The sensitivity study results are summarized in section 5.2 of the revised manuscript as shown in the following.**

**Please see Lines 389-402 of the revised manuscript.**

5.2 The sensitivity of maximum ice fraction assumption to the bulk density retrieval

The bulk snow density is determined from possible combinations of vi/vw, and the maximum bulk density with maximum ice fraction (vi) is selected in the proposed algorithm. This maximum ice fraction assumption has been applied to the entire ICE-POP data. As shown in Fig. 2b, choosing the bulk density with minimum ice fraction (e.g., vi=0) leads to the maximum value of vw and minimum bulk density. However, the particle is unlikely to be composed only of water and air. A sensitivity study of selecting a different water/air/ice combination is conducted. The half-maximum ice fraction is selected to derive the retrieved bulk density. As shown in Fig. 15, the retrieved bulk density from the half-maximum ice fraction (red dots) has systematically lower values than the maximum ice fraction (blue dots). The bulk density retrievals from the half-maximum ice fraction have significant discrepancies compared to PIP retrievals.

On the other hand, the density retrieval from the maximum ice fraction assumption has good agreements with PIP retrievals (gray dots). The consistency of retrieved bulk density from the maximum ice fraction to PIP provides more confidence in the assumption of maximum ice fraction. However, the maximum ice fraction assumption may not be valid in a mixed-phased condition when the ice particles melt at a nearly freezing temperature environment. Further investigation is needed in the future study.

**As both reviewers suggest, the revised manuscript also deemphasized the performance of the retrieved bulk water fraction. An additional discussion section (section 5.5) on its uncertainty is added in the revised manuscript. The following discussion has been added.**

**Please see Lines 440-457 of the revised manuscript.**

5.5 The retrieval uncertainty of bulk water fraction

The performance of retrieved bulk density has been quantitatively validated by comparing collocated Pluvio-derived SR and PIP-derived bulk density. On the other hand, quantitative validation of retrieved bulk water fraction is not available due to the limitation of instrumentation. No instrument is capable of directly measuring the bulk water fraction. This study's retrieved bulk water fraction is considered qualitatively reasonable according to the case studies of the 28 February and 7 March 2018 events and the statistical analysis of warm-/cold-low events over coastal and mountain sites (section 4.3–4.5). The distinct bulk density and bulk water fraction retrievals of coastal and mountain sites are revealed. The results indicate that the winter precipitation systems of coastal sites with warmer and moister environments have higher bulk density and bulk water fraction than mountain sites.

The composition of water/ice/air fraction determines the bulk density. The retrieved bulk water fraction will differ if a different assumption is made when selecting possible bulk density. Therefore, the performance of the retrieved bulk water fraction is partially linked with bulk density retrieval. As shown in Fig. 15a, both the proposed algorithm and PIP capture the fast transition from the mixed-phase ($\rho_{bulk} \approx 1.0$ g cm$^{-3}$) to dry snow ($\rho_{bulk} \approx 0.1$ g cm$^{-3}$). Given the absence of direct measurements of bulk water fraction, the consistency between the retrieved bulk density from the two algorithms is indirect evidence of the qualitative reasonableness of the retrieved bulk water fraction. Combining multiple sophisticated instruments (e.g., 2DVD, PIP, SVI, MASC) and developing a more comprehensive technique can improve our understanding of the critical microphysical characteristics of particles. Further investigation of the particle composition ratio of air/ice/water fraction in different environments is needed.

**In the conclusion, the performance of the proposed method is deemphasized as well. Please see Lines 481-482 of the revised manuscript.**

The consistency of the retrieved bulk density to collocated PIP suggests that the proposed algorithm performs decently in this study.

########## Line-by-line comments on new revision ###############

L 28: Do you specifically mean liquid water content here? Do you instead mean just water content, since remote sensors observe both ice and liquid hydrometeors.

**Reply: To improve the clarity of the manuscript. The "liquid water content (LWC)" has been revised to "liquid/ice water content (LWC/IWC)" as per the reviewer's suggestion. Please see Line 28.**

L 38-39: Since density is not related to the aerodynamic process, maybe rewrite this as ... induces higher density as well as higher fall velocity by the aerodynamic process.

**Reply: The sentence has been revised according to both reviewers' suggestions. The revised sentence is as follows. Please see Lines 38-39 in the revised manuscript.**

On the other hand, in a warm environment, the melting process induces higher density as well as higher fall velocity by the aerodynamic process.

L 107: The word minute by itself in English is often used to mean small or tiny. I suggest using one- minute instead, which has the desired meaning of a sample of length one minute of time, here and at other locations in the paper.

**Reply: The "minute Parsivel data" has been revised to "one-minute Parsivel data" per the reviewer's suggestion. Please see Line 109 in the revised manuscript.**

L 111: Is the ICEP-POP that is used here intentional, rather than ICE-POP?

**Reply: The typo has been corrected. Please see Line 113 in the revised manuscript.**

L 144-147: If each hydrometeor is truly regarded as a symmetric sphere and ice and water are assumed to be evenly distributed within the particle, canting angle is not relevant - the scattering properties will not change with respect to any rotation of the sphere. Why are canting angles considered? Were the particles not actually symmetric spheres? Please enhance this description to be clear and correct about what is being assumed for the calculation of the scattering properties.

**Reply: As introduced in the manuscript, the proposed algorithm assumes a symmetric sphere particle. The definition of a spherical particle's canting angle is irrelevant. The values of the canting angle do not affect the reflectivity from the T-matrix simulation of a symmetric sphere particle. The canting angle affects the reflectivity simulation as the axis ratio 0.5 is applied in the discussion section (section 5.4). The description has been revised to improve the clarity.**

**Please see Lines 148-150 of the revised manuscript.**

The shape of the hydrometeor is regarded as a symmetric sphere since the $Z_{HH}$ measurement of the hydrometer was observed from the bottom of the snow particle by vertical pointing MRR. No canting angle is considered.

**Please see Lines 425-426 of the revised manuscript.**

A sensitivity investigation assuming the particle axis ratio of 0.5 and the mean and standard deviation of the canting angle are 0° and 20°, shows that about 1.5 dBZ variation of MRR reflectivity can be induced.

L 164-171: This description of the methodology is the point of my most significant concern with the study. To obtain distinct vi and vw, the authors make an ad-hoc choice to pick the solution with the maximum $\rho_{bulk}$ and vi. The justification they provide is that this is similar to Huang et al. (2010), which assumes the particles are only ice and air. My opinion is that this justification is not sufficient to allow a claim that vw is being retrieved.

**Reply: Please see the reply to the opening comment. Also, the following sentence is added to the revised manuscript.**

**Please see Line 172 of the revised manuscript.**

The impact of ice fraction assumption on bulk density retrieval will be investigated in the discussion section.

L 172-173 and 188: Per the statements by the authors here, the validation approaches that are being used are to validate the retrieved bulk density, not the bulk water fraction.

**Reply: The study validates the proposed method's retrievals using the "bulk density" related parameters SR and Vz. A direct comparison of the bulk water fraction is not available. The sentence has been revised to improve the clarity.**

**Please see Line 173-174 of the revised manuscript.**

Since a direct comparison of the bulk water fraction is unavailable, this study will use two approaches to evaluate the bulk density derived from the proposed method.

L 184-185: Reflectivity-weighted velocity (for comparison to radar Doppler velocity) is more often seen calculated from PSDs as

$$V_Z^{\rho_{bulk}} = \frac{\sum \sigma_{bk}(D_i)\,V(D_i)\,N(D_i)\,dD_i}{\sum \sigma_{bk}(D_i)\,N(D_i)\,dD_i} \quad (4)$$

where $\sigma_{bk}(D_i)$ is the backscatter cross-section for particles in size bin i. It's not clear here what is meant by Z(ρbulk,D). Please add some description of Z(ρbulk,D), how it is calculated and whether your formula gives results that are the same as this more typical formula. If not the same, the comparisons of V$^{\rho bulk}$ and V$^{MRR}$ may be of concern.

**Reply: The authors have examined the calculation of Vz and confirmed that the calculation of the Reflectivity-weighted velocity utilized the $\sigma_{bk}(D_i)$ rather than the Z(ρbulk,D). The manuscript has been revised to improve clarity. Please see Lines 185-186 of the revised manuscript.**

L 196: Per the reference, Kim et al. (2021), equation (6) gives the volume-weighted mean diameter, not the mass-weighted mean diameter. The Kim et al. statement seems correct, since equation (6) gives the ratio of the fourth moment of the PSD to its third moment.

**Reply: In Kim et al. (2021), the "characteristic diameter, $D_{m'}$" is defined as the following equation.**

$$D_{m'} = \frac{\int_{D_{min}}^{D_{max}} D^4\,N(D)dD}{\int_{D_{min}}^{D_{max}} D^3\,N(D)dD} \quad (6).$$

**The authors concur with the reviewer's suggestion. To improve the manuscript's clarity, volume-weighted mean diameter ($D_v$) is replacing the $D_m$ in the revised manuscript. Please see Lines 198-200 of the revised manuscript. Also, the Fig. 13 has been revised.**

L 211: I think this should be ... and CPO are slightly lower.

**Reply: The typo has been revised. Please see Line 215.**

L 220: Regarding various measurement issues that induce inconsistency, please be more explicit by stating what are these issues.

**Reply: As the reviewer suggested, the sentence has been revised to state those issues explicitly. Please see Lines 224-227 of the revised manuscript.**

For example, Battaglia et al. (2010) indicated Parsivel's fall velocity measurement error due to the internally assumed relationship between horizontal and vertical snow particle dimensions. The low SNR of MRR reduces Vz measurement quality. In addition, the sampling volume discrepancy increases the Vz inconsistency.

L 264: Please check Figure 5. I do not see a gray area.

**Reply: The figure has been corrected. The gray area representing the data with MRR attenuation effect has been shown in Fig. 5, 7-11. Please see the revised manuscript.**

L 291-292 and 295-296: I think these lines overstate the interpretation of Figure 12 somewhat. I agree that the distributions in Figure 12 do show changes in fall velocity-diameter relationships. It is probably OK to say that the particular changes in the relationships are consistent with increases in ρbulk which could be associated with increases in bulk water fraction resulting, for example, from melting of particles. But I believe it is an overstatement to say that gradual increases in density, as well as the bulk water fraction can be found in the V (D) distributions in Figure 12 (L291-292) or that the retrieved bulk density and bulk water fraction reveal distinct V (D) relations (L295-296).

**Reply: The authors intend to indicate that changes in fall velocity-diameter relation are consistent with bulk density and bulk water fraction increases. The sentence has been revised to not "overstate" the interpretation of Fig. 12. Please see Lines 298-299, and Lines 302-303 of the revised manuscript.**

**Original L 291-292:**

**Gradual increase of density, as well as the bulk water fraction, can also be found in the distribution of fall velocity versus the diameter from MHS, BKC, to the GWU sites (Fig. 12).**

**Revised: Lines 298-299 of the revised manuscript.**

Gradual increase of density, as well as the bulk water fraction, can also be found consistent with an increase in the distribution of fall velocity versus the diameter from MHS, BKC, to the GWU sites (Fig. 12).

**Original L295-296:**

**Overall, the retrieved bulk density and bulk water fraction successfully reveal distinct fall velocity-diameter relations of each site due to the different synoptic environments.**

**Revised: Lines 302-303 of the revised manuscript.**

Overall, the retrieved bulk density and bulk water fraction qualitatively reveal distinct fall velocity-diameter relations of each site due to the different mesoclimate environments.

L 303: I'd suggest transitioned rather than transited.

**Reply: The "transited" has been revised to "transitioned" per the reviewer's suggestion. Please see Line 309.**

L 304: Should this be at other sites instead of as other sites?

**Reply: The "as" has been revised to "at" per the reviewer's suggestion. Please see Line 311.**

L 304-305: See my earlier comment regarding L 291-292. Saying that the V (D) relation is consistent with the bulk water fraction seems more appropriate.

**Reply: Please see previous reply to L291-292 and 295-296. The authors intend to indicate that the changes in fall velocity-diameter relation are consistent with increases in bulk density and bulk water fraction. The revised manuscript has added one sentence to "not overstate" the relation.**

**Please see Lines 313-316 of the revised manuscript.**

The decrease of averaged fall velocity between 08 to 19 UTC on 7 March (Figs. 12a, b) and 19 UTC to 03 UTC on 8 March (Figs. 12d, e) is consistent with the decreasing density in MHS and BKC sites.

L 326: See my earlier comment regarding L 196 and what is actually calculated by equation (6).

**Reply: The authors concur with the reviewer's suggestion. The "characteristic diameter, $D_{m'}$" Kim et al. (2021) should be defined as volume-weighted mean diameter ($D_v$). In order to improve the clarity of the manuscript, volume-weighted mean diameter ($D_v$) is replacing the $D_m$ in the revised manuscript. Please see Lines 198-200 of the revised manuscript. Also, the Fig. 13 has been revised.**

L 334: Where does this relationship between $D_m$ and $D_0$ come from? Is there a reference?

**Reply: As mentioned in the previous revision, the bulk density comparison in Fig. 13 does not intend to emphasize the difference in the density-particle size relationship. The authors checked the earlier studies, as summarized in Brandes et al. (2007). The particle diameter definitions vary in each study. Instead of converting various particle diameter definitions, the particle diameter remains as proposed in each study. The density-particle size relationships and the particle diameter definitions are summarized in Table 4 of the revised manuscript. The manuscript has been revised to improve the clarity.**

**Please see Lines 341-346 of the revised manuscript.**

Early studies, namely Magono and Nakamura (1965), Holroyd (1971), Muramoto et al. (1995), and Fabry and Szyrmer (1999), have documented various density-particle size relationships (Table 4 and Fig. 13). The particle diameter definitions vary in each study (Table 4). Instead of converting various particle diameter definitions, the particle diameter remains as proposed in each study. Despite distinct environmental conditions, instrumentations, and retrieval techniques, most of the particles in this study are consistent with the density-particle size relationship from previous studies.

L 359: See my opening comments and concerns along with the related line-by-line comments regarding the ability of the retrieval to determine bulk water fraction. This statement also falls under that concern and should be addressed.

**Reply: Please see the reply to the opening comment. As per the reviewer's suggestion, the revised manuscript adds a sensitivity study and deemphasizes the claim of "retrieving" bulk water fraction.**

L 366: While Battaglia et al. do discuss Parsivel fallspeed errors, I don't believe they are discussed in Wood et al.

**Reply: The reviewer suggested the reference to Wood et al. (2013) in the previous submission. Even though Wood et al. (2013) discussed the PSD observation error from 2DVD, authors consider both Parsivel and 2DVD to have similar measuring principles and share the same issue. In addition, the correction factor (CF) derived from comparing the collocated 2DVD and Parsivel in the MHS site is derived to discuss the Parsivel PSD uncertainty. The sentence has been revised to improve clarity.**

**Please see Lines 404-405 of the revised manuscript.**

As indicated by the study by Battaglia et al. (2010) and Wood et al. (2013), Parsivel and 2DVD have various issues in snowflake particle measurement.

L 376: Especially since this approach of using CF is from personal communications and not from a published reference, the values of the particle size-dependent CF and the method by which its values are determined should be documented here, to allow the results to be reproduced.

**Reply: The values of CF are summarized in Table 5 of the revised manuscript.**

L 410-411: No, I don't think it is justied to say that since the bulk densities are in agreement with those from the PIP, the bulk water fractions are confirmed. $Z_{HH}$ is dependent on bulk density in a way that makes it not possible to discriminate the contributions of $v_i$ and $v_w$. Since bulk density depends on both $v_i$ and $v_w$, offsetting errors in $v_i$ and $v_w$ could still give a correct $\rho_{bulk}$.

**Reply: Please see the previous reply to the opening comment. As per the reviewer's suggestion, the revised manuscript adds a sensitivity study and deemphasizes the claim of "retrieving" bulk water fraction.**

L 413: Usually mixed-phase rather than mixing-phase.

**Reply: The type has been corrected as per the reviewer's suggestion. Please see Line 385 of the revised manuscript.**

L 416-417: See comment regarding L 410-411.

**Reply: Please see the previous reply to the opening comment.**

L 433: There appears to be an incomplete sentence here: The retrieved bulk density.

**Reply: The incomplete sentence has been removed.**

L 441: Again, see and address my overall comments regarding retrieval of bulk water fraction.

**Reply: Please see the previous reply to the opening comment. The sentence has been modified, and one sentence has been added to deemphasize the retrieval of the bulk water fraction.**

**Please see Lines 485-486 of the revised manuscript.**

Moreover, the proposed algorithm in this study provides a possible approach to estimating the bulk water fraction.

L 444: Usually unattended rather than unattentively.

**Reply: The type has been corrected as per the reviewer's suggestion. Please see Line 484 of the revised manuscript.**

General comments

Does EGU have a policy on including information about where to obtain the input datasets used for the study presented in the paper? Is a data availability statement required? In the acknowledgements, I note that the source of the PIP data is not mentioned.

**Reply: The PIP data is provided by Dr. Tokay. Please see Lines 496-497 of the revised manuscript.**

---

## Author Response (AR5)

Dear Editor,

The authors sincerely appreciate your valuable comments and suggestions to help improve the manuscript. We have revised the manuscript titled "Estimating the Snow Density using Collocated Parsivel and MRR Measurements: A Preliminary Study from ICE-POP 2017/2018". that was submitted to ACP (Atmospheric Chemistry and Physics) on 3 January 2024. Based on your suggestions, we have added additional discussion to the wind-induced undercatch issue of Pluvio in the revised manuscript.

Other than applying the undercatch correction to single/no-shield Pluvio measurements, all sites were equipped with double windshields and DFIR for the MHS site to mitigate wind-induced undercatch issues during ICE-POP 2017/2018. The Pluvio at YPO was equipped with a Belfort double Alter windshield. The Pluvios at MHS, BKC, and GWU were equipped with a double windshield with inner Tretyakov and outer Alter shields. The Pluvio at the MHS was within the DFIR (double fence intercomparison reference) in addition to the double shield. The studies from Kochendorfer et al. (2017; 2018) have shown that the SDFIR (small DFIR) and the Belfort double-Alter windshield have much smaller uncorrected biases and also smaller adjusted RMSE relative to the corresponding reference.

Even though the double windshields have been applied to each site and DFIR to the MHS site, some Pluvio measurement bias caused by the undercatch issue may remain. Noticeable discrepancies between density-derived and Pluvio-observed SR (liquid-equivalent snowfall rate) can be found in BKC and GWU sites (Figure 4 and Table 3 of the manuscript), which were equipped with ordinary inner Tretyakov and outer Alter shields. On the other hand, the YPO (Belfort double-Alter windshield) and MHS (double windshield with DFIR configuration), which are ideal for reducing wind-induced undercatch issues, have better agreements of density-derived and Pluvio-observed SR (Figure 4 and Table 3 of the manuscript).

The main goal of this study is to propose a robust method to derive bulk density and bulk water fraction of a population of particles from collocated measurements of MRR and Parsivel. Despite wind-induced undercatch issues being mostly mitigated by double windshields and some SR discrepancies remaining noticeable, the density retrieved from the proposed algorithm shows good agreement with PIP retrieval at the MHS site. The MHS site had a double windshield with inner Tretyakov and outer Alter shields. The instruments at the MHS were within the DFIR in addition to the double shields to mitigate the wind-induced undercatch issues. Pluvio's wind-induced undercatch issues are beyond the scope of this study. A discussion of wind-induced undercatch issues has been added to the revised manuscript.

The manuscript has also been revised carefully. The authors would like to express our sincere appreciation for the comments. The added or modified sentences in the revised manuscript are in red for your convenience. We would appreciate any feedback on the revisions.

############# Opening comments #############

**I do have a concern that I believe should be more carefully addressed. An issue raised by Reviewer 1 is that there is a tendency for under-catch by the Pluvio precipitation gauge which was used for validation. The reviewer estimated that this issue is relatively minor. I'm not sure I agree. You may wish to consider the paper by Colli et al. 2020 "Adjustments for Wind-Induced Undercatch in Snowfall Measurements Based on Precipitation Intensity". Collection efficiency can drop by at least one half in even modest winds. Currently the article does not address collection efficiency concerns seriously and I believe it should.**

**Reply:**

As the reviewers and the editor indicated, wind plays a dominant role in reducing the gauge collection efficiency (CE) in snowfall measurements. The DFIR (double fence intercomparison reference) has been considered as the manual reference configuration for investigating wind-induced undercatch issues (Kochendorfer et al. 2017; Colli et al. 2020). Studies (Kochendorfer et al. 2017; 2018; 2022; and Colli et al. 2020) have utilized DFIR to obtain the CE under different environmental conditions. The wind-induced undercatch (CE < 1) is most severe under high wind speed and light snow intensity (SI) conditions. Colli et al. (2020) suggest that the wind-induced undercatch is unneglectable for high wind speed > 4 m s$^{-1}$ and SI < 0.6 mm hr$^{-1}$ when a single Alter windshield is equipped.

During ICE-POP 2017/2018, the undercatch issue was mitigated by applying double windshields. All of the Pluvios were equipped with double windshields. The Pluvio at YPO was equipped with a Belfort double Alter windshield. The Pluvios at MHS, BKC, and GWU were equipped with a double windshield with inner Tretyakov and outer Alter shields. The Pluvio at the MHS was within the DFIR (double fence intercomparison reference) in addition to the double shield. The studies from Kochendorfer et al. (2017; 2018) have shown that the SDFIR (small DFIR) and the Belfort double-Alter windshield have much smaller uncorrected biases and also smaller adjusted RMSE relative to the corresponding reference.

Due to unavoidable logistic and maintenance issues, the collocated AWS measurements Pluvio to MRR and Parsivel were only sometimes available for each site (YPO, MHS, BKC, and GWU) during ICE-POP 2017/2018. Therefore, wind-induced undercatch correction cannot be implemented, as Colli et al. (2020) suggested. The closest AWS data of each site is utilized

to characterize the background environmental condition of each site and analyze snow density. However, it's not adequate to be applied for wind-induced undercatch correction. According to the closest AWS data of each site, most SI is about 0.0 to 1.0 mm hr$^{-1}$ during ICE-POP 2017/2018 (Figure 4 of the revised manuscript). 74% of the wind speed is less than 4 m s$^{-1}$ (See Figure R1 of the reply). The Pluvio at the YPO site equipped Belfort double-Alter shield is more effective at mitigating wind-induced undercatch than the standard double-Alter shield (Kochendorfer et al. 2017; 2018;). In addition, the Pluvio at the MHS was within the DFIR.

The BKC and GWU sites, equipped with ordinary inner Tretyakov and outer Alter shields, do exhibit noticeable discrepancies between density-derived and Pluvio-observed SR, as expected (Figure 4 and Table 3 of the manuscript). On the other hand, the YPO and MHS with double windshield configuration, which is ideal for reducing wind-induced undercatch issues, have better agreements between density-derived and Pluvio-observed SR (Figure 4 and Table 3 of the manuscript).

A discussion of wind-induced undercatch issues has been added to the revised manuscript. Please see Lines 242-252 of the revised manuscript. Or see the following.

In addition to MRR attenuation, part of the inconsistency can be attributed to Pluvio-observed SR bias caused by the wind-induced undercatch issues (Kochendorfer et al. 2017; 2018; 2022; and Colli et al. 2020). Kochendorfer et al. (2017) and Colli et al. (2020) have proposed wind-speed-based undercatch correction algorithms for single/no-shield instruments. In this study, instead of applying the undercatch correction to single/no-shield Pluvio measurements, all sites were equipped with double windshields to mitigate wind-induced undercatch issues during ICE-POP 2017/2018 (see section 2). Kochendorfer et al. (2017; 2018;) indicate that the SDFIR (small DFIR) and the Belfort double-Alter windshield have much smaller uncorrected biases and also smaller adjusted RMSE relative to the corresponding reference. Kochendorfer et al. (2018) show that the collection efficiency (CE) for the Belfort double-Alter windshield (YPO site) is about 0.9 at a wind speed of 4 m s$^{-1}$. On the other hand, the CE of the double-Alter windshield (BKC and GWU sites) is dropped to 0.7 at a wind speed of 4 m s$^{-1}$. The MHS site with DFIR has fewer undercatch issues. It's postulated that wind-induced undercatch issues partially contribute to the discrepancies between density-derived and measured SR. Further investigation of the wind-induced undercatch issues is needed.

A description of the double windshield and DFIR at the MHS site to mitigate the wind-induced undercatch issues is added in the conclusion section of the revised manuscript. Please see Lines 493-495 of the revised manuscript. Or see the following.

The MHS site had a double windshield with inner Tretyakov and outer Alter shields. The instruments at the MHS were within the DFIR in addition to the double shields to mitigate the wind-induced undercatch issues.

[Figure]

Figure R1: The CFD of wind speed from the closest AWS to YPO, JMS, BKC, and GWU sites. The CFD values of 3 m s$^{-1}$ and 4 m s$^{-1}$ are 60.76% and 73.71%, respectively.